# Mechanism Design for Collaborative Normal Mean Estimation

**Yiding Chen**
UW-Madison
ychen695@wisc.edu

**Xiaojin Zhu**
UW-Madison
jerryzhu@cs.wisc.edu

**Kirthevasan Kandasamy**
UW-Madison
kandasamy@cs.wisc.edu

## Abstract

We study collaborative normal mean estimation, where $m$ strategic agents collect i.i.d samples from a normal distribution $\mathcal{N}(\mu, \sigma^2)$ at a cost. They all wish to estimate the mean $\mu$. By sharing data with each other, agents can obtain better estimates while keeping the cost of data collection small. To facilitate this collaboration, we wish to design mechanisms that encourage agents to collect a sufficient amount of data and share it truthfully, so that they are all better off than working alone. In naive mechanisms, such as simply pooling and sharing all the data, an individual agent might find it beneficial to under-collect and/or fabricate data, which can lead to poor social outcomes. We design a novel mechanism that overcomes these challenges via two key techniques: first, when sharing the others' data with an agent, the mechanism corrupts this dataset proportional to how much the data reported by the agent differs from the others; second, we design minimax optimal estimators for the corrupted dataset. Our mechanism, which is Nash incentive compatible and individually rational, achieves a social penalty (sum of all agents' estimation errors and data collection costs) that is at most a factor 2 of the global minimum. When applied to high dimensional (non-Gaussian) distributions with bounded variance, this mechanism retains these three properties, but with slightly weaker results. Finally, in two special cases where we restrict the strategy space of the agents, we design mechanisms that essentially achieve the global minimum.

## 1 Introduction

With the rise in popularity of machine learning, data is becoming an increasingly valuable resource for businesses, scientific organizations, and government institutions. However, data collection is often costly. For instance, to collect data, businesses may need to carry out market research, scientists may need to conduct experiments, and government institutions may need to perform surveys on public services. However, once data has been generated, it can be freely replicated and used by many organizations [20]. Hence, instead of simply collecting and learning from their own data, by sharing data with each other, organizations can mutually reduce their own data collection costs and improve the utility they derive from data [21]. In fact, there are already several platforms to facilitate data sharing among businesses [1, 40], scientific organizations [2, 3], and public institutions [16, 34].

However, simply pooling everyone's data and sharing with each other can lead to free-riding [23, 35]. For instance, if an agent (e.g an organization) sees that other agents are already contributing a large amount of data, then, the cost she incurs to collect her own dataset may not offset the marginal improvement in *her own* learned model due to diminishing returns of increasing dataset sizes (we describe this rigorously in §2). Hence, while she benefits from others' data, she has no incentive to collect and contribute data to the pool. A seemingly simple fix to this free-riding problem is to only return the datasets of the others if an agent submits a large enough dataset herself. However, this can be easily manipulated by a strategic agent who submits a large fabricated (fake) dataset without incurring any cost, receives the others' data, and then discards her fabricated dataset when learning. While the agent has benefited by this bad behavior, other agents who may use this fabricated dataset

37th Conference on Neural Information Processing Systems (NeurIPS 2023).

are worse off. Moreover, a naive test by the mechanism to check if the agent has fabricated data can be sidestepped by agents who collect only a small dataset and fabricate a larger dataset using this small dataset (e.g by fitting a model to the small dataset and then sampling from this fitted model).

In this work, we study these challenges in data sharing in one of the most foundational statistical problems, normal mean estimation, where the goal is to estimate the mean $\mu$ of a normal distribution $\mathcal{N}(\mu, \sigma^2)$ with known variance $\sigma^2$. We wish to design *mechanisms* for data sharing that satisfy the three fundamental desiderata of mechanism design; *Nash incentive compatibility (NIC):* agents have incentive to collect a sufficiently large amount of data and share it truthfully provided that all other agents are doing so; *individual rationality (IR):* agents are better off participating in the mechanism than working on their own; and *efficiency:* the mechanism leads to outcomes with small estimation error and data collection costs for all agents.

**Contributions:** *(i)* In §2, we formalize collaborative normal mean estimation in the presence of strategic agents. *(ii)* In §3, we design an NIC and IR mechanism for this problem to prevent free-riding and data fabrication and show that its social penalty, i.e sum of all agents' estimation errors and data collection costs, is at most twice that of the global minimum. *(iii)* In Appendix E, we study the same mechanism in high dimensional settings and relax the Gaussian assumption to distributions with bounded variance. We show that the mechanism retains its properties, with only a slight weakening of the NIC and efficiency guarantees. *(iv)* In §4, we consider two special cases where we impose natural restrictions on the agents' strategy space. We show that it is possible to design mechanisms which essentially achieve the global minimum social penalty in both settings. Next, we will summarize our primary mechanism and the associated theorem in §3.

## 1.1   Summary of main results

*Formalism:* We assume that all agents have a fixed cost for collecting one sample, and define an agent's penalty (negative utility) as the sum of her estimation error and the cost she incurred to collect data. To make the problem well-defined, for the estimation error, we find it necessary to consider the *maximum risk*, i.e maximum expected error over all $\mu \in \mathbb{R}$. A mechanism asks agents to collect data, and then shares the data among the agents in an appropriate manner to achieve the three desiderata. An agent's strategy space consists of three components: how much data she wishes to collect, what she chooses to submit after collecting the data, and how she estimates the mean $\mu$ using the dataset she collected, the dataset she submitted, and the information she received from the mechanism.

*Mechanism and theoretical result:* In our mechanism, which we call C3D (Cross-Check and Corrupt based on Difference), each agent $i$ collects a dataset $X_i$ and submits a possibly fabricated or altered version $Y_i$ to the mechanism. The mechanism then determines agent $i$'s allocation in the following manner. It pools the data from the other agents and splits them into two subsets $Z_i, Z_i'$. Then, $Z_i$ is returned as is, while $Z_i'$ is corrupted by adding noise that is proportional to the difference between $Y_i$ and $Z_i$. If an agent collects less or fabricates, she risks looking different to the others, and will receive a dataset $Z_i'$ of poorer quality. We show that this mechanism has a Nash equilibrium where all agents collect a sufficiently large amount of data, submit it truthfully, and use a carefully weighted average of the three datasets $X_i, Z_i$, and $Z_i'$ as their estimate for $\mu$. The weighting uses some additional side information that the mechanism provides to each agent. Below, we state an informal version of the main theoretical result of this paper, which summarizes the properties of our mechanism.

**Theorem 1 (informal):** *The above mechanism is Nash incentive compatible, individually rational, and achieves a social penalty that is at most twice the globally minimum social penalty.*

Corruption is the first of two ingredients to achieving NIC. The second is the design of the weighted average estimator which is (minimax) optimal after corruption. To illustrate why this is important, say that the mechanism had assumed that the agents will use any other sub-optimal estimator (e.g a simple average). Then it will need to lower the amount of corruption to ensure IR and efficiency. However, a strategic agent will realize that she can achieve a lower maximum risk with a better estimator (instead of collecting more data herself and/or receiving less corrupt data from the mechanism). She can leverage this insight to collect less data and lower her overall penalty.

*Proof techniques:* The most challenging part of our analysis is to show NIC, First, to show minimax optimality of our estimator, we construct a sequence of normal priors for $\mu$ and show that the minimum Bayes' risk converges to the maximum risk of the weighted average estimator. However, when compared to typical minimax proofs, we face more significant challenges. The first of these

is that the combined dataset $X_i \cup Z_i \cup Z_i'$ is neither independent nor identically distributed as the corruption is data-dependent. The second is that the agent's submission $Y_i$ also determines the degree of corruption, so we cannot look at the estimator in isolation when computing the minimum Bayes' risk; we should also consider the space of functions an agent may use to determine $Y_i$ from $X_i$. The third is that the expressions for the minimum Bayes' risk do not have closed form solutions and require non-trivial algebraic manipulations. To complete the NIC proof, we show that due to the carefully chosen amount of corruption, the agent should collect a sufficient amount of data to avoid excessive corruption, but not too much so as to increase her data collection costs.

## 1.2 Related Work

Mechanism design is one of the core areas of research in game theory [13, 18, 36]. Our work here is more related to mechanism design without payments, which has seen applications in fair division [31], matching markets [32], and kidney exchange [33] to name a few. There is a long history of work in the intersection of machine learning and mechanism design, although the overwhelming majority apply learning techniques when there is incomplete information about the mechanism or agent preferences, (e.g [6, 8, 22, 28, 30]). On the flip side, some work have designed data marketplaces, where customers may purchase data from contributors [4, 5, 19, 38]. These differ from our focus where we wish to incentivize agents to collaborate without payments.

Due to the popularity of shared data platforms [1, 2, 16, 34] and federated learning [21], there has been a recent interest in designing mechanisms for data sharing. Sim et al. [35] and Xu et al. [39] study fairness in collaborative data sharing, where the goal is to reward agents according to the amount of data they contribute. However, their mechanisms do not apply when strategic agents may try to manipulate a mechanism. Blum et al. [9] and Karimireddy et al. [23] study collaboration in federated learning. However, the strategy space of an agent is restricted to how much data they collect and their mechanism rewards each agent according to the quantity of the data she submitted. The above four works recognize that free-riding can be detrimental to data sharing, but assume that agents will not fabricate data. As discussed above, if this assumption is not true, agents can easily manipulate such mechanisms. Fraboni et al. [17] and Lin et al. [25] study federated learning settings where free-riders may send in fabricated gradients without incurring the computational cost of computing the gradients. However, their focus is on designing gradient descent algorithms that are robust to such attacks and not on incentivizing agents to perform the gradient computations. Some work have designed mechanisms for federated learning so as to elicit private information (such as data collection costs), but their focus is not on preventing free-riding or fabrication [15, 26]. Miller et al. [29] uses scoring systems to develop mechanisms that prevent signal fabrication. However, the agents in their settings can only choose to report either their true signal or something else but can not freely choose how much data to collect. Cai et al. [11] study mechanism design where a learner incentivizes agents to collect data via payments. Their mechanism, which also cross-checks the data submitted by the agents, has connections to our setting in §4.2 where we consider a restricted strategy space for the agents.

Our approach of using corruption to engender good behaviour draws inspiration from the robust estimation literature, which design estimators that are robust to data from malicious agents [12, 14, 27]. However, to the best of our knowledge, the specific form of corruption and the subsequent design of the minimax optimal estimator are new in this work, and require novel analysis techniques.

## 2 Problem Setup

We will now formally define our problem. We have $m$ agents, who are each able to collect i.i.d samples from a normal distribution $\mathcal{N}(\mu, \sigma^2)$, where $\sigma^2$ is known. They wish to estimate the mean $\mu$ of this distribution. To collect one sample, the agent has to incur a cost $c$. We will assume that $\sigma^2$, $c$, and $m$ are public information. However, $\mu \in \mathbb{R}$ is unknown, and no agent has auxiliary information, such as a prior, about $\mu$. An agent wishes to minimize her estimation error, while simultaneously keeping the cost of data collection low. While an agent may collect data on her own to manage this trade-off, by sharing data with other agents, she can reduce costs while simultaneously improving her estimate. We wish to design mechanisms to facilitate such sharing of data.

**Mechanism:** A mechanism receives a dataset from each agent, and in turn returns an *allocation* $A_i$ to each agent. An agent will use her allocation to estimate $\mu$. This allocation could be, for instance, a larger dataset obtained with other agents' datasets. The mechanism designer is free to choose a space of allocations $\mathcal{A}$ to achieve the desired goals. Formally, we define a mechanism as a tuple $M = (\mathcal{A}, b)$ where $\mathcal{A}$ denotes the space of allocations, and $b$ is a procedure to map the datasets collected from the $m$ agents to $m$ allocations. Denoting the universal set by $\mathcal{U}$, we write the space of mechanisms $\mathcal{M}$ as

$$\mathcal{M} = \big\{ M = (\mathcal{A}, b) : \ \mathcal{A} \subset \mathcal{U}, \ \ b : (\textstyle\bigcup_{n \geq 0} \mathbb{R}^n)^m \to \mathcal{A}^m \big\}. \tag{1}$$

As is customary, we will assume that the mechanism designer will publish the space of allocations $\mathcal{A}$ and the mapping $b$ (the procedure used to obtain the allocations) ahead of time, so that agents can determine their strategies. However, specific values computed/realized during the execution of the mechanism are not revealed, unless the mechanism chooses to do so via the allocation $A_i$.

**Agents' strategy space:** Once the mechanism is published, the agent will choose a strategy. In our setting, this will be the tuple $(n_i, f_i, h_i)$, which determines how much data she wishes to collect, what she chooses to submit, and how she wishes to estimate the mean $\mu$. First, the agent samples $n_i$ points to collect her initial dataset $X_i = \{x_{i,j}\}_{j=1}^{n_i}$, where $x_{i,j} \sim \mathcal{N}(\mu, \sigma^2)$, incurring $c n_i$ cost. She then submits $Y_i = \{y_{i,j}\}_j = f_i(X_i)$ to the mechanism. Here $f_i$ is a function which maps the collected dataset to a possibly fabricated or falsified dataset of a potentially different size. In particular, this fabrication can depend on the data she has collected. For instance, the agent could collect only a small dataset, fit a Gaussian, and then sample from it.

Finally, the mechanism returns the agent's allocation $A_i$, and the agent computes an estimate $h_i(X_i, Y_i, A_i)$ for $\mu$ using her initial dataset $X_i$, the dataset she submitted $Y_i$, and the allocation she received $A_i$. We include $Y_i$ as part of the estimate since an agent's submission may affect the allocation she receives. Consequently, agents could try to elicit additional information about $\mu$ via a carefully chosen $Y_i$. We can write the strategy space of an agent as $\mathcal{S} = \mathbb{N} \times \mathcal{F} \times \mathcal{H}$, where $\mathcal{F}$ is the space of functions mapping the dataset collected to the dataset submitted, and $\mathcal{H}$ is the space of all estimators using all the information she has. We have:

$$\mathcal{F} = \big\{ f : \textstyle\bigcup_{n \geq 0} \mathbb{R}^n \to \bigcup_{n \geq 0} \mathbb{R}^n \big\}, \qquad \mathcal{H} = \big\{ h : \textstyle\bigcup_{n \geq 0} \mathbb{R}^n \ \times \ \bigcup_{n \geq 0} \mathbb{R}^n \ \times \ \mathcal{A} \ \to \mathbb{R} \big\}. \tag{2}$$

One element of interest in $\mathcal{F}$ is the identity $\mathbf{I}$ which maps a dataset to itself. A mechanism designer would like an agent to use $f_i = \mathbf{I}$, i.e to submit the data that she collected as is, so that other agents can benefit from her data.

Going forward, when $s = \{s_i\}_i \in \mathcal{S}^m$ denotes the strategies of all agents, we will use $s_{-i} = \{s_j\}_{j \neq i}$ to denote the strategies of all agents except $i$. Without loss of generality, we will assume that agent strategies are deterministic. If they are stochastic, our results will carry through for every realization of any external source of randomness that the agent uses to determine $(n_i, f_i, h_i)$.

**Agent penalty:** The agent's *penalty* $p_i$ (i.e negative utility) is the sum of her squared estimation error and the cost $c n_i$ incurred to collect her dataset $X_i$ of $n_i$ points. The agent's penalty depends on the mechanism $M$ and the strategies $s = \{s_j\}_j$ of all the agents. Making this explicit, $p_i$ is defined as:

$$p_i(M, s) = \sup_{\mu \in \mathbb{R}} \mathbb{E} \left[ (h_i(X_i, Y_i, A_i) - \mu)^2 \ \Big| \ \mu \right] \ + \ c n_i \tag{3}$$

The term inside the expectation is the squared difference between the agent's estimate and the true mean (conditioned on the true mean $\mu$). The expectation is with respect to the randomness of all agents' data and possibly any randomness in the mechanism. We consider the *maximum risk*, i.e supremum over $\mu \in \mathbb{R}$, since the true mean $\mu$ is unknown to the agent a priori, and their strategy should yield good estimates, and hence small penalty, over all possible values $\mu$. To illustrate this further, note that when the value of true mean $\mu$ is $\mu'$, the optimal strategy for an agent will always be to not collect any data and choose the estimator $h_i(\cdot, \cdot, \cdot) = \mu'$ leading to $0$ penalty. However, this strategy can be meaningfully realized by an agent only if she knew that $\mu = \mu'$ a priori which renders the problem meaningless[1]. Considering the maximum risk accounts for the fact that $\mu$ is unknown and makes the problem well-defined.

---

[1]This is akin to the reason why it is customary to study the maximum risk in frequentist statistics [24, 37]. An alternative approach is to take a Bayesian view, considering a prior on $\mu$ and using the Bayes' risk $\mathbb{E}_\mu[\mathbb{E}[(h_i(X_i, Y_i, A_i) - \mu)^2 | \mu]]$ instead of the maximum risk in $p_i$. While we have adopted a frequentist formalism here, our main proof ideas can be ported over to the Bayesian setting as well (See Appendix F for more details)

**Recommended strategies:** In addition to publishing the mechanism, the mechanism designer will recommend strategies $s^\star = \{s_i^\star\}_i \in \mathcal{S}^m$ for the agents so as to incentivize collaboration and induce optimal social outcomes.

**Desiderata:** We can now define the three desiderata for a mechanism:

1. *Nash Incentive compatibility (NIC):* A mechanism $M = (\mathcal{A}, b)$ is said to be NIC at the recommended strategy profile $s^\star$ if, for each agent $i$, and for every other alternative strategy $s_i \in \mathcal{S}$ for that agent, we have $p_i(M, s^\star) \leq p_i(M, (s_i, s^\star_{-i}))$. That is, $s^\star$ is a Nash equilibrium so no agent has incentive to deviate if all other agents are following $s^\star$.

2. *Individual rationality (IR):* We say that a mechanism $M$ is IR at $s^\star$ if no agent suffers from a higher penalty by participating in the mechanism than the lowest possible penalty she could achieve on her own when all other agents are following $s^\star$. If an agent does not participate, she does not submit nor receive any data from the mechanism; she will simply choose how much data to collect and design the best possible estimator. Formally, we say that a mechanism $M$ is IR if the following is true for each agent $i$:

$$p_i(M, s^\star) \leq \inf_{n_i \in \mathbb{N}, \, h_i \in \mathcal{H}} \left\{ \sup_{\mu \in \mathbb{R}} \mathbb{E}\left[ (h_i(X_i, \varnothing, \varnothing) - \mu)^2 \,|\, \mu \right] \, + \, cn_i \right\}. \qquad (4)$$

3. *Efficiency:* The *social penalty* $P(M, s)$ of a mechanism $M$ when agents follow strategies $s$, is the sum of agent penalties (defined below). We define $\mathrm{PR}(M, s^\star)$ to be the ratio between the social penalty of a mechanism at the recommended strategies $s^\star$, and the lowest possible social penalty among all possible mechanisms and strategies (*without* NIC or IR constraints). We have:

$$P(M, s) = \sum_{i \in [m]} p_i(M, s), \qquad \mathrm{PR}(M, s^\star) = \frac{P(M, s^\star)}{\inf_{M' \in \mathcal{M}, \, s \in \mathcal{S}^m} P(M', s)} \qquad (5)$$

Note that $\mathrm{PR} \geq 1$. We say that a mechanism is efficient if $\mathrm{PR}(M, s^\star) = 1$ and that it is approximately efficient if $\mathrm{PR}(M, s^\star)$ is bounded by some constant that does not depend on $m$. If $s^\star$ is a Nash equilibrium, then $\mathrm{PR}(M, s^\star)$ can be viewed as an upper bound on the price of stability [7].

For what follows, we will discuss optimal strategies for agents working on her own and present a simple mechanism which minimizes the social penalty, but has a poor Nash equilibrium.

**Optimal strategies for an agent working on her own:** Recall that, given $n$ samples $\{x_i\}_{i=1}^n$ from $\mathcal{N}(\mu, \sigma^2)$, the sample mean is a minimax optimal estimator [24]; i.e among all possible estimators $h$, the sample mean minimizes the maximum risk $\sup_{\mu \in \mathbb{R}} \mathbb{E}[(\mu - h(\{x_i\}_{i=1}^n, \varnothing, \varnothing))^2 \,|\, \mu]$ (note that the agent only has the dataset she collected). Moreover, its mean squared error is $\sigma^2/n$ for all $\mu \in \mathbb{R}$. Hence, an agent acting on her own will choose the sample mean and collect $n_i = \sigma/\sqrt{c}$ samples so as to minimize their penalty; as long as the amount of data is less than $\sigma/\sqrt{c}$, an agent has incentive to collect more data since the cost of collecting one more point is offset by the marginal decrease in estimation error. This can be seen via the following simple calculation:

$$\inf_{\substack{n_i \in \mathbb{R} \\ h_i \in \mathcal{H}}} \left( \sup_{\mu} \mathbb{E}\left[ (h_i(X_i, \varnothing, \varnothing) - \mu)^2 \,\Big|\, \mu \right] \, + \, cn_i \right) = \min_{n_i \in \mathbb{R}} \left( \frac{\sigma^2}{n_i} + cn_i \right) = 2\sigma\sqrt{c} \ \stackrel{\triangle}{=} \ p_{\min}^{\mathrm{IR}}. \qquad (6)$$

Let $p_{\min}^{\mathrm{IR}} = 2\sigma\sqrt{c}$ denote the lowest achievable penalty by an agent working on her own. If all $m$ agents work independently, then the total social penalty is $mp_{\min}^{\mathrm{IR}} = 2\sigma m\sqrt{c}$. Next, we will look at a simple mechanism and an associated set of strategies which achieve the global minimum penalty. This will show that it is possible for all agents to achieve a significantly lower penalty via collaboration.

**A globally optimal mechanism *without* strategic considerations:** The following simple mechanism $M_{\mathrm{pool}}$, pools all the data from the other agents and gives it back to an agent. Precisely, it chooses the space of allocation $\mathcal{A} = \bigcup_{n \geq 0} \mathbb{R}^n$ to be datasets of arbitrary length, and sets agent $i$'s allocation to be $A_i = \bigcup_{j \neq i} Y_i$. The recommended strategies $s^{\mathrm{pool}} = \{(n_i^{\mathrm{pool}}, f_i^{\mathrm{pool}}, h_i^{\mathrm{pool}})\}_i$ asks each agent to collect $n_i^{\mathrm{pool}} = \sigma/\sqrt{cm}$ points[2], submit it as is $f_i^{\mathrm{pool}} = \mathbf{I}$, and use the sample mean of all points

---

[2]To avoid rounding effects, henceforth we will treat $\sigma/\sqrt{cm}$, and $\sigma/\sqrt{c}$ as integers.

---

**Algorithm 1** $M_{\texttt{C3D}}$

---

1: **Mechanism designer publishes:**
2:     The allocation space $\mathcal{A} = \bigcup_{n \geq 0} \mathbb{R}^n \times \bigcup_{n \geq 0} \mathbb{R}^n \times \mathbb{R}_+$, and the procedure in lines 6–15.
3: **Each agent $i$:**
4:     Choose strategy $s_i = (n_i, f_i, h_i)$.                                     # See (8) for recommended strategies.
5:     Sample $n_i$ points $X_i = \{x_{i,j}\}_{j=1}^{n_i}$ and submit $Y_i = f_i(X_i)$ to the mechanism.
6: **Mechanism:**
7:     **For** each agent $i \in [m]$:                              # can be done simultaneously for all agents
8:         $Y_{-i} \leftarrow \bigcup_{j \neq i} Y_j$.
9:         **If** $m \leq 4$:                    # Simply pool and return all of the other agents' data to agent $i$.
10:             $A_i \leftarrow (Y_{-i}, \varnothing, 0)$. Return $A_i$ to agent $i$.
11:         **Else:**
12:             $Z_i \leftarrow$ sample $\min\{|Y_{-i}|, \sigma/\sqrt{cm}\}$ points in $Y_{-i}$ without replacement.
13:             $\eta_i^2 \leftarrow \alpha^2 \left( \frac{1}{|Y_i|} \sum_{y \in Y_i} y - \frac{1}{|Z_i|} \sum_{z \in Z_i} z \right)^2$                        # See (7) for $\alpha$.
14:             $Z_i' \leftarrow \{z + \epsilon_{z,i}, \text{ for all } z \in Y_{-i} \backslash Z_i \text{ where } \epsilon_{z,i} \sim \mathcal{N}(0, \eta_i^2)\}$
15:             $A_i \leftarrow (Z_i, Z_i', \eta_i^2)$. Return $A_i$ to agent $i$.
16: **Each agent $i$:**
17:     Compute estimate $h_i(X_i, Y_i, A_i)$.                       # See (8) for recommended estimator.

---

as her estimate $h_i^{\texttt{pool}}(X_i, X_i, A_i) = \frac{1}{|X_i \cup A_i|} \sum_{z \in X_i \cup A_i} z$. It is straightforward to show that this minimizes the social penalty if all agents follow $s^{\texttt{pool}}$. After each agent has collected their datasets $\{X_i\}_i$, the social penalty is minimized if all agents have access to each other's datasets and they all use a minimax optimal estimator: this justifies using $M_{\texttt{pool}}$ with $f_i^{\texttt{pool}} = \mathbf{I}$ and setting $h_i^{\texttt{pool}}$ to be the sample mean. The following simple calculation justifies the choice of $\sum_i n_i^{\texttt{pool}}$:

$$\inf_{s \in \mathcal{S}^m} \sum_{i=1}^m \left( \sup_\mu \mathbb{E}\left[ (h_i(X_i, f_i, A_i) - \mu)^2 \,\Big|\, \mu \right] + c n_i \right) = \min_{\{n_i\}_i} \left( \frac{m\sigma^2}{\sum_i n_i} + c \sum_i n_i \right) = 2\sigma\sqrt{mc}.$$

However, $s^{\texttt{pool}}$ is not a Nash equilibrium of this mechanism, as an agent will find it beneficial to free-ride. If all other agents are submitting $\sigma/\sqrt{cm}$ points, by collecting no points, an agent's penalty is $\sigma\sqrt{mc}/(m-1)$, as she does not incur any data collection cost. This is strictly smaller than $2\sigma\sqrt{c/m}$ when $m \geq 3$. In fact, it is not hard to show that $M_{\texttt{pool}}$ is at a Nash equilibrium only when the total amount of data is $\sigma/\sqrt{c}$; for additional points, the marginal reduction in the estimation error for an individual agent does not offset her data collection costs. The social penalty at these equilibria is $\sigma\sqrt{c}(m+1)$ which is significantly larger than the global minimum when there are many agents.

A seemingly simple way to fix this mechanism is to only return the datasets of the other agents if an agent submits at least $\sigma/\sqrt{cm}$ points. However, as we will see in §4.1, such a mechanism can also be manipulated by an agent who submits a fabricated dataset of $\sigma/\sqrt{cm}$ points without actually collecting any data and incurring any cost and then discarding the fabricated dataset when estimating. Any naive test to check for the quality of the data can also be sidestepped by agents who sample only a few points, and use that to fabricate a larger dataset (e.g by sampling a large number of points from a Gaussian fitted to the small sample). Next, we will present our mechanism for this problem which satisfies all three desiderata.

## 3    Method and Results

We have outlined our mechanism $M_{\texttt{C3D}}$, and its interaction with the agents in Algorithm 1 in the natural order of events. We will first describe it procedurally, and then motivate our design choices. Our mechanism uses the following allocation space, $\mathcal{A} = \bigcup_{n \geq 0} \mathbb{R}^n \times \bigcup_{n \geq 0} \mathbb{R}^n \times \mathbb{R}_+$. An allocation $A_i = (Z_i, Z_i', \eta_i^2) \in \mathcal{A}$ consists of an uncorrupted dataset $Z_i$, a corrupted dataset $Z_i'$, and the variance $\eta_i^2$ of the noise added to $Z_i'$ for corruption. Once the mechanism and the allocation space are published, agent $i$ chooses her strategy $s = (n_i, f_i, h_i)$. She collects a dataset $X_i = \{x_{i,j}\}_{j=1}^{n_i}$, where $x_{i,j} \sim \mathcal{N}(\mu, \sigma^2)$, and submits $Y_i = f_i(X_i)$ to the mechanism.

Our mechanism determines agent $i$'s allocation as follows. Let $Y_{-i}$ be the union of all datasets submitted by the other agents. If there are at most four agents, we simply return all of the other

agents' data without corruption by setting $A_i \leftarrow (Y_{-i}, \varnothing, 0)$. If there are more agents, the mechanism first chooses a random subset of size $\min\{|Y_{-i}|, \sigma/\sqrt{cm}\}$ from $Y_{-i}$; denote this $Z_i$. In line 13, the mechanism individually adds Gaussian noise to the remaining points $Y_{-i} \setminus Z_i$ to obtain $Z_i'$ (line 14). The variance $\eta_i^2$ of the noise depends on the difference between the sample means of the subset $Z_i$ and the agent's submission $Y_i$. It is modulated by a value $\alpha$, which is a function of $c$, $m$, and $\sigma^2$. Precisely, $\alpha$ is the smallest number larger than $\sqrt{\sigma}(cm)^{-1/4}$ which satisfies $G(\alpha) = 0$, where:

$$G(\alpha) := \left( \frac{m-4}{m-2} \frac{4\alpha^2}{\sigma/\sqrt{cm}} - 1 \right) \frac{4\alpha}{\sqrt{\sigma}(m/c)^{1/4}} - \left( 4(m+1)\frac{\alpha^2}{\sigma\sqrt{m/c}} - 1 \right) \sqrt{2\pi} \exp\left( \frac{\sigma\sqrt{m/c}}{8\alpha^2} \right) \text{Erfc}\left( \frac{\sqrt{\sigma}(m/c)^{1/4}}{2\sqrt{2}\alpha} \right) \quad (7)$$

Finally, the mechanism returns the allocation $A_i = (Z_i, Z_i', \eta_i^2)$ to agent $i$ and the agent estimates $\mu$.

*Recommended strategies:* The recommended strategy $s_i^\star = (n_i^\star, f_i^\star, h_i^\star)$ for agent $i$ is given in (8). The agent should collect $n_i^\star = \sigma/(m\sqrt{c})$ samples if there are at most four agents, and $n_i^\star = \sigma/\sqrt{cm}$ samples otherwise. She should submit it without fabrication or alteration $f_i = \mathbf{I}$, and then use a weighted average of the datasets $(X_i, Z_i, Z_i')$ to estimate $\mu$. The weighting is proportional to the inverse variance of the data. For $X_i$ and $Z_i$ this is simply $\sigma^2$, but for $Z_i'$, the variance is $\sigma^2 + \eta_i^2$ since the mechanism adds Gaussian noise with variance $\eta_i^2$. We have:

$$n_i^\star = \begin{cases} \frac{\sigma}{m\sqrt{c}} & \text{if } m \leq 4 \\ \frac{\sigma}{\sqrt{cm}} & \text{if } m > 4 \end{cases}, \qquad f_i^\star = \mathbf{I},$$

$$h_i^\star(X_i, Y_i, (Z_i, Z_i', \eta_i^2)) = \frac{\frac{1}{\sigma^2} \sum_{u \in X_i \cup Z_i} u + \frac{1}{\sigma^2 + \eta_i^2} \sum_{u \in Z_i'} u}{\frac{1}{\sigma^2}|X_i \cup Z_i'| + \frac{1}{\sigma^2 + \eta_i^2}|Z_i'|} \qquad (8)$$

*Design choices:* Next, we will describe our design choices and highlight some key challenges. When $m \leq 4$, it is straightforward to show that the mechanism satisfies all our desired properties (see beginning of §3.1), so we will focus on the case $m > 4$. First, recall that the mechanism needs to incentivize agents to collect a sufficient amount of samples. However, simply counting the number of samples can be easily manipulated by an agent who simply submits a fabricated dataset of a large number of points. Instead, Algorithm 1 attempts to infer the quality of the data submitted by the agents using how well an agent's submission $Y_i$ approximates $\mu$. Ideally, we would set the variance $\eta_i^2$ of this corruption to be proportional to the difference $(\frac{1}{|Y_i|} \sum_{y \in Y_i} y - \mu)^2$, so that the more data she submits, the less the variance of $Z_i'$, which in turn yields a more accurate estimate for $\mu$. However, since $\mu$ is unknown, we use a subset $Z_i$ obtained from other agents' data as a proxy for $\mu$, and set $\eta_i^2$ proportional to $\left( \frac{1}{|Y_i|} \sum_{y \in Y_i} y - \frac{1}{|Z_i|} \sum_{z \in Z_i} z \right)^2$. If all agents are following $s^\star$, then $|Y_i| = |Z_i| = \sigma/\sqrt{cm} = n_i^\star$; it is sufficient to use only $n_i^\star$ points for validating $Y_i$ since both $\frac{1}{|Y_i|} \sum_{y \in Y_i} y$ and $\frac{1}{|Z_i|} \sum_{z \in Z_i} z$ will have the same order of error in approximating $\mu$.

The second main challenge is the design of the recommended estimator $h_i^\star$. In §3.1 we show how splitting $Y_{-i}$ into a clean and corrupted parts $Z_i$, $Z_i'$ allows us to design a minimax optimal estimator. A minimax optimal estimator is crucial to achieving NIC. To explain this, say that the mechanism assumes that agents will use a sub-optimal estimator, e.g sample mean of $X_i \cup Z_i \cup Z_i'$. Then, to account for the larger estimation error, it will need to choose a lower level of corruption $\eta_i^2$ to minimize the social penalty. However, a smart agent will realize that she can achieve a lower maximum risk by using a better estimator, such as the weighted average, instead of collecting more data in order to reduce the amount of corruption used by the mechanism. She can leverage this insight to collect less data and reduce her overall penalty.

This concludes the description of our mechanism. The following theorem, which is the main theoretical result of this paper, states that $M_{\text{C3D}}$ achieves the three desiderata outlined in §2.

**Theorem 1.** *Let $m > 1$, $\alpha$ be as defined in (7), and $s_i^\star$ be as defined in (8). Then, the following statements are true about the mechanism $M_{\text{C3D}}$ in Algorithm 1. (i) The strategy profile $s^\star$ is a Nash equilibrium. (ii) The mechanism is individually rational at $s^\star$. (iii) The mechanism is approximately efficient, with $\text{PR}(M_{\text{C3D}}, s^\star) \leq 2$.*

The mechanism is NIC as, provided that others are following $s_i^\star$, there is no reason for any one agent to deviate. Moreover, we achieve low social penalty at $s_i^\star$. Other than $s^\star$, there is also a set of similar Nash equilibria with the same social penalty: the agents can each add a same constant to the data points they collect and subtract the same value from the final estimate. Before we proceed,

the expression for $\alpha$ in (7) warrants explanation. If we treat $\alpha$ is a variable, we find that different choices of $\alpha$ can lead to other Nash equilibria with corresponding bounds on PR. This specific choice of $\alpha$ leads to a Nash equilibrium where agents collect $\sigma/\sqrt{cm}$ points, and a small bound on PR. Throughout this manuscript, we will treat $\alpha$ as the specific value obtained by solving (7), and *not* as a variable.

**High dimensional non-Gaussian distributions:**    In Appendix E, we study $M_{\text{C3D}}$ when applied to $d$–dimensional distributions. In Theorem 7, we show that under bounded variance assumptions, $s^\star$ is an $\varepsilon_m$-approximate Nash equilibrium and that $\text{PR}(M_{\text{C3D}}, s^\star) \leq 2 + \varepsilon_m$ where $\varepsilon_m \in \mathcal{O}(1/m)$.

### 3.1   Proof sketch of Theorem 1

*When $m \leq 4$:* First, consider the (easy) case $m \leq 4$. At $s_i^\star$, the total amount of data collected is $\sigma/\sqrt{c}$ (see $n_i^\star$ in (8)), and as there is no corrupted dataset, $h_i^\star$ simply reduces to the sample mean of $X_i \cup Y_{-i}$. The mechanism is IR since an agent's penalty will be $\sigma\sqrt{c}(1 + 1/m)$ which is smaller than $p_{\min}^{\text{IR}}$ (6). It is approximately efficient since the social penalty is $\sigma\sqrt{c}(m + 1)$ which is at most twice the global minimum $2\sigma\sqrt{mc}$ when $m \leq 4$. Finally, NIC is guaranteed by the same argument used in (6); as long as the total amount of data is less than $\sigma/\sqrt{c}$, the cost of collecting one more point is offset by the marginal decrease in the estimation error; hence, the agent is incentivized to collect more data. Moreover, as $A_i$ does not depend on $f_i$ under these conditions, there is no incentive to fabricate or falsify data.

*When $m > 4$:* We will divide this proof into four parts. We first show that $G(\alpha) = 0$ in line (6) has a solution $\alpha$ larger than $\sqrt{n_i^\star} = \sqrt{\sigma}(cm)^{-1/4}$. This will also be useful when analyzing the efficiency.

**1. Equation (7) has a solution.** We derive an asymptotic expansion of $\text{Erfc}(\cdot)$ using integration by parts to analyze the solution to (7). When $m \geq 5$, we show that $G\big(\sqrt{n_i^\star}\big) \times G\big(\sqrt{n_i^\star}(1 + 8/\sqrt{m})\big) < 0$. By continuity of $G$, there exists $\alpha_m \in \big(\sqrt{n_i^\star}, \sqrt{n_i^\star}(1 + 8/\sqrt{m})\big)$ s.t. $G(\alpha_m) = 0$. For $m$ large enough such that the residual in the asymptotic expansion is negligible, we show $\alpha_m \in \big(\sqrt{n_i^\star}, \sqrt{n_i^\star}(1 + \log m/m)\big)$ via an identical technique.

**2. The strategies $s^\star$ in (8) is a Nash equilibrium:** We show this via the following two steps. First (**2.1**), We show that fixing any $n_i$, the maximum risk and thus the penalty $p_i$ is minimized when agent $i$ submits the raw data and uses the weighted average as specified in (8), i.e for all $n_i$,

$$p_i(M_{\text{C3D}}, ((n_i, f_i^\star, h_i^\star), s_{-i}^\star)) \leq p_i(M_{\text{C3D}}, ((n_i, f_i, h_i), s_{-i}^\star)), \quad \forall (n_i, f_i, h_i) \in \mathbb{N} \times \mathcal{F} \times \mathcal{H}. \quad (9)$$

Second (**2.2**), we show that $p_i$ is minimized when agent $i$ collects $n_i^\star$ samples under $(f_i^\star, h_i^\star)$, i.e.

$$p_i(M_{\text{C3D}}, ((n_i^\star, f_i^\star, h_i^\star), s_{-i}^\star)) \leq p_i(M_{\text{C3D}}, ((n_i, f_i^\star, h_i^\star), s_{-i}^\star)), \quad \forall n_i \in \mathbb{N}. \quad (10)$$

***2.1: Proof of (9).*** As the data collection cost does not change for fixed $n_i$, it is sufficient to show that $(f_i^\star, h_i^\star)$ minimizes the maximum risk. Our proof is inspired by the following well-known recipe for proving minimax optimality of an estimator [24]: design a sequence of priors $\{\Lambda_\ell\}_\ell$, compute the minimum Bayes' risk $\{R_\ell\}_\ell$ for any estimator, and then show that $R_\ell$ converges to the maximum risk of the proposed estimator as $\ell \to \infty$.

To apply this recipe, we use a sequence of normal priors $\Lambda_\ell = \mathcal{N}(0, \ell^2)$ for $\mu$. Howeiver, before we proceed, we need to handle two issues. The first of these concerns the posterior for $\mu$ when conditioned on $(X_i, Z_i, Z_i')$. Since the corruption terms $\epsilon_{z,i}$ added to $Z_i'$ depend on $X_i$ and $Z_i$, this dataset is not independent. Moreover, as the variance $\eta_i^2$ is the difference between two normal random variables, $Z_i'$ is not normal. Despite these, we are able to show that the posterior $\mu|(X_i, Z_i, Z_i')$ is normal. The second challenge is that the submission $f_i$ also affects the estimation error as it determines the amount of noise $\eta_i^2$. We handle this by viewing $\mathcal{F} \times \mathcal{H}$ as a rich class of estimators and derive the optimal Bayes' estimator $(f_{i,\ell}^{\text{B}}, h_{i,\ell}^{\text{B}}) \in \mathcal{F} \times \mathcal{H}$ under the prior $\Lambda_\ell$. We then show that the minimum Bayes' risk converges to the maximum risk when using $(f_i^\star, h_i^\star)$.

Next, under the prior $\Lambda_\ell = \mathcal{N}(0, \ell^2)$, we can minimize the Bayes' risk with respect to $h_i \in \mathcal{H}$ by setting $h^{\mathrm{B}}_{i,\ell}$ to be the posterior mean. Then, the minimum Bayes' risk $R_\ell$ can be written as,

$$R_\ell = \inf_{f_i \in \mathcal{F}} \mathbb{E}\left[\left(|Z'_i|\left(\sigma^2 + \alpha^2\left(\frac{1}{|Y_i|}\sum_{y \in Y_i} y - \frac{1}{|Z_i|}\sum_{z \in Z_i} z\right)^2\right)^{-1} + \frac{|X_i| + |Z_i|}{\sigma^2} + \frac{1}{\ell^2}\right)^{-1}\right]$$

Note that $Y_i = f_i(X_i)$ depends on $f_i$. Via the Hardy-Littlewood inequality [10], we can show that the above quantity is minimized when $f^{\mathrm{B}}_{i,\ell}$ is chosen to be a shrunk version of the agent's initial dataset $X_i$, i.e $f^{\mathrm{B}}_{i,\ell}(X_i) = \left\{\left(1 + \sigma^2/(|X|\ell^2)\right)^{-1} x, \ \forall x \in X_i\right\}$. This gives us an expression for the minimum Bayes' risk $R_\ell$ under prior $\Lambda_\ell$. To conclude the proof, we note that the minimum Bayes' risk under any prior is a lower bound on the maximum risk, and show that $R_\ell$ approaches the maximum risk of $(f^\star_i, h^\star_i)$ from below. Hence, $(f^\star_i, h^\star_i)$ is minimax optimal for any $n_i$. (Above, it is worth noting that $f^{\mathrm{B}}_{i,\ell} \to f^\star_i = \mathbf{I}$ as $\ell \to \infty$. In the Appendix, we also find that $h^{\mathrm{B}}_{i,\ell} \to h^\star_i$. )

***2.2: Proof of*** (10). We can now write $p_i(M_{\mathrm{C3D}}, ((n_i, f^\star_i, h^\star_i), s^\star_{-i})) = R_\infty + cn_i$, where $R_\infty$ is the maximum risk of $(f^\star_i, h^\star_i)$ (and equivalently, the limit of the minimum Bayes' risk):

$$R_\infty := \mathbb{E}_{x \sim \mathcal{N}(0,1)}\left[\left((m-2)n^\star_i\left(\sigma^2 + \alpha^2\left(\sigma^2/n_i + \sigma^2/n^\star_i\right)x^2\right)^{-1} + (n_i + n^\star_i)\sigma^{-2}\right)^{-1}\right]$$

The term inside the expectation is convex in $n_i$ for each fixed $x$. As expectation preserves convexity, we can conclude that $p_i$ is a convex function of $n_i$. The choice of $\alpha$ in (7) ensures that the derivative is 0 at $n^\star$ which implies that $n^\star$ is a minimum of this function.

**3.** $M_{\mathrm{C3D}}$ **is individually rational at** $s^\star$**:** This is a direct consequence of step 2 as we can show that an agent 'working on her own' is a valid strategy in $M_{\mathrm{C3D}}$.

**4.** $M_{\mathrm{C3D}}$ **is approximately efficient at** $s^\star$**:** By observing that the global minimum penalty is $2\sigma\sqrt{cm}$, we use a series of nontrivial algebraic manipulations to show $\mathrm{PR}(M_{\mathrm{C3D}}, s^\star) = \frac{1}{2}\left(\frac{10\alpha^2/n^\star_i - 1}{4(m+1)\alpha^2/(mn^\star_i) - 1} + 1\right)$. As $\alpha > \sqrt{n^\star_i}$, some simple algebra leads to $\mathrm{PR}(M_{\mathrm{C3D}}, s^\star) < 2$.

# 4 Special Cases: Restricting the Agents' Strategy Space

In this section, we study two special cases motivated by some natural use cases, where we restrict the agents' strategy space. In addition to providing better guarantees on the efficiency, this will also help us better illustrate the challenges in our original setting.

## 4.1 Agents cannot fabricate or falsify data

First, we study a setting where agents are not allowed to fabricate data or falsify data. Specifically, in (2), $\mathcal{F}$ is restricted to functions which map a dataset to any subset. This is applicable when there are regulations preventing such behavior (e.g government institutions, hospitals)

*Mechanism:* The discussion at the end of §2 motivates the following modification to the pooling mechanism. We set the allocation space to be $\mathcal{A} = \bigcup_{n \geq 0} \mathbb{R}^n$, i.e the space of all datasets. If an agent $i$ submits at least $\sigma/\sqrt{cm}$ points, then give her all the other agents' datasets, i.e $A_i = \cup_{j \neq i} Y_j$; otherwise, set $A_i = \varnothing$. The recommended strategy $s^\star_i = (n^\star_i, f^\star_i, h^\star_i)$ of each agent is to collect $\sigma/\sqrt{cm}$ points, submit it as is $f^\star_i = \mathbf{I}$, and then use the sample mean of $Z_i \cup A_i$ to estimate $\mu$. The theorem below, whose proof is straightforward, states the main properties of this mechanism.

**Theorem 2.** *The following statements about the mechanism and strategy profile $s^\star$ in the paragraph above are true when $\mathcal{F}$ is restricted to functions which map a dataset to any subset: (i) $s^\star$ is a Nash equilibrium. (ii) The mechanism is individually rational at $s^\star$. (iii) At $s^\star$, the mechanism is efficient.*

It is not hard to see that this mechanism can be easily manipulated by the agent if there are no restrictions on $\mathcal{F}$. As the mechanism only checks for the amount of data submitted, the agent can submit a fabricated dataset of $\sigma/\sqrt{cm}$ points, and then discard this dataset when computing the estimate, which results in detrimental free-riding.

## 4.2 Agents accept an estimated value from the mechanism

Our next setting is motivated by use cases where the mechanism may directly deploy the estimated value for $\mu$ in some downstream application for the agent, i.e the agents are forced to use this value. This is motivated by federated learning, where agents collect and send data to a server (mechanism), which deploys a model (estimate) directly on the agent's device [9, 23]. This requires modifying the agent's strategy space to $\mathcal{S} = \mathbb{N} \times \mathcal{F}$. Now, an agent can only choose $(n_i, f_i)$, how much data she wishes to collect, and how to fabricate or falsify the dataset. A mechanism is defined as a procedure $b : \left( \bigcup_{n \geq 0} \mathbb{R}^n \right)^m \to \mathbb{R}^m$, which maps $m$ datasets to $m$ estimated mean values.

Algorithm 3 (see Appendix D) outlines a family of mechanisms parametrized by $\epsilon > 0$ for this setting. As we will see shortly, with parameter $\epsilon$, the mechanism can achieve a PR of $(1 + \epsilon)$. This mechanism computes agent $i$'s estimate for $\mu$ as follows. First, let $Y_{-i}$ be the union of all datasets submitted by the other agents. Similar to Algorithm 1, the algorithm individually adds Gaussian to each $Y_{-i}$ to obtain $Z_i$ (line 10). Unlike before, this noise is added to the entire dataset and the variance $\eta_i^2$ of this noise depends on the difference between the sample means of the agent's submission $Y_i$ and all of the other agents' submissions $Y_{-i}$. It also depends on two $\epsilon$-dependent parameters defined in line 6. Finally, the mechanism deploys the sample mean of $Y_i \cup Z_i$ as the estimate for $\mu$. The recommended strategies $s_i^\star = (n_i^\star, f_i^\star)$ for the agents is to simply collect $n_i^\star = \sigma / \sqrt{cm}$ points and submit it as is $f_i^\star = \mathbf{I}$. The following theorem states the main properties of the mechanism.

**Theorem 3.** *Let $\epsilon > 0$. The following statements about Algorithm 3 and the strategy profile $s^\star$ given in the paragraph above are true: (i) $s^\star$ is a Nash equilibrium. (ii) The mechanism is individually rational at $s^\star$. (iii) At $s^\star$, the mechanism is approximately efficient with $\mathrm{PR}(M, s^\star) \leq 1 + \epsilon$.*

The above theorem states that it is possible to obtain a social penalty that is arbitrarily close to the global minimum under the given restriction of the strategy space. However, this mechanism is not NIC if agents are allowed to design their own estimator. For instance, if the mechanism returns $A_i = Z_i$ (line 10), then using a weighted average of the data in $X_i$ and $Z_i$ yields a lower estimation error than simple average used by the mechanism (see Appendix D). An agent can leverage this insight to collect and submit less data and obtain a lower overall penalty at the expense of other agents. Cai et al. [11] study a setting where agents are incentivized to collect data and submit it truthfully via payments. Interestingly, their corruption method can be viewed as a special case of Algorithm 3 with $k_\epsilon = 1$ and only achieves a $1.5\times$ factor of the global minimum social penalty. Moreover, when applied to the more general strategy space, it shares the same shortcomings as the mechanism in Theorem 3.

## 5 Conclusion

We studied collaborative normal mean estimation in the presence of strategic agents. Naive mechanisms which only look at the quantity of the dataset submitted, can be manipulated by agents who under-collect and/or fabricate data, leaving all agents worse off. To address this issue, when sharing the others' data with an agent, our mechanism $M_{\text{C3D}}$ corrupts this dataset proportional to how much the data reported by the agent differs from the other agents. We design minimax optimal estimators for this corrupted dataset to achieve a socially desirable Nash equilibrium.

**Future directions:** We believe that designing mechanisms for other collaborative learning settings may require relaxing the *exact* NIC guarantees to make the analysis tractable. For many learning problems, it is difficult to design exactly optimal estimators, and it is common to settle for rate-optimal (i.e up to constants) estimators [24]. For instance, even simply relaxing to high dimensional distributions with bounded variance, $M_{\text{C3D}}$ can only provide an approximate NIC guarantee.

## Acknowledgment

This project is supported in part by NSF grants 1545481, 1704117, 1836978, 2023239, 2041428, 2202457, ARO MURI W911NF2110317, and AF CoE FA9550-18-1-0166.

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

# A  Proof of Theorem 1

In this section, we prove Theorem 1. This section is organized as follows. First, in §A.1, we consider the case where $m \leq 4$. In the remainder of this section, we will assume $m \geq 5$. First, in §A.2, we will show that (7) can be solved for $\alpha$ and state some properties about the solution. Then, in §A.3, we will prove the Nash incentive compatibility result, in §A.4 we will prove individual rationality, and in §A.5, we will prove the result on efficiency.

## A.1  When $m \leq 4$

First, consider the (easy) case $m \leq 4$. At $s_i^\star$, the total amount of data collected is $\sigma/\sqrt{c}$ as each agent will be collecting $n_i^\star = \frac{\sigma}{m\sqrt{c}}$ (see (8)). As there is no corrupted dataset, $h_i^\star$ simply reduces to the sample mean of $X_i \cup Y_{-i}$. The individual rationality property follows from the following simple calculation:

$$p_i(M_{\text{C3D}}, s^\star) = \left(1 + \frac{1}{m}\right)\sqrt{c}\sigma < 2\sqrt{c}\sigma = p_{\min}^{\text{IR}}.$$

Similarly, the bound on the ratio between the penalties can also be obtained via the following calculation:

$$\text{PR} = \frac{m\left(1 + \frac{1}{m}\right)\sqrt{c}\sigma}{2\sigma\sqrt{cm}} < \sqrt{m} \leq 2.$$

Finally, to show NIC, consider agent $i$ and assume that all other agents have followed the recommended strategies, i.e collected $\sigma/(m\sqrt{c})$. Then, the agent will have an *uncorrupted* dataset $Y_{-i} = \bigcup_{j \neq i} X_j$ of $n_{-i}^\star = (m-1)\sigma/(m\sqrt{c})$ points with no corruption. Regardless of what she chooses to submit, the best estimator she could use with the union of this dataset $Y_{-i}$ and the data she collects $X_i$ and will be the sample mean as it is minimax optimal. The number of points that minimizes her penalty is,

$$\operatorname*{argmin}_{n_i} \left(\sup_{\mu} \mathbb{E}\left[(h_i(X_i, Y_i, Y_{-i}) - \mu)^2 \,\Big|\, \mu\right] + cn_i\right) = \operatorname*{argmin}_{n_i \in \mathbb{R}} \left(\frac{\sigma^2}{n_i + n_{-i}^\star} + cn_i\right) = \frac{\sigma}{m\sqrt{c}}$$

Finally, as $A_i$ does not depend on $f_i$ under these conditions, there is no incentive to fabricate or falsify data, i.e choosing anything other than $f^\star = \mathbf{I}$ does not lower her utility.

In the remainder of this section, will study the harder case, $m \geq 4$.

## A.2  Existence of a solution to (7) and some of its properties

In this section, we show that $G\left(\frac{\sigma^{1/2}}{(cm)^{1/4}}\right) < 0$ and $G\left(\left(1 + \frac{C_m}{m}\right)\frac{\sigma^{1/2}}{(cm)^{1/4}}\right) > 0$, where $C_m = 20$ when $m \leq 20$ and $C_m = 5$ when $m > 20$. This means equation $G(\alpha) = 0$ has solution in $\left(\frac{\sigma^{1/2}}{(cm)^{1/4}}, \left(1 + \frac{C_m}{m}\right)\frac{\sigma^{1/2}}{(cm)^{1/4}}\right)$.

First, in Lemma 12, we derive an asymptotic expansion of the Gaussian complementary error function, and construct lower and upper bounds for $G(\alpha)$ that are easier to work with. We have restated these lower ($\text{Erfc}_{\text{LB}}$) and upper ($\text{Erfc}_{\text{UB}}$) bounds below.

$$\text{Erfc}_{\text{UB}}(x) := \frac{1}{\sqrt{\pi}}\left(\frac{\exp(-x^2)}{x} - \frac{\exp(-x^2)}{2x^3} + \frac{3\exp(-x^2)}{4x^5}\right) \tag{11}$$

$$\text{Erfc}_{\text{LB}}(x) := \frac{1}{\sqrt{\pi}}\left(\frac{\exp(-x^2)}{x} - \frac{\exp(-x^2)}{2x^3}\right) \tag{12}$$

We can now use this to derive the following lower ($G_{\text{LB}}$) and upper ($G_{\text{UB}}$) bounds on $G$. Here, we have used the fact that $4(m+1)\frac{\alpha^2}{\sigma\sqrt{m/c}} - 1 > 0$ when $\alpha \geq (\sigma/\sqrt{cm})^{1/2}$. We have:

$$G_{\text{LB}}(\alpha) := \left(\frac{m-4}{m-2}\frac{4\alpha^2}{\sigma/\sqrt{cm}} - 1\right)\frac{4\alpha}{\sqrt{\sigma}(m/c)^{1/4}}$$
$$- \left(4(m+1)\frac{\alpha^2}{\sigma\sqrt{m/c}} - 1\right)\sqrt{2\pi}\exp\left(\frac{\sigma\sqrt{m/c}}{8\alpha^2}\right)\text{Erfc}_{\text{UB}}\left(\frac{\sqrt{\sigma}(m/c)^{1/4}}{2\sqrt{2}\alpha}\right),$$

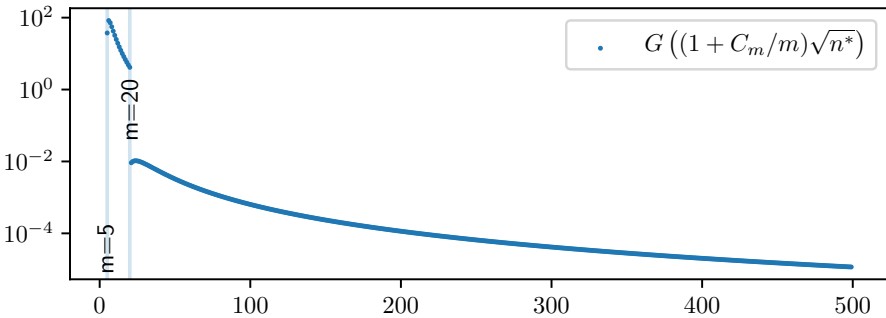

Figure 1: Plot for $G\left(\left(1 + \frac{C_m}{m}\right)\frac{\sigma^{1/2}}{(cm)^{1/4}}\right)$. See `G_em_plot.py`. The discontinuity at $m = 20$ is due to the different values for $C_m$ when $m \le 20$ and when $m > 20$.

$$G_{\text{UB}}(\alpha) := \left(\frac{m-4}{m-2}\frac{4\alpha^2}{\sigma/\sqrt{cm}} - 1\right)\frac{4\alpha}{\sqrt{\sigma}(m/c)^{1/4}}$$
$$- \left(4(m+1)\frac{\alpha^2}{\sigma\sqrt{m/c}} - 1\right)\sqrt{2\pi}\exp\left(\frac{\sigma\sqrt{m/c}}{8\alpha^2}\right)\text{Erfc}_{\text{LB}}\left(\frac{\sqrt{\sigma}(m/c)^{1/4}}{2\sqrt{2}\alpha}\right).$$

By first, substituting $\sigma/\sqrt{cm}$ for $\alpha$ in the expressions for $G_{\text{UB}}$ and $\text{Erfc}_{\text{UB}}$, and then via a sequence of algebraic manipulations, we can verify that

$$G\left(\frac{\sigma^{1/2}}{(cm)^{1/4}}\right) \le G_{\text{UB}}\left(\frac{\sigma^{1/2}}{(cm)^{1/4}}\right)$$

$$= \frac{4\left(\frac{4(m-4)}{m-2} - 1\right)\left(\frac{\sigma}{\sqrt{cm}}\right)^{1/2}}{\sqrt{\sigma}\left(\frac{m}{c}\right)^{1/4}} - \sqrt{2}\left(\frac{4(m+1)}{\sqrt{\frac{m}{c}}\sqrt{cm}} - 1\right)\left(\frac{2\sqrt{2}\left(\frac{\sigma}{\sqrt{cm}}\right)^{1/2}}{\sqrt{\sigma}\left(\frac{m}{c}\right)^{1/4}} - \frac{8\sqrt{2}\left(\frac{\sigma}{\sqrt{cm}}\right)^{3/2}}{\sigma^{3/2}\left(\frac{m}{c}\right)^{3/4}}\right)$$

$$= -\frac{128}{(m-2)m^{5/2}} < 0.$$

Next, we will show that $G\left(\left(1 + \frac{C_m}{m}\right)\frac{\sigma^{1/2}}{(cm)^{1/4}}\right) > 0$ by studying the lower bound $G_{\text{LB}}$. For $m \in [5, 500]$, we can verify individually that $G\left(\left(1 + \frac{C_m}{m}\right)\frac{\sigma^{1/2}}{(cm)^{1/4}}\right) > 0$ (See Figure 1). For $m > 500$, we have:

$$G\left(\left(1 + \frac{C_m}{m}\right)\frac{\sigma^{1/2}}{(cm)^{1/4}}\right) = G\left(\left(1 + \frac{5}{m}\right)\frac{\sigma^{1/2}}{(cm)^{1/4}}\right) \ge G_{\text{LB}}\left(\left(1 + \frac{5}{m}\right)\frac{\sigma^{1/2}}{(cm)^{1/4}}\right)$$

$$= \frac{4\left(\frac{4\left(\frac{5}{m}+1\right)^2(m-4)}{m-2} - 1\right)\left(\frac{5}{m} + 1\right)\left(\frac{\sigma}{\sqrt{cm}}\right)^{1/2}}{\sqrt{\sigma}\left(\frac{m}{c}\right)^{1/4}}$$

$$- \sqrt{2}\left(\frac{4\left(\frac{5}{m}+1\right)^2(m+1)}{\sqrt{\frac{m}{c}}\sqrt{cm}} - 1\right)\left(\frac{2\sqrt{2}\left(\frac{5}{m}+1\right)\left(\frac{\sigma}{\sqrt{cm}}\right)^{1/2}}{\sqrt{\sigma}\left(\frac{m}{c}\right)^{1/4}} - \frac{8\sqrt{2}\left(\frac{5}{m}+1\right)^3\left(\frac{\sigma}{\sqrt{cm}}\right)^{3/2}}{\sigma^{3/2}\left(\frac{m}{c}\right)^{3/4}}\right)$$

$$+ \frac{96\sqrt{2}\left(\frac{5}{m}+1\right)^5\left(\frac{\sigma}{\sqrt{cm}}\right)^{5/2}}{\sigma^{5/2}\left(\frac{m}{c}\right)^{5/4}}\Bigg)$$

$$= \frac{64(m+5)^3\left(m^6 - 191m^5 - 1566m^4 - 3920m^3 + 2100m^2 + 19500m + 15000\right)}{(m-2)m^{21/2}}.$$

When $m > 500$,

$$m^6 - 191m^5 - 1566m^4 - 3920m^3 = m^3(m^3 - 191m^2 - 1566m - 3920)$$

$$>m^3((200+200+100)m^2 - 191m^2 - 1566m - 3920)$$
$$>m^3(200m^2 + 10^5m + 2.5 \times 10^7 - 191m^2 - 1566m - 3920) > 0.$$

Combining the results from the two previous displays, we have, $G\left(\left(1 + \frac{C_m}{m}\right)\frac{\sigma^{1/2}}{(cm)^{1/4}}\right) > 0$ which completes the proof for this section.

## A.3 Algorithm 1 is Nash incentive compatible

In this section, we will prove the following lemma which states that $s_i^\star$, as defined in (8) is a Nash equilibrium in $M_{\text{C3D}}$.

**Lemma 4** (NIC). *The recommended strategies $s^\star = \{(n_i^\star, f_i^\star, h_i^\star)\}_i$ as defined in (8) in mechanism $M_{\text{C3D}}$ (Algorithm 1) satisfies:*
$$p_i(M_{\text{C3D}}, s^\star) \leq p_i(M_{\text{C3D}}, (s_i, s_{-i}^\star))$$
*for all $i \in [m]$ and $s_i \in \mathbb{N} \times \mathcal{F} \times \mathcal{H}$.*

The Proof of Lemma 4 relies on the following two lemmas:

**Lemma 5** (Optimal Estimation and Submission). *For all $i \in [m]$ and $(n_i, f_i, h_i) \in \mathbb{N} \times \mathcal{F} \times \mathcal{H}$.*
$$p_i(M_{\text{C3D}}, ((n_i, f_i^\star, h_i^\star), s_{-i}^\star)) \leq p_i(M_{\text{C3D}}, ((n_i, f_i, h_i), s_{-i}^\star)).$$

See the Proof of Lemma 5 in §A.3.1

**Lemma 6** (Optimal Sample Size). *For all $i \in [m]$ and $n_i \in \mathbb{N}$.*
$$p_i(M_{\text{C3D}}, ((n_i^\star, f_i^\star, h_i^\star), s_{-i}^\star)) \leq p_i(M_{\text{C3D}}, ((n_i, f_i^\star, h_i^\star), s_{-i}^\star)).$$

See the Proof of Lemma 6 in §A.3.2

*Proof of Lemma 4.* By Lemma 5 and 6, we have, for all $i \in [m]$ and $s_i' = (n_i, f_i, h_i) \in \mathbb{N} \times \mathcal{F} \times \mathcal{H}$,
$$p_i(M_{\text{C3D}}, s^\star) = p_i(M_{\text{C3D}}, ((n_i^\star, f_i^\star, h_i^\star), s_{-i}^\star)) \leq p_i(M_{\text{C3D}}, ((n_i, f_i^\star, h_i^\star), s_{-i}^\star))$$
$$\leq p_i(M_{\text{C3D}}, ((n_i, f_i, h_i), s_{-i}^\star)) = p_i(M_{\text{C3D}}, (s_i', s_{-i}^\star))$$

$\square$

### A.3.1 Proof of Lemma 5

In this section, we will prove Lemma 5, which, intuitively states that, regardless of the amount of data collected, agent $i$ should submit the data as is ($f_i^\star = \mathbf{I}$) and use the weighted average estimator in (8) to estimate $\mu$. We will do so via the following three step procedure, inspired by well–known techniques for proving minimax optimality of estimators (e.g see Theorem 1.12, Chapter 5 of Lehmann and Casella [24]).

1. First, we construct a sequence of prior distributions $\{\Lambda_\ell\}_{\ell \geq 1}$ for $\mu$ and calculate the sequence of Bayesian risks under the prior distributions:
$$R_\ell := \inf_{f_i \in \mathcal{A}, h_i \in \mathcal{H}} \mathbb{E}_{\mu \sim \Lambda_\ell}\left[\mathbb{E}\left[(h_i(X_i, f_i(X_i), A_i) - \mu)^2 \,\middle|\, \mu\right]\right], \quad \ell \geq 1.$$

2. Then, we will show that $\lim_{\ell \to \infty} R_\ell = \sup_\mu \mathbb{E}\left[(h_i^\star(X_i, f_i^\star(X_i), A_i) - \mu)^2 \,\middle|\, \mu\right]$.

3. Finally, as the Bayesian risk is a lower bound on maximum risk, we will conclude that $(f_i^\star, h_i^\star)$ is minimax optimal.

Without loss of generality, we focus only on the deterministic $f_i$ and $h_i$. If either of them are stochastic, we can condition on the external source of randomness and treat them as deterministic functions. Our proof holds for any realization of this external source of randomness, and hence it will hold in expectation as well. Similarly, $Z_i$ is randomly chosen in Algorithm 1. In the following, we condition on this randomness and the entire proof will carry through.

Note that $Y_i = f_i(X_i)$. We will use both of them interchangeably in the subsequent proof.

**Step 1 (Bounding the Bayes' risk under the sequence of priors):** We will use a sequence of normal priors $\Lambda_\ell := \mathcal{N}(0, \ell^2)$ for all $\ell \geq 1$. To bound the Bayes' risk under these priors, we will first note that for a fixed $f_i \in \mathcal{F}$,

$$x|\mu \sim \mathcal{N}(\mu, \sigma^2) \quad \forall x \in X_i \cup Z_i; \tag{13}$$

$$x|\mu, \eta_i^2 \sim \mathcal{N}(\mu, \sigma^2 + \eta_i^2) \quad \forall x \in Z_i'. \tag{14}$$

Here, recall that $\eta_i^2$ is a function of $Y_i$ and $Z_i$. Because both $Y_i = f_i(X_i)$ and $\eta_i^2$ are deterministic functions of $X_i, Z_i$ when $f_i$ is fixed, the posterior distribution for $\mu$ conditioned on $(X_i, Y_i, A_i)$ can be calculated as follows:

$$p(\mu|X_i, Y_i, A_i) = p\big(\mu|X_i, Y_i, Z_i, Z_i', \eta_i^2\big) = p(\mu|X_i, Z_i, Z_i')$$

$$\propto p(\mu, X_i, Z_i, Z_i') = p(Z_i'|X_i, Z_i, \mu)p(X_i, Z_i|\mu)p(\mu) = p(Z_i'|X_i, Z_i, \mu)p(X_i|\mu)p(Z_i|\mu)p(\mu)$$

$$\propto \exp\left(-\frac{1}{2(\sigma^2 + \eta_i^2)}\sum_{x \in Z_i'}(x-\mu)^2\right)\exp\left(-\frac{1}{2\sigma^2}\sum_{x \in X_i \cup Z_i}(x-\mu)^2\right)\exp\left(-\frac{\mu^2}{2\ell^2}\right)$$

$$\propto \exp\left(-\frac{1}{2}\left(\frac{|Z_i'|}{\sigma^2 + \eta_i^2} + \frac{|X_i| + |Z_i|}{\sigma^2} + \frac{1}{\ell^2}\right)\mu^2\right)\exp\left(\frac{1}{2}2\left(\frac{\sum_{x \in Z_i'} x}{\sigma^2 + \eta_i^2} + \frac{\sum_{x \in X_i \cup Z_i} x}{\sigma^2}\right)\mu\right)$$

$$= \exp\left(-\frac{1}{2}\left(\frac{1}{\sigma_\ell^2}\mu^2 - 2\frac{\mu_\ell}{\sigma_\ell^2}\mu\right)\right) \propto \exp\left(-\frac{1}{2\sigma_\ell^2}(\mu - \mu_\ell)^2\right),$$

where

$$\mu_\ell = \frac{\frac{\sum_{x \in Z_i'} x}{\sigma^2 + \eta_i^2} + \frac{\sum_{x \in X_i \cup Z_i} x}{\sigma^2}}{\frac{|Z_i'|}{\sigma^2 + \eta_i^2} + \frac{|X_i| + |Z_i|}{\sigma^2} + \frac{1}{\ell^2}}, \quad \text{and} \quad \sigma_\ell^2 = \frac{1}{\frac{|Z_i'|}{\sigma^2 + \eta_i^2} + \frac{|X_i| + |Z_i|}{\sigma^2} + \frac{1}{\ell^2}}. \tag{15}$$

We can therefore conclude that (despite the non i.i.d nature of the data), the posterior for $\mu$ is Gaussian with mean and variance as shown above. We have:

$$\mu|X_i, Y_i, A_i \sim \mathcal{N}(\mu_\ell, \sigma_\ell^2).$$

Next, following standard steps (See Corollary 1.2 in Chapter 4 of [24]), we know that $\mathbb{E}_\mu\Big[(h_i(X_i, Y_i, A_i) - \mu)^2|X_i, Y_i, A_i\Big]$ is minimized when $h_i(X_i, Y_i, A_i) = \mathbb{E}_\mu[\mu|X_i, Y_i, A_i] = \mu_\ell$. This shows that for any $f_i \in h_i$, the optimal $h_i$ is simply the posterior mean of $\mu$ under the prior $\Lambda_\ell$ conditioned on $(X_i, f_i(X_i), A_i)$. We can rewrite the minimum averaged risk over $\mathcal{H}$ by switching the order of expectation:

$$\inf_{h_i \in \mathcal{H}} \mathbb{E}_{\mu \sim \Lambda_\ell}\Big[\mathbb{E}\Big[(h_i(X_i, Y_i, A_i) - \mu)^2|\mu\Big]\Big]$$

$$= \inf_{h_i \in \mathcal{H}} \mathbb{E}_{X_i, Z_i, Z_i'}\Big[\mathbb{E}_\mu\Big[(h_i(X_i, Y_i, A_i) - \mu)^2|X_i, Z_i, Z_i'\Big]\Big]$$

$$= \mathbb{E}_{X_i, Z_i, Z_i'}\Big[\mathbb{E}_\mu\Big[(\mu_\ell - \mu)^2|X_i, Z_i, Z_i'\Big]\Big] = \mathbb{E}_{X_i, Z_i, Z_i'}\Big[\sigma_\ell^2\Big]$$

$$= \mathbb{E}_{X_i, Z_i}\left[\frac{1}{\frac{|Z_i'|}{\sigma^2 + \eta_i^2} + \frac{|X_i| + |Z_i|}{\sigma^2} + \frac{1}{\ell^2}}\right], \tag{16}$$

the expectation in the last step involves only $X_i, Z_i$ because $\sigma_\ell^2$ depends only on $X_i, Z_i$ and $|Z_i'|$, but not the instantiation of $Z_i'$.

Next, we will show that (16) is minimized for the following choice of $f_i$ which shrinks each points in $X_i$ by an amount that depends on the prior $\Lambda_\ell$'s variance $\ell^2$:

$$f_i(X_i) = \left\{\frac{|X_i|/\sigma^2}{|X_i|/\sigma^2 + 1/\ell^2}x \text{ , for each } x \in X_i\right\}. \tag{17}$$

**Remark 1.** *An interesting observation (albeit not critical to the proof) here is that $f_i$ in (17) converges pointwise to $f_i^\star$, i.e.* **I***, as $\ell \to \infty$. This shows that the optimal submission function under the prior converges to $f_i^\star$. We can make a similar observation about the posterior mean in (15), where $\mu_\ell$ converges to $h_i^\star$ as $\ell \to \infty$.*

To prove (17), we first define the following quantities.

$$\widehat{\mu}(X_i) := \frac{1}{|X_i|} \sum_{x \in X_i} x, \quad \widehat{\mu}(Y_i) := \frac{1}{|Y_i|} \sum_{x \in Y_i} x, \quad \widehat{\mu}(Z_i) := \frac{1}{|Z_i|} \sum_{s \in Z_i} x.$$

We will also find it useful to express $\eta_i^2$ as follows. Here $\alpha$ is as defined in (7). We have:

$$\eta_i^2 = \alpha^2 (\widehat{\mu}(Y_i) - \widehat{\mu}(Z_i))^2$$

The following calculations show that, conditioned on $X_i$, $\widehat{\mu}(Z_i) - \mu$ and $\mu - \frac{|X_i|/\sigma^2}{|X_i|/\sigma^2 + 1/\ell^2} \widehat{\mu}(X_i)$ are independent Gaussian random variables[3]:

$$
\begin{aligned}
p(\widehat{\mu}(Z_i) &- \mu, \mu | X_i) \propto p(\widehat{\mu}(Z_i) - \mu, \mu, X_i) \\
&= p(\widehat{\mu}(Z_i) - \mu, X_i | \mu) p(\mu) = p(\widehat{\mu}(Z_i) - \mu | \mu) p(X_i | \mu) p(\mu) \\
&\propto \exp\left(-\frac{1}{2} \frac{|Z_i|}{\sigma^2} (\widehat{\mu}(Z_i) - \mu)^2\right) \exp\left(-\frac{1}{2\sigma^2} \sum_{x \in X_i} (x - \mu)^2\right) \exp\left(-\frac{1}{2\ell^2} \mu^2\right) \\
&\propto \underbrace{\exp\left(-\frac{1}{2} \frac{|Z_i|}{\sigma^2} (\widehat{\mu}(Z_i) - \mu)^2\right)}_{\propto p(\widehat{\mu}(Z_i) - \mu | X_i)} \underbrace{\exp\left(-\frac{1}{2} \left(\frac{|X_i|}{\sigma^2} + \frac{1}{\ell^2}\right) \left(\mu - \frac{|X_i|/\sigma^2}{|X_i|/\sigma^2 + 1/\ell^2} \widehat{\mu}(X_i)\right)^2\right)}_{\propto p\left(\mu - \frac{|X_i|/\sigma^2}{|X_i|/\sigma^2 + 1/\ell^2} \widehat{\mu}(X_i) | X_i\right)}
\end{aligned}
$$

Thus conditioning on $X_i$, we can write

$$
\begin{pmatrix} \widehat{\mu}(Z_i) - \mu \\ \mu - \frac{|X_i|/\sigma^2}{|X_i|/\sigma^2 + 1/\ell^2} \widehat{\mu}(X_i) \end{pmatrix} \sim \mathcal{N}\left( \begin{pmatrix} 0 \\ 0 \end{pmatrix}, \begin{pmatrix} \frac{\sigma^2}{|Z_i|} & 0 \\ 0 & \frac{1}{|X_i|/\sigma^2 + 1/\ell^2} \end{pmatrix} \right).
$$

which leads us to

$$
\widehat{\mu}(Z_i) - \frac{|X_i|/\sigma^2}{|X_i|/\sigma^2 + 1/\ell^2} \widehat{\mu}(X_i) \bigg| X_i \sim \mathcal{N}\left( 0, \underbrace{\frac{\sigma^2}{|Z_i|} + \frac{1}{|X_i|/\sigma^2 + 1/\ell^2}}_{=:\tilde{\sigma}_\ell^2} \right) \tag{18}
$$

Next, we will rewrite the squared difference in $\eta_i^2$ as follows:

$$
\begin{aligned}
\frac{\eta_i^2}{\alpha^2} &= (\widehat{\mu}(Y_i) - \widehat{\mu}(Z_i))^2 \\
&= \left( \underbrace{\widehat{\mu}(Z_i) - \frac{|X_i|/\sigma^2}{|X_i|/\sigma^2 + 1/\ell^2} \widehat{\mu}(X_i)}_{=\tilde{\sigma}_\ell e} + \underbrace{\frac{|X_i|/\sigma^2}{|X_i|/\sigma^2 + 1/\ell^2} \widehat{\mu}(X_i) - \widehat{\mu}(Y_i)}_{=:\phi(X_i, f_i)} \right)^2.
\end{aligned}
$$

Here, we observe that the first part of the RHS above is equal to $\tilde{\sigma}_\ell$, where $e$ is a normal noise $e|X_i \sim \mathcal{N}(0, 1)$ and $\tilde{\sigma}_\ell$ is as defined in (18). For brevity, we will denote the second part of the RHS as $\phi(X_i, f_i)$, which intuitively characterizes the difference between $X_i$ and $Y_i$. Importantly, $\phi(X_i, f_i) = 0$ when $f_i$ is chosen to be (17).

Using $e$ and $\phi$, we can rewrite (16) using conditional expectation:

$$
\begin{aligned}
\mathbb{E}_{X_i, Z_i} \left[ \frac{1}{\frac{|Z_i'|}{\sigma^2 + \eta_i^2} + \frac{|X_i| + |Z_i|}{\sigma^2} + \frac{1}{\ell^2}} \right] &= \mathbb{E}_{X_i} \left[ \mathbb{E}_{Z_i | X_i} \left[ \frac{1}{\frac{|Z_i'|}{\sigma^2 + \eta_i^2} + \frac{|X_i| + |Z_i|}{\sigma^2} + \frac{1}{\ell^2}} \right] \right] \\
&= \mathbb{E}_{X_i} \left[ \mathbb{E}_{e | X_i} \left[ \frac{1}{\frac{|Z_i'|}{\sigma^2 + \alpha^2 (\tilde{\sigma}_\ell e + \phi(X_i, f_i))^2} + \frac{|X_i| + |Z_i|}{\sigma^2} + \frac{1}{\ell^2}} \right] \right]
\end{aligned}
$$

---

[3]This is akin to the observation that given $u, v \sim \mathcal{N}(0, 1)$, then $u - v$ and $u + v$ are independent.

$$= \mathbb{E}_{X_i}\left[\int_{-\infty}^{\infty} \underbrace{\frac{1}{\frac{|Z_i'|}{\sigma^2+\alpha^2\tilde{\sigma}_\ell^2(e+\phi(X_i,f_i)/\tilde{\sigma}_\ell)^2}+\frac{|X_i|+|Z_i|}{\sigma^2}+\frac{1}{\ell^2}}}_{=:F_1(e+\phi(X_i,f_i)/\tilde{\sigma}_\ell)} \underbrace{\frac{1}{\sqrt{2\pi}}\exp\left(-\frac{e^2}{2}\right)}_{=:F_2(e)} de\right], \qquad (19)$$

where we use the fact that $e|X_i \sim \mathcal{N}(0,1)$ in the last step. To proceed, we will consider the inner expectation in the RHS above. For any fixed $X_i$, $F_1(\cdot)$ (as marked on the RHS) is an even function that monotonically increases on $[0,\infty)$ bounded by $\frac{\sigma}{|X_i|+|Z_i|}$ and $F_2(\cdot)$ (as marked on the RHS) is an even function that monotonically decreases on $[0,\infty)$. That means, for any $a \in \mathbb{R}$,

$$\int_{-\infty}^{\infty} F_1(e-a)F_2(e)de \leq \int_{-\infty}^{\infty} \frac{\sigma}{|X_i|+|Z_i|}F_2(e)de = \frac{\sigma}{|X_i|+|Z_i|} < \infty.$$

By a corollary of the Hardy-Littlewood inequality in Lemma 9, we have

$$\int_{-\infty}^{\infty} F_1(e+\phi(X_i,f_i)/\tilde{\sigma}_\ell)F_2(e)de \geq \int_{-\infty}^{\infty} F_1(e)F_2(e)de, \qquad (20)$$

the equality is achieved when $\phi(X_i,f_i)/\tilde{\sigma}_\ell = 0$. In particular, the equality holds when $f_i$ is chosen as specified in (17).

Now, to complete Step 1, we combine (16), (19) and (20) to obtain

$$\inf_{h_i \in \mathcal{H}} \mathbb{E}_{\mu \sim \Lambda_\ell}\left[\mathbb{E}\left[(h_i(X_i,Y_i,A_i)-\mu)^2|\mu\right]\right] = \mathbb{E}_{X_i}\left[\int_{-\infty}^{\infty} F_1(e+\phi(X_i,f_i)/\tilde{\sigma}_\ell)F_2(e)de\right]$$

$$\geq \mathbb{E}_{X_i}\left[\int_{-\infty}^{\infty} F_1(e)F_2(e)de\right] = \int_{-\infty}^{\infty} F_1(e)F_2(e)de, \qquad (21)$$

where the last step is because conditioning on each realization of $X_i$, the term inside the expectation is a constant. Using (21), we can write the Bayes risk $R_\ell$ under any prior $\Lambda_\ell$ as:

$$R_\ell := \inf_{f_i \in \mathcal{A}, h_i \in \mathcal{H}} \mathbb{E}_{\mu \sim \Lambda_\ell}\left[\mathbb{E}\left[(h_i(X_i,Y_i,A_i)-\mu)^2\big|\mu\right]\right] = \int_{-\infty}^{\infty} F_1(e)F_2(e)de$$

$$= \mathbb{E}_{e \sim \mathcal{N}(0,1)}\left[\frac{1}{\frac{|Z_i'|}{\sigma^2+\alpha^2\tilde{\sigma}_\ell^2 e^2}+\frac{|X_i|+|Z_i|}{\sigma^2}+\frac{1}{\ell^2}}\right]$$

Because the term inside the expectation is bounded by $\frac{\sigma^2}{|X_i|+|Z_i|}$ and $\lim_{\ell \to \infty} \tilde{\sigma}_\ell^2 = \frac{\sigma^2}{|Z_i|}+\frac{\sigma^2}{|X_i|}$, we can use dominated convergence theorem to show that:

$$R_\infty := \lim_{\ell \to \infty} R_\ell = \mathbb{E}_{e \sim \mathcal{N}(0,1)}\left[\frac{1}{\frac{|Z_i'|}{\sigma^2+\alpha^2\left(\frac{\sigma^2}{|Z_i|}+\frac{\sigma^2}{|X_i|}\right)e^2}+\frac{|X_i|+|Z_i|}{\sigma^2}}\right] \qquad (22)$$

**Step 2: Maximum risk of $(f_i^\star, h_i^\star)$:** Next, we will compute the maximum risk of the $(f_i^\star, h_i^\star)$ (see (8)) and show that it is equal to the RHS of (22). First note that we can write,

$$\begin{pmatrix}\widehat{\mu}(X_i)-\mu \\ \widehat{\mu}(Z_i)-\mu\end{pmatrix} \sim \mathcal{N}\left(\begin{pmatrix}0 \\ 0\end{pmatrix}, \begin{pmatrix}\frac{\sigma^2}{|X_i|} & 0 \\ 0 & \frac{\sigma^2}{|Z_i|}\end{pmatrix}\right).$$

By a linear transformation of this Gaussian vector, we obtain

$$\begin{pmatrix}\frac{|X_i|}{\sigma^2}(\widehat{\mu}(X_i)-\mu)+\frac{|Z_i|}{\sigma^2}(\widehat{\mu}(Z_i)-\mu) \\ \widehat{\mu}(X_i)-\widehat{\mu}(Z_i)\end{pmatrix} = \begin{pmatrix}\frac{|X_i|}{\sigma^2} & \frac{|Z_i|}{\sigma^2} \\ 1 & -1\end{pmatrix}\begin{pmatrix}\widehat{\mu}(X_i)-\mu \\ \widehat{\mu}(Z_i)-\mu\end{pmatrix}$$

$$\sim \mathcal{N}\left(\begin{pmatrix}0 \\ 0\end{pmatrix}, \begin{pmatrix}\frac{|X_i|+|Z_i|}{\sigma^2} & 0 \\ 0 & \frac{\sigma^2}{|X_i|}+\frac{\sigma^2}{|Z_i|}\end{pmatrix}\right),$$

which means $\frac{|X_i|}{\sigma^2}(\widehat{\mu}(X_i)-\mu)+\frac{|Z_i|}{\sigma^2}(\widehat{\mu}(Z_i)-\mu)$ and $\frac{\eta_i}{\alpha}=\widehat{\mu}(X_i)-\widehat{\mu}(Z_i)$ are independent Gaussian random variables. Therefore, the the maximum risk of $(f_i^\star, h_i^\star)$ is:

$$\sup_\mu \mathbb{E}\big[(h_i^\star(X_i, Y_i, A_i)-\mu)^2|\mu\big] = \sup_\mu \mathbb{E}_{\eta_i}\left[\mathbb{E}\left[\left(\frac{\frac{\sum_{x\in Z_i'} x}{\sigma^2+\eta_i^2}+\frac{|X_i|}{\sigma^2}\widehat{\mu}(X_i)+\frac{|Z_i|}{\sigma^2}\widehat{\mu}(Z_i)}{\frac{|Z_i'|}{\sigma^2+\eta_i^2}+\frac{|X_i|+|Z_i|}{\sigma^2}}-\mu\right)^2\middle|\eta_i\right]\right]$$

$$=\sup_\mu \mathbb{E}_{\eta_i}\left[\mathbb{E}\left[\left(\frac{\frac{\sum_{x\in Z_i'}(x-\mu)}{\sigma^2+\eta_i^2}+\frac{|X_i|}{\sigma^2}(\widehat{\mu}(X_i)-\mu)+\frac{|Z_i|}{\sigma^2}(\widehat{\mu}(Z_i)-\mu)}{\frac{|Z_i'|}{\sigma^2+\eta_i^2}+\frac{|X_i|+|Z_i|}{\sigma^2}}\right)^2\middle|\eta_i\right]\right]$$

$$=\sup_\mu \mathbb{E}_{\eta_i}\left[\frac{\mathbb{E}\left[\left(\frac{\sum_{x\in Z_i'}(x-\mu)}{\sigma^2+\eta_i^2}+\frac{|X_i|}{\sigma^2}(\widehat{\mu}(X_i)-\mu)+\frac{|Z_i|}{\sigma^2}(\widehat{\mu}(Z_i)-\mu)\right)^2\middle|\eta_i\right]}{\left(\frac{|Z_i'|}{\sigma^2+\eta_i^2}+\frac{|X_i|+|Z_i|}{\sigma^2}\right)^2}\right]$$

$$=\sup_\mu \mathbb{E}_{\eta_i}\left[\frac{1}{\left(\frac{|Z_i'|}{\sigma^2+\eta_i^2}+\frac{|X_i|+|Z_i|}{\sigma^2}\right)^2}\left(\frac{|Z_i'|(\sigma^2+\eta_i^2)}{(\sigma^2+\eta_i^2)^2}+\frac{|X_i|+|Z_i|}{\sigma^2}\right)\right]$$

$$=\mathbb{E}_{\eta_i}\left[\frac{1}{\frac{|Z_i'|}{\sigma^2+\eta_i^2}+\frac{|X_i|+|Z_i|}{\sigma^2}}\right]=\mathbb{E}\left[\frac{1}{\frac{|Z_i'|}{\sigma^2+\alpha^2(\widehat{\mu}(Z_i)-\widehat{\mu}(X_i))^2}+\frac{|X_i|+|Z_i|}{\sigma^2}}\right]$$

Because $\widehat{\mu}(Z_i)-\widehat{\mu}(X_i)\sim\mathcal{N}\left(0,\frac{\sigma^2}{|X_i|}+\frac{\sigma^2}{|Z_i|}\right)$, we can further write the maximum risk as:

$$\sup_\mu \mathbb{E}\big[(h_i^\star(X_i, Y_i, A_i)-\mu)^2|\mu\big] = \mathbb{E}_{e\sim\mathcal{N}(0,1)}\left[\frac{1}{\frac{|Z_i'|}{\sigma^2+\alpha^2\left(\frac{\sigma^2}{|Z_i|}+\frac{\sigma^2}{|X_i|}\right)e^2}+\frac{|X_i|+|Z_i|}{\sigma^2}}\right]=R_\infty$$

Here, we have observed that the final expression in the above equation is exactly the same as the Bayes' risk in the limit in (22) from Step 1.

**Step 3: Minimax optimality of $(f_i^\star, h_i^\star)$:** As the maximum is larger than the average, we can write, for any prior $\Lambda_\ell$, and any $(f_i, h_i)\in\mathcal{F}\times\mathcal{H}$,

$$\sup_\mu \mathbb{E}\big[(h_i(X_i, f_i(X_i), A_i)-\mu)^2|\mu\big]\geq\mathbb{E}_{\Lambda_\ell}\big[\mathbb{E}\big[(h_i(X_i, f_i(X_i), A_i)-\mu)^2|\mu\big]\big]\geq R_\ell.$$

As this is true for all $\ell$, by taking the limit we have, for all $(f_i, h_i)\in\mathcal{F}\times\mathcal{H}$,

$$\sup_\mu \mathbb{E}\big[(h_i(X_i, f_i(X_i), A_i)-\mu)^2|\mu\big]\geq R_\infty=\sup_\mu \mathbb{E}\big[(h_i^\star(X_i, f_i^\star(X_i), A_i)-\mu)^2|\mu\big].$$

That is, the recommended $(f_i^\star, h_i^\star)$ has a smaller maximum risk than all other $(f_i, h_i)\in\mathcal{F}\times\mathcal{H}$. This establishes that for any $n_i$,

$$p_i(M_{\text{C3D}}, ((n_i, f_i^\star, h_i^\star), s_{-i}^\star))=\inf_{f_i\in\mathcal{A}}\inf_{h_i\in\mathcal{H}}p_i(M_{\text{C3D}}, ((n_i, f_i, h_i), s_{-i}^\star)).$$

### A.3.2 Proof of Lemma 6

In the previous section, we showed that for any $n_i$, the optimal $(f_i, h_i)$ were $(f_i^\star, h_i^\star)$ as given in (8). Now, we show that for the given $(f_i^\star, h_i^\star)$, the optimal number of samples is $n_i^\star=\sigma/\sqrt{cm}$. For this, we will show that $p_i$ is a convex function of $n_i$ and then show that its gradient is 0 at $n_i^\star$.

First, noting that

$$\widehat{\mu}(Z_i)-\widehat{\mu}(X_i)\sim\mathcal{N}\left(0, \frac{\sigma^2}{|X_i|}+\frac{\sigma^2}{|Z_i|}\right),$$

we can rewrite the penalty term as:

$$p(n_i) := p_i\big(M_{\text{C3D}}, ((n_i, f_i^\star, h_i^\star), s_{-i}^\star)\big) = \mathbb{E}\left[\frac{1}{\frac{|Z_i'|}{\sigma^2 + \alpha^2(\widehat{\mu}(Z_i) - \widehat{\mu}(X_i))^2} + \frac{|X_i| + |Z_i|}{\sigma^2}}\right] + cn_i$$

$$= \mathbb{E}_{x \sim \mathcal{N}(0,1)}\left[\frac{1}{\frac{|Z_i'|}{\sigma^2 + \alpha^2\left(\frac{\sigma^2}{|X_i|} + \frac{\sigma^2}{|Z_i|}\right)x^2} + \frac{|X_i| + |Z_i|}{\sigma^2}}\right] + cn_i$$

$$= \mathbb{E}_{x \sim \mathcal{N}(0,1)}\Bigg[\underbrace{\frac{1}{\frac{(m-2)n_i^\star}{\sigma^2 + \alpha^2\left(\frac{\sigma^2}{n_i} + \frac{\sigma^2}{n_i^\star}\right)x^2} + \frac{n_i + n_i^\star}{\sigma^2}}}_{=:l(n_i, x; \alpha)}\Bigg] + cn_i \tag{23}$$

*Convexity of penalty function:* To show that $p(n_i)$ is convex in $n_i$, let us consider $l(n_i, x; \alpha)$. Fixing $\alpha$ and $x$, we have

$$\frac{\partial}{\partial n_i} l(n_i, x; \alpha) = -\sigma^2 \frac{1 + \frac{(m-2)n_i^\star}{\left(1 + \alpha^2\left(\frac{1}{n_i} + \frac{1}{n_i^\star}\right)x^2\right)^2}\frac{\alpha^2 x^2}{n_i^2}}{\left(\frac{(m-2)n_i^\star}{1 + \alpha^2\left(\frac{1}{n_i} + \frac{1}{n_i^\star}\right)x^2} + n_i + n_i^\star\right)^2} = -\sigma^2 \frac{1 + \frac{(m-2)n_i^\star \alpha^2 x^2}{\left(n_i + \alpha^2\left(1 + \frac{n_i}{n_i^\star}\right)x^2\right)^2}}{\left(\frac{(m-2)n_i^\star}{1 + \alpha^2\left(\frac{1}{n_i} + \frac{1}{n_i^\star}\right)x^2} + n_i + n_i^\star\right)^2}$$

$$\tag{24}$$

As $\frac{\partial}{\partial n_i} l(n_i, x; \alpha)$ is an increasing function of $n_i$, we have that $l(n_i, x; \alpha)$ is a convex function in $n_i$. As expectation preserves convexity (see Lemma 10), $p(n_i)$ is a convex function.

*Penalty is minimized when $n_i = n_i^\star$.* Lemma 13 provides an expression for the derivative of $p(n_i)$ (obtained purely via algebraic manipulations). Using this, we have

$$p'(n_i^\star) = -\frac{\sigma^2}{64\frac{\alpha^2}{m-2}\frac{\alpha}{\sqrt{mn_i^\star}}mn_i^\star}\left(\frac{4\alpha}{\sqrt{mn_i^\star}}\left(\frac{4\alpha^2 m}{(m-2)n_i^\star} - 1\right)\right.$$

$$\left. - \exp\left(\frac{mn_i^\star}{8\alpha^2}\right)\left(\frac{4\alpha^2}{mn_i^\star}(m+1) - 1\right)\sqrt{2\pi}\,\text{Erfc}\left(\frac{1}{2\sqrt{2}\sqrt{\frac{\alpha^2}{mn_i^\star}}}\right)\right)$$

$$+ c \hspace{4cm} \text{(By Lemma 13)}$$

$$= -\frac{\sigma^2}{64\frac{\alpha^2}{m-2}\frac{\alpha}{\sqrt{mn_i^\star}}mn_i^\star}\left(\frac{4\alpha}{\sqrt{mn_i^\star}}\left(\frac{4\alpha^2(m-4)}{(m-2)n_i^\star} - 1\right)\right.$$

$$\left. - \exp\left(\frac{mn_i^\star}{8\alpha^2}\right)\left(\frac{4\alpha^2}{mn_i^\star}(m+1) - 1\right)\sqrt{2\pi}\,\text{Erfc}\left(\frac{1}{2\sqrt{2}\sqrt{\frac{\alpha^2}{mn_i^\star}}}\right)\right)$$

$$= G(\alpha) = 0.$$

Here, the second step uses the fact that $n_i^\star = \frac{\sigma}{\sqrt{cm}}$. Finally, we have observed that the expression is equal to $G(\alpha)$ as defined in (7) which is 0 by our choice of $\alpha$. Since $p'(n_i^\star) = 0$ and $p(\cdot)$ is convex, we can conclude that $p(n_i)$ is minimized when $n_i = n_i^\star$. Therefore,

$$p_i\big(M_{\text{C3D}}, ((n_i^\star, f_i^\star, h_i^\star), s_{-i}^\star)\big) \leq p_i\big(M_{\text{C3D}}, ((n_i, f_i^\star, h_i^\star), s_{-i}^\star)\big).$$

## A.4 Algorithm 1 is individually rational

As outlined in the main text, the NIC property implies IR since 'working on her own' is a valid strategy in the mechanism. Precisely, if an agent collects any number of points $n_i$, chooses not to submit anything $f_i(\cdot) = \varnothing$, and then uses the sample average of the points she collected $h_i(X_i, \varnothing, A_i) = |X_i|^{-1} \sum_{x \in X_i} x$, then $(n_i, f_i, h_i) \in \mathcal{S}$.

Below, we will prove this more formally and also show that the agent's penalty is strictly smaller when participating. For any fixed $n_i$, without participating in the mechanism, the smallest penalty the agent can achieve is by using empirical mean estimation and the penalty is:

$$\frac{\sigma^2}{n_i} + cn_i$$

When participating, the agent gets an additional $n_i^\star$ number of clean data along with some noisy data, provided that all other agents are following $s_{-i}^\star$. By using the empirical mean over the clean data, the penalty is:

$$\frac{\sigma^2}{n_i + n_i^\star} + cn_i < \frac{\sigma^2}{n_i} + cn_i$$

Now, since the weighted average estimator in $s_i^\star$ is minimax optimal, the agent gets even smaller maximum risk and hence smaller penalty. In other words, for any $n_i$,

$$p_i(M_{\text{C3D}}, s^\star) \leq p_i(M_{\text{C3D}}, ((n_i, f_i^\star, h_i^\star), s_{-i}^\star)) \leq \frac{\sigma^2}{n_i + n_i^\star} + cn_i < \frac{\sigma^2}{n_i} + cn_i$$

By minimizing the RHS with respect to $n_i$, we get $p_i(M_{\text{C3D}}, s^\star) < p_{\min}^{\text{IR}}$. Thus Algorithm 1 is IR.

## A.5 Algorithm 1 is approximately efficient

In this section, we will bound the penalty ratio PR for $M_{\text{C3D}}$ at the strategy profiles $s_i^\star$.

First, noting that $G(\alpha) = 0$ (see (7)), we can rearrange the terms in the equation to obtain:

$$\exp\left(\frac{mn_i^\star}{8\alpha^2}\right) \text{Erfc}\left(\frac{1}{2\sqrt{2}\sqrt{\frac{\alpha^2}{mn_i^\star}}}\right) = \frac{1}{\sqrt{2\pi}} \frac{\frac{4\alpha}{\sqrt{mn_i^\star}}\left(\frac{4\alpha^2(m-4)}{(m-2)n_i^\star} - 1\right)}{\frac{4\alpha^2}{mn_i^\star}(m+1) - 1} \tag{25}$$

Next, we will use the expression for $p(n_i) = p_i(M_{\text{C3D}}, (s_{-i}^\star, (n_i, f_i^\star, h_i^\star)))$ in Lemma 13 and the equation in (25) to simplify $p(n_i^\star)$ as follows:

$$p(n_i^\star) = \frac{\sqrt{\frac{\alpha^2}{mn_i^\star}}\sigma^2 \left(2m\sqrt{2\pi}\sqrt{\frac{\alpha^2}{mn_i^\star}} - \exp\left(\frac{mn_i^\star}{8\alpha^2}\right)(m-2)\pi\, \text{Erfc}\left(\frac{1}{2\sqrt{2}\sqrt{\frac{\alpha^2}{mn_i^\star}}}\right)\right)}{4\sqrt{2\pi}\alpha^2} + cn_i^\star$$

$$\text{(By Lemma 13)}$$

$$= \frac{\sqrt{\frac{\alpha^2}{mn_i^\star}}\sigma^2 \left(2m\sqrt{2\pi}\sqrt{\frac{\alpha^2}{mn_i^\star}} - (m-2)\pi \frac{1}{\sqrt{2\pi}} \frac{\frac{4\alpha}{\sqrt{mn_i^\star}}\left(\frac{4\alpha^2(m-4)}{(m-2)n_i^\star} - 1\right)}{\frac{4\alpha^2}{mn_i^\star}(m+1) - 1}\right)}{4\sqrt{2\pi}\alpha^2} + cn_i^\star \quad \text{(By (25))}$$

$$= \frac{\sigma^2 \left(m - (m-2)\frac{\frac{4\alpha^2(m-4)}{(m-2)n_i^\star} - 1}{\frac{4\alpha^2}{mn_i^\star}(m+1) - 1}\right)}{2mn_i^\star} + cn_i^\star$$

$$= \frac{\sigma^2}{2mn_i^\star} \frac{\frac{4\alpha^2}{n_i^\star}(m+1) - m - \frac{4\alpha^2}{n_i^\star}(m-4) + (m-2)}{\frac{4\alpha^2}{n_i^\star}\frac{m+1}{m} - 1} + cn_i^\star$$

$$= \frac{\sigma^2}{2mn_i^\star} \frac{\frac{20\alpha^2}{n_i^\star} - 2}{\frac{4\alpha^2}{n_i^\star}\frac{m+1}{m} - 1} + cn_i^\star = \frac{\sigma^2}{mn_i^\star} \frac{\frac{10\alpha^2}{n_i^\star} - 1}{\frac{4\alpha^2}{n_i^\star}\frac{m+1}{m} - 1} + cn_i^\star$$

**Algorithm 2** $M_{\text{PCS}}$

---

1: **Mechanism designer publishes:**
2:      The allocation space $\mathcal{A} = \bigcup_{n \geq 0} \mathbb{R}^n$, and the procedure in lines 6–11.
3: **Each agent $i$:**
4:      Choose strategy $s_i = (n_i, f_i, h_i)$.
5:      Sample $n_i$ points $X_i = \{x_{i,j}\}_{j=1}^{n_i}$ and submit $Y_i = f_i(X_i)$ to the mechanism.
6: **Mechanism:**
7:      For each agent $i \in [m]$:                                                     # can be done simultaneously for all agents
8:          $A_i \leftarrow \bigcup_{j \neq i} Y_j$   if $|Y_i| \geq \sigma/\sqrt{cm}$,   $A_i \leftarrow \varnothing$  otherwise.
9:          Return $A_i$ to each agent.
10: **Each agent $i$:**
11:      Compute estimate $h_i(X_i, Y_i, A_i)$.

---

$$= \sigma\sqrt{\frac{c}{m}}\left(\frac{\frac{10\alpha^2}{n_i^\star} - 1}{\frac{4\alpha^2}{n_i^\star}\frac{m+1}{m} - 1} + 1\right)$$

From our conclusion in §A.2, we have $\alpha^2 > \frac{\sigma}{\sqrt{cm}} = n_i^\star$, i.e. $\frac{\alpha^2}{n_i^\star} > 1$. Therefore, we have:

$$\text{PR}(M_{\text{C3D}}, s^\star) = \frac{mp(n_i^\star)}{2\sigma\sqrt{cm}} = \frac{1}{2}\left(\frac{\frac{10\alpha^2}{n_i^\star} - 1}{\frac{4\alpha^2}{n_i^\star}\frac{m+1}{m} - 1} + 1\right)$$

$$< \frac{1}{2}\left(\frac{\frac{10\alpha^2}{n_i^\star} - 1 + \frac{10\alpha^2}{n_i^\star}\frac{1}{m} + \left(\frac{2\alpha^2}{n_i^\star}\frac{m+1}{m} - 2\right)}{\frac{4\alpha^2}{n_i^\star}\frac{m+1}{m} - 1} + 1\right) = 2.$$

# B  Proof of Theorem 2

We will use $M_{\text{PCS}}$ to denote the mechanism in §4.1, as it *pools* the datsets, but *checks* for the *size* of the dataset submitted by each agent. For clarity, we have stated $M_{\text{PCS}}$ algorithmically in Algorithm 2. We will also re-state the recommended strategies $s_i^\star = \{(n_i^\star, f_i^\star, h_i^\star)\}_i$ below:

$$n_i^\star = \frac{\sigma}{\sqrt{cm}}, \qquad f_i^\star = \mathbf{I}, \qquad h_i^\star(X_i, Y_i, A_i) = \frac{1}{|X_i \cup A_i|}\sum_{u \in X_i \cup A_i} u \qquad (26)$$

Throughout this section, $s_i^\star$ will refer to (26) (and not (8)).

We will first prove that $s_i^\star$ is a Nash equilibrium. Because the sample mean achieves minimax error for Normal mean estimation [24], we immediately have, for all $(n_i, f_i, h_i) \in \mathcal{S}$.

$$p_i(M_{\text{PCS}}, ((n_i, f_i, h_i^\star), s_{-i}^\star)) \leq p_i(M_{\text{PCS}}, ((n_i, f_i, h_i), s_{-i}^\star)).$$

Because the agent can only submit the raw dataset or a subset, and the agent's allocation only depends on the size of the dataset, the size of the dataset she receives can always be maximized by submittng the whole data set she collects, i.e. chooses $f_i = \mathbf{I}$. Therefore, we have for all $(n_i, f_i, h_i) \in \mathcal{S}$,

$$p_i(M_{\text{PCS}}, ((n_i, f_i^\star, h_i^\star), s_{-i}^\star)) \leq p_i(M_{\text{PCS}}, ((n_i, f_i, h_i^\star), s_{-i}^\star)) \leq p_i(M_{\text{PCS}}, ((n_i, f_i, h_i), s_{-i}^\star)).$$

Finally, we can use the fact that the maximum risk of the sample mean estimator using $n$ points is $\sigma^2/n$ to show that the penalty is minimized when $n_i = n_i^\star = \sigma/\sqrt{cm}$. In particular, we have that if $n_i < \sigma/\sqrt{cm}$,

$$p_i(M_{\text{PCS}}, ((n_i, f_i^\star, h_i^\star), s_{-i}^\star)) = \frac{\sigma^2}{n_i} + cn_i > 2\sigma\sqrt{c}.$$

And if $n_i \geq \sigma/\sqrt{cm}$,

$$p_i(M_{\text{PCS}}, ((n_i, f_i^\star, h_i^\star), s_{-i}^\star)) = \frac{\sigma^2}{n_i + (m-1)\sigma/\sqrt{cm}} + cn_i \geq 2\sigma\sqrt{\frac{c}{m}}$$

Because $2\sigma\sqrt{c} \geq 2\sigma\sqrt{c/m}$, $p_i(M_{\text{PCS}}, ((n_i, f_i^\star, h_i^\star), s_{-i}^\star))$ is minimized when $n_i = \sigma/\sqrt{cm}$. We thus conclude that $s^\star$ is a Nash equilibrium. That is, for all $(n_i, f_i, h_i) \in \mathbb{N} \times \mathcal{F} \times \mathcal{H}$

$$p_i(M_{\text{PCS}}, s^\star) \leq p_i(M_{\text{PCS}}, ((n_i, f_i, h_i), s_{-i}^\star)).$$

Next, the IR and efficiency properties follow trivially from the fact that $p_i(M_{\text{PCS}}, s^\star) = 2\sigma\sqrt{c/m}$ for each agent $i$. In particular, $p_i(M_{\text{PCS}}, s^\star) < p_{\min}^{\text{IR}}$ and $P(M_{\text{PCS}}, s^\star) = 2\sigma\sqrt{cm}$.

## C  Proof of Theorem 3

We will use $M_{\text{CDED}}$ to denote our mechanism in §4.2, as it *corrupts* the *deployed estimate* based on the *difference*. We have stated this mechanism formally in Algorithm 3. We will also re-state the recommended strategies $s_i^\star = \{(n_i^\star, f_i^\star)\}_i$ below:

$$n_i^\star = \frac{\sigma}{\sqrt{cm}}, \qquad f_i^\star = \mathbf{I}. \tag{27}$$

Throughout this section, $s_i^\star$ will refer to (27) (and not (8) or (26)).

We will now present the proof of Theorem 3. First, in §C.1, we show that $s^\star$ is a Nash equilibrium of $M_{\text{CDED}}$ as the Nash incentive compatibility result. Then, in §C.2, we show individual rationality at $s_i^\star$. In §C.3, we conclude by showing that $M_{\text{CDED}}$ is approximately efficient by showing that its social penalty at most a $(1 + \epsilon)$ factor of the global minimum.

### C.1  Algorithm 3 is Nash incentive compatible

**Step 1.** We will first show that fixing any $n_i$, the best strategy is to submit the raw data, i.e. for all $(n_i, f_i) \in \mathbb{N} \times \mathcal{F}$.

$$p_i(M_{\text{CDED}}, ((n_i, f_i^\star), s_{-i}^\star)) \leq p_i(M_{\text{CDED}}, ((n_i, f_i), s_{-i}^\star)). \tag{28}$$

Let $e_{z,i} = \epsilon_{z,i}/\eta_i$, where $\eta_i$, and $\epsilon_{z,i}$ are as given in lines 9 and 10 respectively. We have that $e_{z,i}$'s are i.i.d. standard Normal samples. Because the cost term $cn_i$ is fixed when $n_i$ is fixed, we only need to consider the risk term. We will first define,

$$\widehat{\mu}(X_i) := \frac{1}{|X_i|} \sum_{x \in X_i} x, \quad \widehat{\mu}(Y_i) := \frac{1}{|Y_i|} \sum_{x \in Y_i} x, \quad \widehat{\mu}(Y_{-i}) := \frac{1}{|Y_{-i}|} \sum_{x \in Y_{-i}} x. \tag{29}$$

Via some algebraic manipulations, we can express the maximum risk as:

$$\sup_\mu \mathbb{E}\left[ \left( \frac{1}{|Y_i| + (m-1)n_i^\star} \left( \sum_{y \in Y_i} (y - \mu) + \sum_{z \in Y_{-i}} (z + e_{z,i}\eta_i - \mu) \right) \right)^2 \middle| \mu \right]$$

$$= \frac{1}{(|Y_i| + (m-1)n_i^\star)^2} \sup_\mu \mathbb{E}\left[ \left( \sum_{y \in Y_i} (y - \mu) \right)^2 + \left( \sum_{z \in Y_{-i}} (z + e_{z,i}\eta_i - \mu) \right)^2 \middle| \mu \right]$$

$$= \frac{1}{(|Y_i| + (m-1)n_i^\star)^2} \sup_\mu \mathbb{E}\left[ |Y_i| (\widehat{\mu}(Y_i) - \mu)^2 + \left( \sum_{z \in Y_{-i}} (z - \mu) \right)^2 + \left( \sum_{z \in Y_{-i}} e_{z,i}\eta_i \right)^2 \middle| \mu \right]$$

$$= \frac{1}{(|Y_i| + (m-1)n_i^\star)^2} \sup_\mu \mathbb{E}\left[ |Y_i| (\widehat{\mu}(Y_i) - \mu)^2 + (m-1)n_i^\star \beta_\epsilon^2 (\widehat{\mu}(Y_i) - \widehat{\mu}(Y_{-i}))^{2k_\epsilon} \middle| \mu \right]$$

$$+ \frac{(m-1)n_i^\star \sigma^2}{(|Y_i| + (m-1)n_i^\star)^2}$$

Recall that $\beta_\epsilon$ also involves $|Y_i|$. Note that as we have fixed $n_i$ and $s_{-i} = s_{-i}^\star$, the maximum risk depends only on $|Y_i|$ and $\widehat{\mu}(Y_i)$, that is, the agent's maximum risk and hence penalty only depends on the number of points she submitted, and their average value. Hence, to find the optimal submission

---

**Algorithm 3** $M_{\text{CDED}}$

---

**Require:** Approximation parameter $\epsilon > 0$                # to obtain a $1 + \epsilon$ bound on PR.
1: **Mechanism designer publishes:**    The procedure in lines $5 - 11$.
2: **Each agent $i$:**
3:       Choose strategy $s_i = (n_i, f_i)$.
4:       Sample $n_i$ points $X_i = \{x_{i,j}\}_{j=1}^{n_i}$ and submit $Y_i = f_i(X_i)$ to the mechanism.
5: **Mechanism:**
6:       $k_\epsilon \leftarrow \lceil \frac{1}{2\epsilon} \rceil, \quad \beta_\epsilon \leftarrow \sqrt{\dfrac{\left(\sum_{i=1}^m |Y_i|\right)^2 (m-1)^{k_\epsilon - 1}}{k_\epsilon (2k_\epsilon - 1)!! \sigma^{k_\epsilon} c^{\frac{k_\epsilon - 2}{2}} m^{3k_\epsilon / 2}}}$
7:       **For** each agent $i \in [m]$:             # can be done simultaneously for all agents
8:           $Y_{-i} \leftarrow \bigcup_{j \neq i} Y_j$.
9:           $\eta_i^2 \leftarrow \beta_\epsilon^2 \left( \frac{1}{|Y_i|} \sum_{y \in Y_i} y - \frac{1}{|Y_{-i}|} \sum_{y \in Y_{-i}} y \right)^{2k_\epsilon}$.
10:           $Z_i \leftarrow \{z + \epsilon_{z,i}, \text{ for all } z \in Y_{-i} \text{ where } \epsilon_{z,i} \sim \mathcal{N}(0, \eta_i^2)\}$
11:           Deploy estimate $\left( \frac{1}{|Y_i \cup Z_i|} \sum_{u \in Y_i \cup Z_i} u \right)$ for agent $i$.

---

$Y_i$, we will first fix the size of the agent's submission $|Y_i|$ and optimize for the sample mean $\widehat{\mu}(Y_i)$ (step 1.1), and then we will optimize for $|Y_i|$ (step 1.2).

**Step 1.1.** Since the other agents have each collected $\sigma/\sqrt{cm} = n_i^\star$ points and submitted it truthfully, we have $\widehat{\mu}(Y_{-i}) \sim \mathcal{N}\left(\mu, \frac{\sigma^2}{(m-1)n_i^\star}\right)$. Via a binomial expansion , we can write,

$$
\begin{aligned}
\mathbb{E}\left[ (\widehat{\mu}(Y_i) - \widehat{\mu}(Y_{-i}))^{2k_\epsilon} \right] &= \mathbb{E}\left[ ((\widehat{\mu}(Y_i) - \mu) - (\widehat{\mu}(Y_{-i}) - \mu))^{2k_\epsilon} \right] \\
&= \sum_{j=0}^{2k_\epsilon} (-1)^j \binom{2k_\epsilon}{j} \mathbb{E}\left[ (\widehat{\mu}(Y_i) - \mu)^j \right] \mathbb{E}\left[ (\widehat{\mu}(Y_{-i}) - \mu)^{2k_\epsilon - j} \right] \\
&= \sum_{j=0}^{k_\epsilon} \binom{2k_\epsilon}{2j} \mathbb{E}\left[ (\widehat{\mu}(Y_i) - \mu)^{2j} \right] \mathbb{E}\left[ (\widehat{\mu}(Y_{-i}) - \mu)^{2k_\epsilon - 2j} \right]
\end{aligned}
$$

Thus the maximum risk can be written as:

$$
\sup_\mu \mathbb{E}\left[ \sum_{j=0}^{k_\epsilon} A_j (\widehat{\mu}(Y_i) - \mu)^{2j} \,\middle|\, \mu \right] \tag{30}
$$

where $A_0, \ldots, A_{k_\epsilon}$ is a sequence of positive coefficients.

Similar to the proof of Theorem 1, we construct a lower bound on the maximum risk using a sequence of Bayesian risks. Let $\Lambda_\ell := \mathcal{N}(0, \ell^2)$, $\ell = 1, 2, \ldots$ be a sequence of prior for $\mu$. For fixed $\ell$, the posterior distribution is:

$$
\begin{aligned}
p(\mu | X_i) &\propto p(X_i | \mu) p(\mu) \propto \exp\left( -\frac{1}{2\sigma^2} \sum_{x \in X_i} (x - \mu)^2 \right) \exp\left( -\frac{1}{2\ell^2} \mu^2 \right) \\
&\propto \exp\left( -\frac{1}{2} \left( \frac{n_i}{\sigma^2} + \frac{1}{\ell^2} \right) \mu^2 + \frac{1}{2} 2 \frac{\sum_{x \in X_i} x}{\sigma^2} \mu \right).
\end{aligned}
$$

This means the posterior of $\mu$ given $X_i$ is Gaussian with:

$$
\mu | X_i \sim \mathcal{N}\left( \frac{n_i \widehat{\mu}(X_i)/\sigma^2}{n_i/\sigma^2 + 1/\ell^2}, \frac{1}{n_i/\sigma^2 + 1/\ell^2} \right) =: \mathcal{N}(\mu_\ell, \sigma_\ell^2).
$$

Therefore, the posterior risk is:

$$
\mathbb{E}\left[ \sum_{j=0}^{k_\epsilon} A_j (\widehat{\mu}(Y_i) - \mu)^{2j} \,\middle|\, X_i \right] = \mathbb{E}\left[ \sum_{j=0}^{k_\epsilon} A_j ((\widehat{\mu}(Y_i) - \mu_\ell) - (\mu - \mu_\ell))^{2j} \,\middle|\, X_i \right]
$$

$$= \int_{-\infty}^{\infty} \underbrace{\sum_{j=0}^{k_\epsilon} A_j (e - (\widehat{\mu}(Y_i) - \mu_\ell))^{2j}}_{=:F_1(e-(\widehat{\mu}(Y_i)-\mu_\ell))} \underbrace{\frac{1}{\sigma_\ell \sqrt{2\pi}} \exp\left(-\frac{e^2}{2\sigma_\ell^2}\right)}_{=:F_2(e)} de$$

Because:

- $F_1(\cdot)$ is even function and increases on $[0, \infty)$;
- $F_2(\cdot)$ is even function and decreases on $[0, \infty$, and $\int_{\mathbb{R}} F_2(e)de < \infty$
- For any $a \in \mathbb{R}$, $\int_{\mathbb{R}} F_1(e-a)F_2(e)de < \infty$,

By the corollary of Hardy-Littlewood inequality in Lemma 9,

$$\int_{\mathbb{R}} F_1(e-a)F_2(e)de \geq \int_{\mathbb{R}} F_1(e)F_2(e)de,$$

which means the posterior risk is minimized when $\widehat{\mu}(Y_i) = \mu_\ell$. In Lemma 11, we have stated expressions for the expected value of the power of a normal random variable. Using this, we can write the Bayes risk as:

$$R_\ell := \mathbb{E}\left[\sum_{j=0}^{k_\epsilon} A_j \mathbb{E}\left[(\mu - \mu_\ell)^{2j} \middle| X_i\right]\right] = \sum_{j=0}^{k_\epsilon} A_j(2j-1)!!\sigma_\ell^{2j}$$

and the limit of Bayesian risk as $\ell \to \infty$ is

$$R_\infty := \lim_{\ell \to \infty} \sum_{j=0}^{k_\epsilon} A_j(2j-1)!!\frac{\sigma^{2j}}{n_i^j}$$

When $\widehat{\mu}(Y_i) = \widehat{\mu}(X_i)$, the maximum risk is:

$$\sup_\mu \mathbb{E}\left[\sum_{j=0}^{k_\epsilon} A_j(\widehat{\mu}(Y_i) - \mu)^{2j} \middle| \mu\right] = \sup_\mu \mathbb{E}\left[\sum_{j=0}^{k_\epsilon} A_j(\widehat{\mu}(X_i) - \mu)^{2j} \middle| \mu\right]$$

$$= \sum_{j=0}^{k_\epsilon} A_j(2j-1)!!\sigma^{2j}n_i^{-j} = R_\infty.$$

This means, fixing $n_i$ and $|Y_i|$, agent $i$ achieves minimax risk when choosing $\widehat{\mu}(Y_i) = \widehat{\mu}(X_i)$; as the maximum is larger than the average, this follows using a similar argument to Step 3 in §A.3.

**Step 1.2.** Next, we will show that the best size of the submission is $|Y_i| = |X_i| = n_i$, assuming $\widehat{\mu}(Y_i) = \widehat{\mu}(X_i)$. For this, we will first use $n_i^\star$ to rewrite $\beta_\epsilon^2$ as

$$\beta_\epsilon^2 = \frac{n_i^{\star k_\epsilon - 2}(m-1)^{k_\epsilon-1}(|Y_i| + (m-1)n_i^\star)^2}{k_\epsilon(2k_\epsilon-1)!!m^{k_\epsilon+1}\sigma^{2k_\epsilon-2}}.$$

Because

$$\widehat{\mu}(X_i) - \widehat{\mu}(Y_{-i}) \sim \mathcal{N}\left(0, \left(\frac{1}{n_i} + \frac{1}{(m-1)n_i^\star}\right)\sigma^2\right),$$

the risk term in the penalty can be rewritten and lower bounded as follows:

$$\frac{1}{(|Y_i| + (m-1)n_i^\star)^2}\left(|Y_i|^2 \sigma^2/n_i + (m-1)n_i^\star\beta_\epsilon^2(2k_\epsilon-1)!!\left(\frac{1}{n_i} + \frac{1}{(m-1)n_i^\star}\right)^{k_\epsilon}\sigma^{2k_\epsilon}\right)$$

$$+ \frac{(m-1)n_i^\star\sigma^2}{(|Y_i| + (m-1)n_i^\star)^2}$$

$$= \frac{|Y_i|^2 \frac{\sigma^2}{n_i} + (m-1)n_i^\star\sigma^2}{(|Y_i| + (m-1)n_i^\star)^2} + \frac{n_i^{\star k_\epsilon-1}(m-1)^{k_\epsilon}}{k_\epsilon m^{k_\epsilon+1}}\left(\frac{1}{n_i} + \frac{1}{(m-1)n_i^\star}\right)^{k_\epsilon}\sigma^2$$

$$\geq \frac{\sigma^2}{n_i + (m-1)n_i^\star} + \frac{n_i^{\star k_\epsilon - 1}(m-1)^{k_\epsilon}}{k_\epsilon m^{k_\epsilon+1}} \left( \frac{1}{n_i} + \frac{1}{(m-1)n_i^\star} \right)^{k_\epsilon} \sigma^2.$$

Here, the last step follows from the fact that

$$\frac{|Y_i|^2 \frac{\sigma^2}{n_i} + (m-1)n_i^\star \sigma^2}{(|Y_i| + (m-1)n_i^\star)^2} = \frac{|Y_i|^2 \frac{\sigma^2}{n_i} + (m-1)n_i^\star \sigma^2}{n_i \frac{|Y_i|^2}{n_i} + 2|Y_i|(m-1)n_i^\star + (m-1)^2 n_i^{\star 2}}$$

$$\geq \frac{|Y_i|^2 \frac{\sigma^2}{n_i} + (m-1)n_i^\star \sigma^2}{n_i \frac{|Y_i|^2}{n_i} + \left( n_i + \frac{|Y_i|^2}{n_i} \right)(m-1)n_i^\star + (m-1)^2 n_i^{\star 2}} = \frac{|Y_i|^2 \frac{\sigma^2}{n_i} + (m-1)n_i^\star \sigma^2}{(n_i + (m-1)n_i^\star)\left( \frac{|Y_i|^2}{n_i} + (m-1)n_i^\star \right)}$$

$$= \frac{\sigma^2}{n_i + (m-1)n_i^\star}.$$

Equality holds in this inequality if and only if $|Y_i| = n_i$.

In conclusion, fixing $n_i$, the agent can minimize her penalty by submitting $n_i$ points with the same sample mean as the dataset $X_i$ she collected. One way to achieve this is set $f_i = \mathbf{I}$. This completes the proof of (28).

**Step 2:** Our next step is to show that the agent's best strategy is to collect $n_i^\star$ data points. That is, we will show for all $n_i \in \mathbb{N}$.

$$p_i(M_{\text{CDED}}, ((n_i^\star, f_i^\star), s_{-i}^\star)) \leq p_i(M_{\text{CDED}}, ((n_i, f_i^\star), s_{-i}^\star)). \tag{31}$$

In the following, we will use $p(n_i)$ as a shorthand for $p_i(M_{\text{CDED}}, ((n_i, f_i^\star), s_{-i}^\star))$. The penalty can be rewritten as:

$$p(n_i) = \frac{\sigma^2}{n_i + (m-1)n_i^\star} + \frac{n_i^{\star k_\epsilon - 1}(m-1)^{k_\epsilon}}{k_\epsilon m^{k_\epsilon+1}} \left( \frac{1}{n_i} + \frac{1}{(m-1)n_i^\star} \right)^{k_\epsilon} \sigma^2 + cn_i$$

We need to show that $p_i(n_i)$ achieves minimum at $n_i = n_i^\star$. The derivative of $p_i(\cdot)$ is:

$$p'(n_i) = -\frac{\sigma^2}{(n_i + (m-1)n_i^\star)^2} + \frac{n_i^{\star k_\epsilon - 1}(m-1)^{k_\epsilon}}{m^{k_\epsilon+1}} \left( \frac{1}{n_i} + \frac{1}{(m-1)n_i^\star} \right)^{k_\epsilon - 1} \sigma^2 \left( -\frac{1}{n_i^2} \right) + c$$

Because $p'(n_i)$ increase in $n_i$, $p(n_i)$ is convex. Moreover, because

$$p'(n_i^\star) = -\frac{\sigma^2}{m^2 n_i^{\star 2}} + \frac{n_i^{\star k_\epsilon - 1}(m-1)^{k_\epsilon}}{m^{k_\epsilon+1}} \left( \frac{1}{n_i^\star} + \frac{1}{(m-1)n_i^\star} \right)^{k_\epsilon - 1} \sigma^2 \left( -\frac{1}{n_i^{\star 2}} \right) + c$$

$$= -\frac{\sigma^2}{m^2 n_i^{\star 2}} - \frac{(m-1)\sigma^2}{m^2 n_i^{\star 2}} + c = -\frac{\sigma^2}{m n_i^{\star 2}} + c = 0,$$

we know $p(n_i)$ reaches minimum at $n_i = n_i^\star$. This concludes the proof for (31).

### C.2 Algorithm 3 is individually rational

The penalty of an agent at the recommended strategies can be expressed as:

$$p_i(M_{\text{CDED}}, s_i^\star) = p(n_i^\star) = \frac{\sigma^2}{m n_i^\star} + \frac{n_i^{\star k_\epsilon - 1}(m-1)^{k_\epsilon}}{k_\epsilon m^{k_\epsilon+1}} \left( \frac{1}{n_i^\star} + \frac{1}{(m-1)n_i^\star} \right)^{k_\epsilon} \sigma^2 + cn_i^\star$$

$$= \frac{\sigma^2}{m n_i^\star} + \frac{n_i^{\star k_\epsilon - 1}(m-1)^{k_\epsilon}}{k_\epsilon m^{k_\epsilon+1}} \frac{m^{k_\epsilon}}{n_i^{\star k_\epsilon}(m-1)^{k_\epsilon}} \sigma^2 + cn_i^\star$$

$$= \frac{\sigma^2}{m n_i^\star} + \frac{1}{k_\epsilon} \frac{\sigma^2}{m n_i^\star} + cn_i^\star = \left( 2 + \frac{1}{k_\epsilon} \right) \frac{\sigma\sqrt{c}}{\sqrt{m}}. \tag{32}$$

We have that $M_{\text{CDED}}$ is IR when $m \geq 2$, via the following simple calculation:

$$\left( 2 + \frac{1}{k_\epsilon} \right) \frac{\sigma\sqrt{c}}{\sqrt{m}} \leq \left( 2 + \frac{1}{2} \right) \frac{\sigma\sqrt{c}}{\sqrt{2}} < 2\sigma\sqrt{c} = p_{\min}^{\text{IR}}$$

### C.3 Algorithm 3 is approximately efficient

Using the expression for $p_i(M_{\text{CDED}}, s_i^\star)$ in (32), the penalty ratio can be bounded by:

$$\text{PR}(M_{\text{CDED}}, s^\star) = \frac{\left(2 + \frac{1}{k_\epsilon}\right)\sigma\sqrt{cm}}{2\sigma\sqrt{cm}} = 1 + \frac{1}{2k_\epsilon} \leq 1 + \epsilon.$$

## D  Additional Materials for Section 4.2

### D.1  Mechanism detail

See Algorithm 3.

### D.2  Using a weighted average under the original strategy space from §2

In this section, we will consider a variation of $M_{\text{CDED}}$ when applied to our original strategy space $\mathbb{N} \times \mathcal{F} \times \mathcal{H}$. For this, we will assume that $M_{\text{CDED}}$ will return $A_i = Z_i$ as the agent's allocation, and then an agent can use $X_i, Y_i, Z_i$ to estimate $\mu$. In this situation, below we show that the agent can achieve a smaller penalty using a weighted average over $X_i \cup Z_i$ instead of the sample mean used by the mechanism. Here, the weights are proportional to the inverse of the variance of each data point. (Our mechanism purposefully uses the sub-optimal sample mean in the restricted strategy space $\mathbb{N} \times \mathcal{F}$ as a way to shape the agent's penalty and incentivize good behavior.)

This shows that $M_{\text{CDED}}$ (with the above modification) is not NIC in this more general strategy space. The agent can obtain a lower penalty using a better estimator (such as the weighted average we show over here) and achieve a lower penalty. More importantly, as the agent knows that she can achieve a lower estimation error via a better estimator instead of more data, she can leverage this insight to collect less data and reduce her penalty even further.

We should emphasize that it is unclear if this weighted average is minimax optimal. It is also unclear if there exists a Nash equilibrium for $M_{\text{CDED}}$ (or any straightforward modification of $M_{\text{CDED}}$) in the expanded strategy space.

**The weighted average estimator:**  We will now present the weighted average estimator that achieves a lower maximum risk. To show this, first note that for all $x \in X_i$, $\mathbb{V}[x] = \sigma^2$; when $(n_i, f_i) = (n_i^\star, f_i^\star)$, for all $x \in Z_i$,

$$
\begin{aligned}
\mathbb{V}[x] &= \mathbb{E}\left[(z + \epsilon_{z,i} - \mu)^2\right] = \sigma^2 + \beta_\epsilon^2 \mathbb{E}\left[(\widehat{\mu}(X_i) - \widehat{\mu}(Y_{-i}))^{2k_\epsilon}\right] \\
&= \sigma^2 + \frac{n_i^{\star k_\epsilon - 2}(m-1)^{k_\epsilon - 1}(mn_i^\star)^2}{k_\epsilon(2k_\epsilon - 1)!!m^{k_\epsilon + 1}\sigma^{2k_\epsilon - 2}}(2k_\epsilon - 1)!!\left(\frac{1}{n_i^\star} + \frac{1}{(m-1)n_i^\star}\right)^{k_\epsilon}\sigma^{2k_\epsilon} \\
&= \sigma^2 + \frac{n_i^{\star k_\epsilon}(m-1)^{k_\epsilon - 1}}{k_\epsilon m^{k_\epsilon - 1}}\frac{m^{k_\epsilon}}{(m-1)^{k_\epsilon}n_i^{\star k_\epsilon}}\sigma^2 \\
&= \sigma^2 + \frac{1}{k_\epsilon}\frac{m}{m-1}\sigma^2
\end{aligned}
$$

Consider the following weighted-average estimator:

$$h_i(X_i, Y_i, (Z_i, \eta_i^2)) = \frac{\frac{1}{\sigma^2}\sum_{x \in X_i} x + \frac{1}{\sigma^2 + \frac{1}{k_\epsilon}\frac{m}{m-1}\sigma^2}\sum_{x \in Z_i} x}{\frac{n_i^\star}{\sigma^2} + \frac{(m-1)n_i^\star}{\sigma^2 + \frac{1}{k_\epsilon}\frac{m}{m-1}\sigma^2}}$$

The maximum risk of $h_i$ is

$$
\begin{aligned}
\mathbb{E}\left[\left(h_i(X_i, Y_i, (Z_i, \eta_i^2)) - \mu\right)^2\right] &= \frac{1}{\frac{n_i^\star}{\sigma^2} + \frac{(m-1)n_i^\star}{\sigma^2 + \frac{1}{k_\epsilon}\frac{m}{m-1}\sigma^2}} = \frac{1}{1 + \frac{m-1}{1 + \frac{1}{k_\epsilon}\frac{m}{m-1}}}\frac{\sigma^2}{n_i^\star} = \frac{1 + \frac{1}{k_\epsilon}\frac{m}{m-1}}{m + \frac{1}{k_\epsilon}\frac{m}{m-1}}\frac{\sigma^2}{n_i^\star} \\
&< \frac{\left(1 + \frac{1}{k_\epsilon}\right)\left(1 + \frac{1}{k_\epsilon}\frac{1}{m-1}\right)}{m + \frac{1}{k_\epsilon}\frac{m}{m-1}}\frac{\sigma^2}{n_i^\star} = \left(1 + \frac{1}{k_\epsilon}\right)\frac{\sigma^2}{mn_i^\star} \quad (33)
\end{aligned}
$$

Note that the RHS of (33) is the risk of the sample average deployed by $M_{\text{CDED}}$. This means, suppose all other agents choose $s^\star$, then agent $i$ can choose a weighted average to reduce her penalty without collecting more data.

# E  High dimensional mean estimation with bounded variance

In this section, we will study estimating a $d$–dimensional mean $\mu(\theta) \in \mathbb{R}^d$ for distributions $\theta$ with bounded variance. We will focus on our original setting in §2, but will outline the modifications to the formalism to accommodate the generality. For $x \in \mathbb{R}^d$, let $x^{(i)}$ denote the $i^{\text{th}}$ dimension.

**Modifications to the setting in §2:** First, we should change the definitions of $\mathcal{F}, \mathcal{H}$ and $\mathcal{M}$ in equations 1 and (2) to account for the fact that the data is $d$ dimensional. For instance, the space of functions mapping the dataset collected to the dataset submitted should be defined as $\mathcal{F} = \{f : \bigcup_{n\geq 0} \mathbb{R}^{d\times n} \to \bigcup_{n\geq 0} \mathbb{R}^{d\times n}\}$. Next, let $\Theta = \{\theta; \; \textbf{supp}\,(\theta) \subset \mathbb{R}^d, \; \mathbb{E}_{x\sim\theta}\left[(x^{(i)} - \mu(\theta)^{(i)})^2\right] \leq \sigma^2, \; \forall\, i \in [d]\}$ be the class of all $d$–dimensional distributions where the variance along each dimension is bounded by $\sigma^2$. Here, the maximum variance $\sigma^2$ is known and is public information. Note that we do not assume that the individual dimensions are independent. An agent's penalty $p_i$ is defined similar to (3) but considers the maximum risk over $\Theta$, i.e

$$p_i(M, s) = \sup_{\theta\in\Theta} \mathbb{E}\left[\|h_i(X_i, Y_i, A_i) - \mu(\theta)\|_2^2 \mid \theta\right] + cn_i. \tag{34}$$

Finally, the social penalty and ratio PR are as defined in (5), but with the above definition for $p_i$.

**Mechanism:** Our mechanism for this problem is the same as the one outlined in Algorithm 1, with the following cosmetic modifications. First, the allocation space should now be $\mathcal{A} = \bigcup_{n\geq 0} \mathbb{R}^{d\times n} \times \bigcup_{n\geq 0} \mathbb{R}^{d\times n} \times \mathbb{R}_+^d$. The noise modulating parameter $\alpha$ is determined by a similar equation as in (7), but with $c$ replaced with $c/d$. In line 12 of Algorithm 1, we should set the size of the dataset $Z_i$ to be $\min\{|Y_{-i}|, \sigma\sqrt{d/(cm)}\}$. Finally, the operations in lines 13 and 14 should be interpreted as $d$–dimensional operations that are performed elementwise. The recommended strategy $s_i^\star = (n_i^\star, f_i^\star, h_i^\star)$ for agent $i$ is as follows:

$$n_i^\star = \begin{cases} \frac{\sigma}{m}\sqrt{\frac{d}{c}} & \text{if } m \leq 4, \\ \sigma\sqrt{\frac{d}{cm}} & \text{if } m \geq 5 \end{cases}, \qquad\qquad f_i^\star = \mathbf{I}, \tag{35}$$

$$h_i^\star(X_i, Y_i, (Z_i, Z_i', \eta_i^2)) = \frac{\frac{1}{\sigma^2}\sum_{u\in X_i\cup Z_i} u + \frac{1}{\sigma^2+\tau_i^2}\sum_{u\in Z_i'} u}{\frac{1}{\sigma^2}|X_i\cup Z_i'| + \frac{1}{\sigma^2+\tau_i^2}|Z_i'|}, \qquad \text{where,}\; \tau_i^2 = \frac{2\alpha^2\sigma^2}{n_i^\star} \in \mathbb{R}_+.$$

Above, one difference worth highlighting is the change in the recommended estimator $h_i^\star$. Previously, the weighting used the $\eta_i^2$ term returned by the mechanism, which is a function of $Y_i$ and $Z_i$. This data-dependent weighting was necessary to obtain an *exactly* (i.e including constants) minimax optimal estimator for the corrupted dataset, which in turn was necessary to achieve an exact Nash equilibrium. However, bounding the risk when using a data-dependent weighting is challenging when the Gaussian assumption does not hold. Instead, here we use a deterministic weighting via the quantity $\tau_i^2$. While this is not exactly minimax optimal, we can show that its maximum risk is very close to a lower bound, which helps us obtain an approximate Nash equilibrium. It is worth pointing out that designing exactly minimax optimal estimators, even under i.i.d assumptions, is challenging for general classes of distributions [24].

The following theorem states the main properties of this mechanism.

**Theorem 7.** *The following statements are true about the mechanism $M_{\text{C3D}}$ in Algorithm 1 with the above modifications. (i) The strategy profile $s^\star$ as defined in (35) is an approximate Nash equilibrium, i.e if all agents except $i$ are following $s^\star$, then for any alternative strategy $s_i$ for agent $i$, we have $p_i(M_{\text{C3D}}, s^\star) \leq p_i(M_{\text{C3D}}, (s_{-i}^\star, s_i))(1 + 5/m)$ (ii) The mechanism is individually rational at $s^\star$. (iii) The mechanism is approximately efficient at $s_i^\star$, with $\text{PR}(M_{\text{C3D}}, s^\star) < 2 + 10/m$.*

We see that even under this more general setting, our mechanism retains its main properties with only a slight weakening of the results. We now have approximate, instead of exact, NIC, with the benefit of deviation diminishing as there are more agents. Similarly, the bound on the efficiency is only slightly weaker than the one in Theorem 1.

## E.1 Proof of Theorem 7

When $m \leq 4$, the claims follow using the exact steps in §A.1. Therefore, we focus on the case $m \geq 5$. Moreover, some of the key steps of this proof follows along similar lines to Theorem 1, so we will provide an outline and focus on the differences.

**Approximate Nash incentive compatibility.** We will first prove the statement *(i)* of Theorem 7, which states that $s_i^\star$, as defined in (35), is an approximate Nash equilibrium for $M_{\text{C3D}}$. That is, we will show that the maximum possible reduction in penalty for an agent $i$ when deviating from $s_i^\star$ is small, provided that all other agents are following $s_{-i}^\star$.

For this, we will first lower bound the penalty $p_i$ (34) using the family of independent Gaussian distributions. Let $\Theta_{\mathcal{N}} = \left\{ \mathcal{N}(\mu, \sigma^2 I_d) : \mu \in \mathbb{R}^d \right\}$ denote the space of $d$–dimensional normal distributions with identity covariance matrix. For any mechanism $M$ and strategy profile $s \in \mathcal{S}^m$, we define the penalty of agent $i$ restricted to $\Theta_{\mathcal{N}}$ as:

$$p_i^{\mathcal{N}}(M, s) = \sup_{\theta \in \Theta_{\mathcal{N}}} \mathbb{E}\left[ \|h_i(X_i, Y_i, A_i) - \mu(\theta)\|_2^2 \,\big|\, \theta \right] + cn_i.$$

Since $\Theta_{\mathcal{N}} \subset \Theta$, it is straightforward to see that for all $M \in \mathcal{M}$ and $s \in \mathcal{S}^m$,

$$p_i^{\mathcal{N}}(M, s) \leq p_i(M, s). \tag{36}$$

We will now use this result to lower bound the penalty of an agent for any other alternative strategy. First note that, by independence, the mean estimation problem on $\Theta_{\mathcal{N}}$ can be viewed as $d$ independent copies of the univariate normal mean estimation problem considered in Theorem 1 but with $c$ replaced with $c/d$. Let $\tilde{h}_i^\star$ be the weighted average that applies the estimator in (8) along each dimension. And let $\tilde{s}_i^\star = (n_i^\star, f_i^\star, \tilde{h}_i^\star)$. We can now lower bound the penalty of agent $i$ when following any (alternative) strategy $s_i \in \mathcal{S}$, provided that other agents are following $s_{-i}^\star$. We have:

$$
\begin{aligned}
p_i(M_{\text{C3D}}, (s_i, s_{-i}^\star)) &= p_i\big(M_{\text{C3D}}, \big(s_i, \big(n_{-i}^\star, f_{-i}^\star, h_{-i}^\star\big)\big)\big) \\
&\geq p_i^{\mathcal{N}}\big(M_{\text{C3D}}, \big(s_i, \big(n_{-i}^\star, f_{-i}^\star, h_{-i}^\star\big)\big)\big) \quad\quad \text{(By (36))} \\
&= p_i^{\mathcal{N}}\Big(M_{\text{C3D}}, \Big(s_i, \big(n_{-i}^\star, f_{-i}^\star, \tilde{h}_{-i}^\star\big)\Big)\Big)
\end{aligned}
$$

(As agent $i$'s penalty will not be affected by other agents' estimators)

$$
\begin{aligned}
&\geq p_i^{\mathcal{N}}\Big(M_{\text{C3D}}, \Big(\big(n_i^\star, f_i^\star, \tilde{h}_i^\star\big), \big(n_{-i}^\star, f_{-i}^\star, \tilde{h}_{-i}^\star\big)\Big)\Big)
\end{aligned}
$$

( By adapting the analysis in §A.3. )

$$
= p_i^{\mathcal{N}}(M_{\text{C3D}}, \tilde{s}^\star) \tag{37}
$$

Above, the second step uses (36) and the third step uses the fact that other agent's *estimator* will not affect agent $i$'s penalty. The fourth step uses the fact that for estimation problems in $\Theta_{\mathcal{N}}$, the strategy profile $\tilde{s}^\star = \{(n_i^\star, f_i^\star, \tilde{h}_i^\star)\}_i$ is a Nash equilibrium; in §A.3, we showed this for the one dimensional case, but this proof can be easily adapted to $d$ dimensions since we are assuming an identity covariance matrix in $\Theta_{\mathcal{N}}$. Finally, by adapting the analysis in §A.5, we can obtain the following expression for agent $i$'s penalty $p_i^{\mathcal{N}}(M_{\text{C3D}}, \tilde{s}^\star)$ in $\Theta_{\mathcal{N}}$:

$$p_i^{\mathcal{N}}(M_{\text{C3D}}, \tilde{s}^\star) = d\sigma \sqrt{\frac{c/d}{m}} \left( \frac{\frac{10\alpha^2}{n_i^\star} - 1}{\frac{4\alpha^2}{n_i^\star} \frac{m+1}{m} - 1} + 1 \right) \tag{38}$$

To state the approximate NIC result, we will now upper bound the penalty of the agent when following $s_i^\star$. Using the bounded variance assumption, we have:

$$p_i(M, s^\star) = \sup_{\theta \in \Theta} \mathbb{E}\left[ \left\| \frac{\frac{1}{\sigma^2} \sum_{u \in X_i \cup Z_i} u + \frac{1}{\sigma^2 + \tau_i^2} \sum_{u \in Z_i'} u}{\frac{1}{\sigma^2} |X_i \cup Z_i'| + \frac{1}{\sigma^2 + \tau_i^2} |Z_i'|} - \mu(\theta) \right\|_2^2 \,\Bigg|\, \theta \right] + cn_i^\star$$

$$= \sup_{\theta \in \Theta} \sum_{k=1}^{d} \mathbb{E}\left[ \left( \frac{\frac{1}{\sigma^2} \sum_{u \in X_i \cup Z_i} \big(u^{(k)} - \mu(\theta)^{(k)}\big) + \frac{1}{\sigma^2 + \tau_i^2} \sum_{u \in Z_i'} \big(u^{(k)} - \mu(\theta)^{(k)}\big)}{\frac{1}{\sigma^2} |X_i \cup Z_i'| + \frac{1}{\sigma^2 + \tau_i^2} |Z_i'|} \right)^2 \,\Bigg|\, \theta \right] + cn_i^\star$$

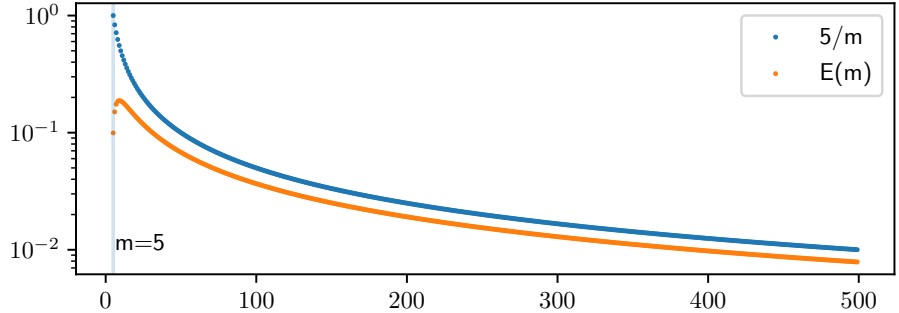

Figure 2: $E(m)$ plot. See `G_em_plot.py`.

$$= \sup_{\theta \in \Theta} \sum_{k=1}^{d} \frac{\frac{1}{\sigma^2} \sum_{u \in X_i \cup Z_i} \mathbb{E}\big[(u^{(k)} - \mu(\theta)^{(k)})\big] + \frac{1}{\sigma^2 + \tau_i^2} \sum_{u \in Z_i'} \mathbb{E}\big[(u^{(k)} - \mu(\theta)^{(k)})\big]}{\frac{1}{\sigma^2}|X_i \cup Z_i'| + \frac{1}{\sigma^2 + \tau_i^2}|Z_i'|} + c n_i^{\star} \quad (39)$$

$$\leq \frac{d}{\frac{2n_i^{\star}}{\sigma^2} + \frac{(m-2)n_i^{\star}}{\sigma^2 + \frac{2\alpha^2 \sigma^2}{n_i^{\star}}}} + c n_i^{\star} = \frac{\sigma^2}{n_i^{\star}} \frac{d}{2 + \frac{m-2}{1 + \frac{2\alpha^2}{n_i^{\star}}}} + c n_i^{\star} = \sigma \sqrt{\frac{cd}{m}} \left( \frac{m}{2 + \frac{m-2}{1 + \frac{2\alpha^2}{n_i^{\star}}}} + 1 \right), \quad (40)$$

where (39) is because: for all $k \in [d]$, $\forall x_1^{(k)}, x_2^{(k)} \in X_i \cup Z_i$, $\forall z_1^{(k)}, z_2^{(k)} \in Z_i'$, $x_1^{(k)} - \mu^{(k)}, x_2^{(k)} - \mu^{(k)}, z_1^{(k)} - \mu^{(k)}, z_2^{(k)} - \mu^{(k)}$ are uncorrelated pairwise. The final inequality is due to the bounded variance assumption.

Next, for brevity, let us write $A_m := \frac{\alpha}{\sqrt{n_i^{\star}}}$ where $\alpha$ is as defined in (7). By adapting the analysis in §A.2, we can show that

$$A_m := \frac{\alpha}{\sqrt{n_i^{\star}}} \in \left(1, 1 + \frac{C_m}{m}\right), \qquad \text{where,} \quad C_m = \begin{cases} 20, & \text{if } m \leq 20 \\ 5, & \text{if } m > 20 \end{cases}. \quad (41)$$

By combining the results in (37), (40), and (41), we obtain the following bound:

$$\frac{p_i(M_{\text{C3D}}, s^{\star})}{\inf_{s_i} p_i(M_{\text{C3D}}, (s_i, s_{-i}^{\star}))} - 1 \leq \frac{p_i(M_{\text{C3D}}, s^{\star})}{p_i^{\mathcal{N}}(M_{\text{C3D}}, \tilde{s}^{\star})} - 1$$

$$\leq \frac{\sigma \sqrt{\frac{cd}{m}} \left( \frac{m}{2 + \frac{m-2}{1 + 2A_m^2}} + 1 \right)}{d\sigma \sqrt{\frac{c/d}{m}} \left( \frac{10A_m^2 - 1}{4A_m^2 \frac{m+1}{m} - 1} + 1 \right)} - 1 = \frac{\frac{m}{2 + \frac{m-2}{1 + \frac{2\alpha^2}{n_i^{\star}}}} + 1}{\frac{\frac{10\alpha^2}{n_i^{\star}} - 1}{\frac{4\alpha^2}{n_i^{\star}} \frac{m+1}{m} - 1} + 1} - 1$$

$$= \frac{4A_m^2 \big((A_m^2 - 1)m + 1 - 4A_m^2\big)m}{(4A_m^2 + m)((7A_m^2 - 1)m + 2A_m^2)} =: E(m). \quad (42)$$

Let $E(m)$ denote the final upper bound obtained above. Next, we will prove $E(m) < 5/m$. When $m \in [5, 500]$, this can be individually verified for each value of $E(m)$ (see Figure 2). When $m \geq 500$, we have $A_m \leq 1.01$ (see (41)). From this we can conclude,

$$E(m) \leq \frac{4 \times 1.01^2 \times (2.01 \times \frac{5}{m}m - 3)m}{6m^2} < \frac{5}{m}. \quad (43)$$

Combining the results in (42) and (43), we obtain the following approximate NIC result:

$$\forall\, i \in [m],\, s_i \in \mathcal{S}, \qquad p_i(M_{\text{C3D}}, s^{\star}) \leq p_i(M_{\text{C3D}}, (s_i, s_{-i}^{\star})) \left(1 + \frac{5}{m}\right).$$

**Individual rationality:** This proof is very similar to the proof in §A.4. In particular, using calculations similar to (40), we can show that regardless of the choice of $n_i$, the agent's penalty is strictly

smaller when using the uncorrupted $(Z_i)$ and corrupted $(Z_i')$ datasets along with the weighted average in (35).

**Approximate efficiency:** To bound the penalty ratio, first note that by (38) and using the same reasoning as §A.5, we have that

$$\frac{\sum_i p_i^{\mathcal{N}}(M_{\text{C3D}}, \tilde{s}^\star)}{\inf_{M \in \mathcal{M}, s \in \mathcal{S}^m} \sum_i p_i^{\mathcal{N}}(M, s)} = \frac{m p_i^{\mathcal{N}}(M_{\text{C3D}}, \tilde{s}^\star)}{\inf_{M \in \mathcal{M}, s \in \mathcal{S}^m} \sum_i p_i^{\mathcal{N}}(M, s)} = \frac{m p_i^{\mathcal{N}}(M_{\text{C3D}}, \tilde{s}^\star)}{2\sigma\sqrt{cmd}} \le 2. \quad (44)$$

Next, as $\Theta_{\mathcal{N}} \subset \Theta$, and noting that $P(M, s) = \sum_i p_i(M, s)$ for all $M, s$, we can also write,

$$\inf_{M \in \mathcal{M}, s \in \mathcal{S}^m} \sum_i p_i^{\mathcal{N}}(M, s) \le \inf_{M \in \mathcal{M}, s \in \mathcal{S}^m} P(M, s). \quad (45)$$

We can combine the above results to obtain the following upper bound on PR:

$$
\begin{aligned}
\text{PR}(M_{\text{C3D}}, s^\star) = \frac{P(M_{\text{C3D}}, s^\star)}{\inf_{M \in \mathcal{M}, s \in \mathcal{S}^m} P(M, s)} &\le \frac{m p_i(M_{\text{C3D}}, s^\star)}{\inf_{M \in \mathcal{M}, s \in \mathcal{S}^m} \sum_i p_i^{\mathcal{N}}(M, s)} && \text{(By (45))} \\
&= \frac{m p_i^{\mathcal{N}}(M_{\text{C3D}}, \tilde{s}^\star)}{\inf_{M \in \mathcal{M}, s \in \mathcal{S}^m} \sum_i p_i^{\mathcal{N}}(M, s)} \frac{p_i(M_{\text{C3D}}, s^\star)}{p_i^{\mathcal{N}}(M_{\text{C3D}}, \tilde{s}^\star)} \\
&\le 2 \frac{p_i(M_{\text{C3D}}, s^\star)}{p_i^{\mathcal{N}}(M_{\text{C3D}}, \tilde{s}^\star)} && \text{(By (44))} \\
&= 2(1 + E(m))) && \text{(By definition of } E(m), \text{ see (42))} \\
&< 2 + \frac{10}{m}. && \text{(By (43))}
\end{aligned}
$$

This establishes approximate efficiency for $M_{\text{C3D}}$ for the high dimensional setting.

# F   Application to Bayesian Settings

While our results study the Normal mean estimation in frequentist statistics, the main ideas can also be applied to the Bayesian setting. When the Normal mean admits a zero-mean normal prior, the major proof steps remain the same. Specifically, our current analysis constructs a sequence of Gaussian priors and takes the limit to prove the minimax optimality. In the Bayesian setting, one can simply skip the step in (22), which takes the limit w.r.t. the prior sequence. The other steps remain the same.

# G   Useful Results

In this section, we will state some useful results that we have used throughout this proof.

**Lemma 8** (Hardy-Littlewood inequality, Lemma 1.6 in Burchard [10]). *Let $f$ and $g$ be non-negative measurable functions that vanish at infinity. Let $f^*$ and $g^*$ to denote the symmetric decreasing rearrangement of $f$ and $g$. If $\int f^* g^* < \infty$, then,*

$$\int fg \le \int f^* g^*.$$

Next, we will use the above result to derive a corollary that will be useful in our proofs.

**Lemma 9** (A corollary of Hardy-Littlewood). *Let $f, g$ be nonnegative even functions such that,*

- *$f$ is monotonically increasing on $[0, \infty)$.*
- *$g$ is monotonically decreasing on $[0, \infty)$, and has a finite integral $\int_{\mathbb{R}} g(x)dx < \infty$.*
- *$\forall a, \int_{\mathbb{R}} f(x - a)g(x)dx < \infty$.*

*Then for all $a$,*

$$\int_{\mathbb{R}} f(x)g(x)dx \le \int_{\mathbb{R}} f(x - a)g(x)dx$$

*Proof.* We will break this proof into two cases. The first is when $\sup f < \infty$ and the second is when $\sup f = \infty$. First consider the case $\sup f < \infty$. Let

$$M := \lim_{x \to \infty} f(x).$$

By using Lemma 8, $\forall a$,

$$\int_{\mathbb{R}} (M - f(x))g(x)dx \geq \int_{\mathbb{R}} (M - f(x - a))g(x)dx.$$

The result follows after rearrangement.

If $\sup f = \infty$, let $f_n(x) := \min\{f(x), n\}$. For all $n$ and $a$, by Lemma 8,

$$\int_{\mathbb{R}} (n - f_n(x))g(x)dx \geq \int_{\mathbb{R}} (n - f_n(x - a))g(x)dx,$$

thus

$$\int_{\mathbb{R}} f_n(x)g(x)dx \leq \int_{\mathbb{R}} f_n(x - a)g(x)dx.$$

Note that $|f_n(x)g(x)| \leq f(x)g(x)$, the result follows by letting $n \to \infty$ on both sides and using dominated convergence theorem. $\square$

Below, we provide a brief example on using Lemma 9 to calculate the Bayes risk in a normal mean estimation problem with i.i.d data. While it is not necessary to use Hardy-Littlewood for this problem, this example will illustrate how we have used it in our proofs.

**Example 1.** *Consider the Normal mean estimation problem given samples $X_{[n]} \sim \mathcal{N}(\mu, \sigma^2)$, where $\mu$ admits a prior distribution $\mathcal{N}(0, \ell^2)$. The goal is to minimize the average risk:*

$$\mathbb{E}_{\mu \sim \mathcal{N}(0, \ell^2)}\left[\mathbb{E}_{X_{[n]} \sim \mathcal{N}(\mu, \sigma^2)}[L(\hat{\mu} - \mu)|\mu]\right],$$

*where the loss function, $L(\cdot)$, is an even function that increases on $[0, \infty)$. By a standard argument, one can show that the posterior distribution of $\mu$ conditioned on $X_{[n]}$ is Gaussian with data-dependent parameters $\bar{\mu}, \bar{\sigma}^2$:*

$$\mu|X_{[n]} \sim \mathcal{N}(\bar{\mu}, \bar{\sigma}^2).$$

*The posterior risk is:*

$$\mathbb{E}_{\mu|X_{[n]}}[L(\hat{\mu} - \mu)] = \mathbb{E}_{\mu|X_{[n]}}[L((\mu - \bar{\mu}) + (\bar{\mu} - \hat{\mu}))] = \int_{\mathbb{R}} \underbrace{L(x + (\bar{\mu} - \hat{\mu}))}_{=:f(x+(\bar{\mu}-\hat{\mu}))} \underbrace{\frac{\exp\left(-\frac{x^2}{2\bar{\sigma}^2}\right)}{\bar{\sigma}\sqrt{2\pi}}}_{=:g(x)} dx$$

*By applying Lemma 9 with $f$ and $g$, the posterior risk above is minimized when $\hat{\mu} = \bar{\mu}$. So is the average risk.*

The next Lemma shows that convexity is preserved under expectation under certain conditions.

**Lemma 10.** *Let $y$ be a random variable and $f(x, y)$ be a function s.t.*

- $f(x, y)$ *is convex in $x$;*

- $\mathbb{E}_y[|f(x, y)|] < \infty$ *for all $x$.*

*Then $\mathbb{E}_y[f(x, y)]$ is also convex in $x$.*

*Proof.* For any $x_1, x_2$, we have

$$\frac{\mathbb{E}_y[f(x_1, y)] + \mathbb{E}_y[f(x_2, y)]}{2} = \mathbb{E}_y\left[\frac{f(x_1, y) + f(x_2, y)}{2}\right] \geq \mathbb{E}_y\left[f\left(\frac{x_1 + x_2}{2}, y\right)\right]$$

$\square$

**Lemma 11** (Centered moments of normal random variable). *Let $X \sim \mathcal{N}(\mu, \sigma^2)$ be a normal random variable and $p \in \mathbb{Z}_+$, then*

$$\mathbb{E}[(X - \mu)^p] = \begin{cases} 0 & \text{if } p \text{ is odd} \\ \sigma^p(p - 1)!! & \text{if } p \text{ is even} \end{cases}.$$

## G.1 Some technical results

Next, we will state some technical results that were obtained purely using algebraic manipulations and are not central to the main proof ideas. The first result states upper and lower bounds on the Gaussian complementary error function using an asymptotic expansion.

**Lemma 12** (Erfc bound). *For all $x > 0$,*

$$\text{Erfc}(x) \leq \frac{1}{\sqrt{\pi}}\left(\frac{\exp(-x^2)}{x} - \frac{\exp(-x^2)}{2x^3} + \frac{3\exp(-x^2)}{4x^5}\right) \tag{46}$$

$$\text{Erfc}(x) \geq \frac{1}{\sqrt{\pi}}\left(\frac{\exp(-x^2)}{x} - \frac{\exp(-x^2)}{2x^3}\right) \tag{47}$$

*Proof.* By integration by parts:

$$\frac{\sqrt{\pi}}{2}\text{Erfc}(x) = \int_x^\infty \exp(-t^2)dt = \left(-\frac{\exp(-t^2)}{2t}\right)\Big|_x^\infty - \int_x^\infty \frac{\exp(-t^2)}{2t^2}dt$$

$$= \frac{\exp(-x^2)}{2x} - \left(\left(-\frac{\exp(-t^2)}{4t^3}\right)\Big|_x^\infty - \int_x^\infty \frac{3\exp(-t^2)}{4t^4}dt\right)$$

$$= \frac{\exp(-x^2)}{2x} - \frac{\exp(-x^2)}{4x^3} + \underbrace{\int_x^\infty \frac{3\exp(-t^2)}{4t^4}dt}_{\geq 0} \tag{48}$$

$$= \frac{\exp(-x^2)}{2x} - \frac{\exp(-x^2)}{4x^3} + \left(-\frac{3\exp(-t^2)}{8t^5}\right)\Big|_x^\infty - \int_x^\infty \frac{15\exp(-t^2)}{8t^6}dt$$

$$= \frac{\exp(-x^2)}{2x} - \frac{\exp(-x^2)}{4x^3} + \frac{3\exp(-x^2)}{8x^5} \underbrace{- \int_x^\infty \frac{15\exp(-t^2)}{8t^6}dt}_{\leq 0} \tag{49}$$

The results follow by (48) and (49). $\qquad\square$

Our next result, states an expression for the function $p(n_i)$ and its derivative as defined in (23).

**Lemma 13** (Value and derivative of penalty function at $s^\star$). *Let $p(n_i) = p_i(M_{\text{C3D}}, ((n_i, f_i^\star, h_i^\star), s_{-i}^\star))$ (see (23)) and $s_i^\star, f_i^\star, h_i^\star$ be as specified in (8). The penalty of agent $i$ in Algorithm 1 satisfies:*

$$p(n_i^\star) = \frac{\sqrt{\frac{\alpha^2}{mn_i^\star}}\sigma^2\left(2m\sqrt{2\pi}\sqrt{\frac{\alpha^2}{mn_i^\star}} - \exp\left(\frac{mn_i^\star}{8\alpha^2}\right)(m-2)\pi\,\text{Erfc}\left(\frac{1}{2\sqrt{2}\sqrt{\frac{\alpha^2}{mn_i^\star}}}\right)\right)}{4\sqrt{2\pi}\alpha^2} + cn_i^\star \tag{50}$$

$$p'(n_i^\star) = -\frac{\sigma^2}{64\frac{\alpha^2}{m-2}\frac{\alpha}{\sqrt{mn_i^\star}}mn_i^\star}\left(\frac{4\alpha}{\sqrt{mn_i^\star}}\left(\frac{4\alpha^2 m}{(m-2)n_i^\star} - 1\right)\right.$$

$$\left. - \exp\left(\frac{mn_i^\star}{8\alpha^2}\right)\left(\frac{4\alpha^2}{mn_i^\star}(m+1) - 1\right)\sqrt{2\pi}\,\text{Erfc}\left(\frac{1}{2\sqrt{2}\sqrt{\frac{\alpha^2}{mn_i^\star}}}\right)\right) + c. \tag{51}$$

This proof involves several algebraic manipulations, so we will provide an outline of our proof strategy. First, we will rearrange the denominator inside the expectation in (23), to write the LHS of (50) as $J + K\mathbb{E}\left[\frac{1}{L+x^2}\right]$, and the LHS of (51) as $J' + K'\mathbb{E}\left[\frac{1}{L+x^2}\right] + K''\mathbb{E}\left[\frac{1}{(L+x^2)^2}\right]$, where the expectation is with respect to a standard normal $\mathcal{N}(0,1)$ variable, $J, K, K', K'', L$ are quantities that depend on $n_i, m, c, \sigma^2, \alpha^2$, and importantly, $L$ is strictly larger than 0. Using properties of the normal distribution, in Lemma 14, we prove the following result:

$$\mathbb{E}\left[\frac{1}{L+x^2}\right] = \sqrt{\frac{\pi}{2L}}\exp\left(\frac{L}{2}\right)\text{Erfc}\left(\sqrt{\frac{L}{2}}\right) \tag{52}$$

$$\mathbb{E}\left[\frac{1}{(L+x^2)^2}\right] = \frac{\sqrt{\pi}}{2\sqrt{2}L^{3/2}}(1-L)\exp\left(\frac{L}{2}\right)\mathrm{Erfc}\left(\sqrt{\frac{L}{2}}\right) + \frac{1}{2L} \tag{53}$$

By plugging in these expressions and then substituting $n_i = n_i^\star$, we obtain (50) and (51).

*Proof of Lemma 13.* We will rewrite $p(n_i^\star)$ and $p'(n_i^\star)$ as the Gaussian integral of rational functions and use (52) to calculate their values. By (23),

$$p(n_i^\star) = \mathbb{E}_{x\sim\mathcal{N}(0,1)}\left[\frac{1}{\frac{(m-2)n_i^\star}{\sigma^2+\alpha^2\left(\frac{\sigma^2}{n_i^\star}+\frac{\sigma^2}{n_i^\star}\right)x^2} + \frac{n_i^\star+n_i^\star}{\sigma^2}}\right] + cn_i^\star$$

$$= \mathbb{E}_{x\sim\mathcal{N}(0,1)}\left[\frac{1}{\frac{(m-2)n_i^\star}{\sigma^2+\alpha^2\frac{2\sigma^2}{n_i^\star}x^2} + \frac{2n_i^\star}{\sigma^2}}\right] + cn_i^\star = \frac{\sigma^2}{n_i^\star}\mathbb{E}_{x\sim\mathcal{N}(0,1)}\left[\frac{1}{\frac{m-2}{1+\frac{2\alpha^2}{n_i^\star}x^2}+2}\right] + cn_i^\star$$

$$= \frac{\sigma^2}{n_i^\star}\mathbb{E}_{x\sim\mathcal{N}(0,1)}\left[\frac{1}{2} - \frac{m-2}{2}\frac{1}{\frac{4\alpha^2}{n_i^\star}x^2+m}\right] + cn_i^\star$$

$$= \frac{\sigma^2}{2n_i^\star} - \frac{\sigma^2}{n_i^\star}\frac{m-2}{2}\frac{n_i^\star}{4\alpha^2}\mathbb{E}_{x\sim\mathcal{N}(0,1)}\left[\frac{1}{x^2+\frac{mn_i^\star}{4\alpha^2}}\right] + cn_i^\star$$

$$= \frac{\sigma^2}{2n_i^\star} - \frac{\sigma^2}{4\alpha^2}\frac{m-2}{2}\exp\left(\frac{mn_i^\star}{8\alpha^2}\right)\mathrm{Erfc}\left(\sqrt{\frac{mn_i^\star}{8\alpha^2}}\right)\sqrt{\frac{\pi}{\frac{mn_i^\star}{2\alpha^2}}} + cn_i^\star$$

$$\left(\text{In (52), let } L = \frac{mn_i^\star}{4\alpha^2}\right)$$

$$= \text{RHS of (50).}$$

To prove the second statement of Lemma 13, by (24) and the dominated convergence theorem, we have:

$$p'(n_i^\star) = \mathbb{E}_{x\sim\mathcal{N}(0,1)}\left[-\sigma^2\frac{1+\frac{(m-2)n_i^\star}{\left(1+\alpha^2\left(\frac{1}{n_i^\star}+\frac{1}{n_i^\star}\right)x^2\right)^2}\frac{\alpha^2x^2}{n_i^{\star 2}}}{\left(\frac{(m-2)n_i^\star}{1+\alpha^2\left(\frac{1}{n_i^\star}+\frac{1}{n_i^\star}\right)x^2}+n_i^\star+n_i^\star\right)^2}\right] + c$$

(By (24) and dominated convergence theorem)

$$= -\sigma^2\mathbb{E}_{x\sim\mathcal{N}(0,1)}\left[\frac{1+\frac{(m-2)n_i^\star}{\left(1+\frac{2\alpha^2}{n_i^\star}x^2\right)^2}\frac{\alpha^2x^2}{n_i^{\star 2}}}{\left(\frac{(m-2)n_i^\star}{1+\frac{2\alpha^2}{n_i^\star}x^2}+2n_i^\star\right)^2}\right] + c$$

$$= -\frac{\sigma^2}{n_i^{\star 2}}\mathbb{E}_{x\sim\mathcal{N}(0,1)}\left[\frac{1+\frac{(m-2)n_i^\star\alpha^2x^2}{\left(n_i^\star+2\alpha^2x^2\right)^2}}{\left(\frac{(m-2)n_i^\star}{n_i^\star+2\alpha^2x^2}+2\right)^2}\right] + c$$

$$= -\frac{\sigma^2}{4n_i^{\star 2}}\mathbb{E}_{x\sim\mathcal{N}(0,1)}\left[\frac{4\left(n_i^\star+2\alpha^2x^2\right)^2+4(m-2)n_i^\star\alpha^2x^2}{((m-2)n_i^\star+2(n_i^\star+2\alpha^2x^2))^2}\right] + c$$

$$= -\frac{\sigma^2}{4n_i^{\star 2}}\mathbb{E}_{x\sim\mathcal{N}(0,1)}\left[1+\frac{-(m-2)^2n_i^{\star 2}-4(m-2)n_i^\star\left(n_i^\star+2\alpha^2x^2\right)+4(m-2)n_i^\star\alpha^2x^2}{((m-2)n_i^\star+2(n_i^\star+2\alpha^2x^2))^2}\right] + c$$

$$= -\frac{\sigma^2}{4n_i^{\star 2}} \mathbb{E}_{x \sim \mathcal{N}(0,1)} \left[ 1 + (m-2)n_i^{\star} \frac{-(m-2)n_i^{\star} - 4(n_i^{\star} + 2\alpha^2 x^2) + 4\alpha^2 x^2}{(4\alpha^2 x^2 + mn_i^{\star})^2} \right] + c$$

$$= -\frac{\sigma^2}{4n_i^{\star 2}} \mathbb{E}_{x \sim \mathcal{N}(0,1)} \left[ 1 + (m-2)n_i^{\star} \frac{-(m+2)n_i^{\star} - 4\alpha^2 x^2}{(4\alpha^2 x^2 + mn_i^{\star})^2} \right] + c$$

$$= -\frac{\sigma^2}{4n_i^{\star 2}} + \frac{\sigma^2}{4n_i^{\star 2}}(m-2)n_i^{\star} \mathbb{E}_{x \sim \mathcal{N}(0,1)} \left[ \frac{(4\alpha^2 x^2 + mn_i^{\star}) + 2n_i^{\star}}{(4\alpha^2 x^2 + mn_i^{\star})^2} \right] + c$$

$$= -\frac{\sigma^2}{4n_i^{\star 2}} + \frac{\sigma^2}{4n_i^{\star 2}}(m-2)n_i^{\star} \mathbb{E}_{x \sim \mathcal{N}(0,1)} \left[ \frac{1}{4\alpha^2 x^2 + mn_i^{\star}} + \frac{2n_i^{\star}}{(4\alpha^2 x^2 + mn_i^{\star})^2} \right] + c$$

$$= -\frac{\sigma^2}{4n_i^{\star 2}} + \frac{\sigma^2}{4n_i^{\star 2}}(m-2)n_i^{\star} \mathbb{E}_{x \sim \mathcal{N}(0,1)} \left[ \frac{1}{4\alpha^2} \frac{1}{x^2 + \frac{mn_i^{\star}}{4\alpha^2}} + \frac{2n_i^{\star}}{16\alpha^4} \frac{1}{\left(x^2 + \frac{mn_i^{\star}}{4\alpha^2}\right)^2} \right] + c$$

$$= c - \frac{\sigma^2}{4n_i^{\star 2}} + \frac{\sigma^2}{4n_i^{\star 2}}(m-2)n_i^{\star} \left( \frac{1}{4\alpha^2} + \frac{2n_i^{\star}}{16\alpha^4} \frac{1 - \frac{mn_i^{\star}}{4\alpha^2}}{\frac{mn_i^{\star}}{2\alpha^2}} \right) \exp\left( \frac{mn_i^{\star}}{8\alpha^2} \right) \mathrm{Erfc}\left( \sqrt{\frac{mn_i^{\star}}{8\alpha^2}} \right) \sqrt{\frac{\pi}{\frac{mn_i^{\star}}{2\alpha^2}}}$$

$$+ \frac{\sigma^2}{4n_i^{\star 2}}(m-2)n_i^{\star} \frac{2n_i^{\star}}{16\alpha^4} \frac{1}{\frac{mn_i^{\star}}{2\alpha^2}}$$

$$\left( \text{In (52) and (53) and let } L = \frac{mn_i^{\star}}{4\alpha^2} \right)$$

$$= c - \frac{\sigma^2}{4n_i^{\star 2}} \left( 1 - \frac{(m-2)n_i^{\star}}{4\alpha^2 m} \right)$$

$$+ \frac{\sigma^2}{4n_i^{\star 2}}(m-2)n_i^{\star} \left( \frac{1}{4\alpha^2} + \frac{1}{4m\alpha^2} - \frac{n_i^{\star}}{16\alpha^4} \right) \exp\left( \frac{mn_i^{\star}}{8\alpha^2} \right) \mathrm{Erfc}\left( \sqrt{\frac{mn_i^{\star}}{8\alpha^2}} \right) \sqrt{\frac{\pi}{\frac{mn_i^{\star}}{2\alpha^2}}}$$

$$= c - \frac{\sigma^2}{4n_i^{\star 2}} \left( 1 - \frac{(m-2)n_i^{\star}}{4\alpha^2 m} \right)$$

$$+ \frac{\sigma^2}{4n_i^{\star 2}}(m-2)n_i^{\star} \frac{\alpha\sqrt{2\pi}}{\sqrt{mn_i^{\star}}} \frac{n_i^{\star}}{16\alpha^4} \left( \frac{4\alpha^2}{mn_i^{\star}}(m+1) - 1 \right) \exp\left( \frac{mn_i^{\star}}{8\alpha^2} \right) \mathrm{Erfc}\left( \sqrt{\frac{mn_i^{\star}}{8\alpha^2}} \right)$$

$$= \text{RHS of (51)}$$

$\square$

We will now prove the statements in (52) and (53). Both statements follow from the Lemma below by substituting $t = 1/2$.

**Lemma 14.** *For all $t \geq 0$ and some $L > 0$,*

$$I(t) := \int_{-\infty}^{\infty} \frac{1}{L + x^2} \frac{1}{\sqrt{2\pi}} \exp(-tx^2) dx = \exp(Lt) \, \mathrm{Erfc}(\sqrt{Lt}) \sqrt{\frac{\pi}{2L}}$$

$$J(t) := \int_{-\infty}^{\infty} \frac{1}{(L + x^2)^2} \frac{1}{\sqrt{2\pi}} \exp(-tx^2) dx = \sqrt{\frac{\pi}{2L}} \left( \frac{1}{2L} - t \right) \exp(Lt) \, \mathrm{Erfc}(\sqrt{Lt}) + \frac{\sqrt{t}}{\sqrt{2L}}$$

*Proof.* We derive $I(t)$ and $J(t)$ as the solutions to two ODEs and solve the ODEs to obtain the results. Firstly, by calculation:

$$-I'(t) + LI(t) = \int_{-\infty}^{\infty} \frac{x^2 + L}{L + x^2} \frac{1}{\sqrt{2\pi}} \exp(-tx^2) dx = \frac{1}{\sqrt{2t}}.$$

and

$$I(0) = \int_{-\infty}^{\infty} \frac{1}{L + x^2} \frac{1}{\sqrt{2\pi}} dx = \sqrt{\frac{\pi}{2L}}.$$

This means $I(t)$ satisfies the following ODE:

$$\begin{cases} -I'(t) + LI(t) = \frac{1}{\sqrt{2t}} \\ I(0) = \sqrt{\frac{\pi}{2L}} \end{cases} . \tag{54}$$

We solve (54) by multiplying integrating factor $-\exp(-Lt)$:

$$\exp(-Lt)I'(t) - L\exp(-Lt)I(t) = -\frac{1}{\sqrt{2t}}\exp(-Lt)$$

Note that the LHS is the derivative of $\exp(-Lt)I(t)$, the ODE becomes:

$$\frac{d}{dt}\left(\exp(-Lt)I(t)\right) = -\frac{1}{\sqrt{2t}}\exp(-Lt)$$

Integrating both sides over $t$, we get:

$$\exp(-Lt)I(t) = -\int \frac{1}{\sqrt{2t}}\exp(-Lt)dt = -\int \frac{2}{\sqrt{2L}}\exp(-Lt)d\sqrt{Lt} = \operatorname{Erfc}(\sqrt{Lt})\sqrt{\frac{\pi}{2L}} + C,$$

where we use integration by substitution for the last two equalities and $C$ is some constant that does not depend on $t$. This means $I(t)$ satisfies the following form:

$$I(t) = \exp(Lt)\left(\operatorname{Erfc}(\sqrt{Lt})\sqrt{\frac{\pi}{2L}} + C\right)$$

Using the initial condition $I(0) = \sqrt{\frac{\pi}{2L}}$ and the fact that $\operatorname{Erfc}(0) = 0$, we conclude that $C = 0$. Thus

$$I(t) = \exp(Lt)\operatorname{Erfc}(\sqrt{Lt})\sqrt{\frac{\pi}{2L}}.$$

We can similarly derive an ODE for $J(t)$. By calculation:

$$-J'(t) + LJ(t) = \int_{-\infty}^{\infty} \frac{x^2 + L}{(L + x^2)^2}\frac{1}{\sqrt{2\pi}}\exp(-tx^2)dx = I(t)$$

$$J(0) = \int_{-\infty}^{\infty} \frac{1}{(L + x^2)^2}\frac{1}{\sqrt{2\pi}}dx = \frac{1}{2L^{3/2}}\sqrt{\frac{\pi}{2}}$$

Thus $J(t)$ satisfies the following ODE:

$$\begin{cases} -J'(t) + LJ(t) = I(t) \\ J(0) = \frac{1}{2L^{3/2}}\sqrt{\frac{\pi}{2}} \end{cases} . \tag{55}$$

We similarly multiply integrating factor $-\exp(-Lt)$ and integrate both sides:

$$\int_0^t d\exp(-Lx)J(x) = -\int_0^t I(x)\exp(-Lx)dx = -\int_0^t \operatorname{Erfc}(\sqrt{Lx})\sqrt{\frac{\pi}{2L}}dx$$

$$= -\left(x\operatorname{Erfc}(\sqrt{Lx})\sqrt{\frac{\pi}{2L}}\Big|_0^t + \int_0^t x\frac{\exp(-Lx)}{\sqrt{2x}}dx\right)$$

(Integration by parts)

$$= -t\operatorname{Erfc}(\sqrt{Lt})\sqrt{\frac{\pi}{2L}} - \frac{\sqrt{2}}{L^{3/2}}\int_0^{\sqrt{Lt}} y^2\exp(-y^2)dy$$

$$\left(\text{Change of variable: } y = \sqrt{Lx}\right)$$

$$= -t\operatorname{Erfc}(\sqrt{Lt})\sqrt{\frac{\pi}{2L}} + \frac{\sqrt{2}}{L^{3/2}}\left(\frac{1}{2}y\exp(-y^2)\Big|_0^{\sqrt{Lt}} - \int_0^{\sqrt{Lt}} \frac{1}{2}\exp(-y^2)dy\right)$$

(Integration by parts)

$$= -t\operatorname{Erfc}(\sqrt{Lt})\sqrt{\frac{\pi}{2L}} + \frac{\sqrt{2}}{L^{3/2}}\frac{1}{2}\sqrt{Lt}\exp(-Lt) - \frac{\sqrt{2}}{L^{3/2}}\int_0^{\sqrt{Lt}} \frac{1}{2}\exp(-y^2)dy$$

$$= -t\operatorname{Erfc}(\sqrt{Lt})\sqrt{\frac{\pi}{2L}} + \frac{\sqrt{t}}{\sqrt{2}L}\exp(-Lt) - \frac{\sqrt{\pi}}{2\sqrt{2}L^{3/2}}\operatorname{Erf}\left(\sqrt{Lt}\right)$$

(By definition of Erf)

$$= -t\operatorname{Erfc}(\sqrt{Lt})\sqrt{\frac{\pi}{2L}} + \frac{\sqrt{t}}{\sqrt{2}L}\exp(-Lt) - \frac{\sqrt{\pi}}{2\sqrt{2}L^{3/2}}\left(1 - \operatorname{Erfc}\left(\sqrt{Lt}\right)\right)$$

(By definition of Erfc)

$$= \left(\frac{1}{2L} - t\right)\operatorname{Erfc}(\sqrt{Lt})\sqrt{\frac{\pi}{2L}} + \frac{\sqrt{t}}{\sqrt{2}L}\exp(-Lt) - J(0)$$

(By (55))

This means:

$$J(t) = \exp(Lt)\left(\int_0^t d\exp(-Lx)J(x) + J(0)\right)$$
$$= \exp(Lt)\left(\left(\frac{1}{2L} - t\right)\operatorname{Erfc}(\sqrt{Lt})\sqrt{\frac{\pi}{2L}} + \frac{\sqrt{t}}{\sqrt{2}L}\exp(-Lt)\right)$$
$$= \sqrt{\frac{\pi}{2L}}\left(\frac{1}{2L} - t\right)\exp(Lt)\operatorname{Erfc}(\sqrt{Lt}) + \frac{\sqrt{t}}{\sqrt{2}L}$$

$\square$

