# OpenReview forum: "Mechanism Design for Collaborative Normal Mean Estimation"
_NeurIPS.cc/2023/Conference — NeurIPS 2023 spotlight_

### Official Review · Reviewer_8RFX · 2023-06-16

**Soundness:** 3 good
**Presentation:** 3 good
**Contribution:** 3 good
**Rating:** 6
**Confidence:** 3

**Summary:**

- The authors study normal mean estimation in a collaborative setting. N agents each aim to obtain a good estimate for the unknown mean while incurring as little cost for data acquisition as possible.
- The authors show that a naive data aggregation mechanism leads to freeriding. Then, they propose a novel mechanism in which a central entity collects all data and only sends players noisy information about other players' samples, with the magnitude of the noise depending on the deviation from a player's sample to others' samples.
- The authors show that their mechanism fulfills desirable properties (Incentive compatibility, Individual rationality and Efficiency) in the single-dimensional Gaussian case and retains approximate versions of these properties more generally.
**I am keeping my positive score after reading other's reviews and the rebuttal.**

**Strengths:**

- The paper is generally well written and offers a relevant new take on data-sharing incentives in federated learning.
- The proposed mechanism does not rely on side payments and is therefore quite flexible.
- The authors consider a very general class of player strategies and still manage to prove strong theorems.

**Weaknesses:**

- The placement of "recommended strategies" within the formalism is a bit confusing, as there does not appear to be any difference in the analysis of recommended vs non-recommended strategies.
- As the recommended, desirable strategy profiles are only shown to be Nash equilbria but not dominant strategies, some discussion of other equilbria would be nice (in particular, are all equilbria essentially equivalent to the recommended strategies modulo some simple transformation, or are there very different equilbria?)
- Some very relevant citations seem to be missing from the related work section. In particular, the proposed mechanism is very similar to the idea of [peer prediction](https://pubsonline.informs.org/doi/abs/10.1287/mnsc.1050.0379) and connections to that strain of work could be highlighted better.
- Minor comments:
    - 256-258 contains "simply" twice in the same sentence

    - 352 seems to contain a grammar error/typo ("given her")

**Questions:**

- Is it necessary for the affordance to contain $\eta$, when the players can compute that from $Y$ and $Z$?
- Should $2\sigma \sqrt{c/m}$ in line 221 use $cm$ instead of the fraction?
- Why is Algorithm 1 using $m\geq 4$ as the cutoff, while 221 suggests a phase transition for $m\geq 3$?
- Is the general form of the allowances necessary? Since the goal is mean estimation I would imagine that allowances that only use the mean (like in section 5) could simplify the exposition and analysis.

**Limitations:**

- The scope of the paper is firmly limited to the (fundamental) problem of mean estimation.
- All players are assumed to be perfectly homogeneous in the sense that they have the exact same data acquisition costs and their samples have the same conditional means and variances. However, I do expect the proof intuitions to extend beyond that case.

---

> ### Author Rebuttal · Authors · 2023-08-09
>
> Thank you for the comments and suggestions.
>
> *Recommended vs non-recommended strategies:* We have shown that the recommended strategies are a Nash equilibrium, which means that when all other agents are following the recommended strategies, then the best response for an agent is to also follow the recommended strategy. Moreover, if all agents follow the recommended strategies, the social penalty is small.
>
> *Regarding other equilibria:* This is an interesting question. Due to the complexity of the space of estimators, it's not clear to us what other optimal estimators could lead to other Nash equilibria. You are correct though, that there are multiple Nash equilibria that could be similar to each other. For instance, all agents can add some constant $a$ to each sample they collect, and then subtract $a$ from the final estimate. Each value of $a$ would correspond to different Nash equilibria.
>
> *References on peer prediction:* Thank you for mentioning the references on peer prediction. We will add these to our related work section.
>
> *Inclusion of $\eta$:* We chose to present it this way for clarity. We thought it would be clearer if all numerical quantities in Algorithm 1 were hidden (it is crucial that the random values in line 14 be hidden), and the mechanism reveal what is necessary. See lines 142--143. However, you are correct that if $\alpha$ was also published as part of the mechanism, then revealing $\eta$ would not be necessary.
>
> *Line 221:* The current formula in line 221 is accurate.
>
> *$m\ge3$ vs $m\ge4$:*
> $m\ge3$ is the threshold at which the agent achieves the smallest penalty by collecting no data points when other agents are submitting $\sigma/\sqrt{cm}$ points in the mechanism described in line 209.
> While $m\ge4$ is the threshold at which the equilibrium in the mechanism described in line 209 is approximately efficient with $PR \le 2$.
>
> *Is the general form of allowance [allocations] necessary? Why not use something similar to Section 4.2?* We study general allocation spaces for three reasons:
>
> - First, we do wish to point out that even if the mechanism returns a mean, the agent need not simply accept it (like in Section 4.2). The agents could very well post-process this estimate to obtain a more accurate estimate. In fact, in Appendix 4.2, we show that under the more general strategy space, it is possible for agents to use a convex combination of the mean returned by the mechanism and the mean of their original dataset to reduce their MSE. They can then leverage this insight to collect less data and obtain a lower overall penalty.
> - Second, you are correct that a mechanism that simply returns the estimate in (8) instead of the 3-tuple, will have a similar guarantee to our current mechanism (agents will submit truthfully and accept this estimate). However, this will at best marginally simplify the presentation of the mechanism, while the analysis would be exactly the same.
> - Third, this form demonstrates that we have studied the problem in its fullest generality and better elucidates our contributions. For instance, if we had only studied mechanisms which return an estimate (i.e a scalar value), it could lead to other questions such as, "Can you return something else instead of a simple estimate and get better guarantees?".

---

> > ### Comment · Reviewer_8RFX · 2023-08-11
> >
> > Thank you for the response.
> >
> > Regarding recommended strategies: Do I understand correctly, that "recommended strategy" is just a name given to the strategies at the particular analyzed equilibrium, and that there is no formal meaning of the term beyond that?
> >
> > Regarding other equilbria: I see that these simple other equilbria exist. Do you have an intuition on whether there might also be equilibria that are less socially optimal (i.e. agents corrupt the samples in a way that can not be corrected for)?

---

> > > ### Author Response · Authors · 2023-08-13
> > >
> > > Thank you for your questions and your patience. Both these questions are related, so we will answer the second question first.
> > >
> > > *Are you aware of other Nash equilibria?* Despite trying, we are unable to find other Nash equilibria that are less social optimal and are hence unable to bound the price of *anarchy*. We will clarify this in the manuscript.
> > >
> > > *Regarding recommended strategies:* Yes, you are correct that the 'recommended strategies' are the Nash equilibrium which we have analyzed and where the efficiency is also very good. Recommending a set of strategies is a way for the mechanism designer to communicate a good Nash equilibrium to the agents (as it may not be obvious at the outset).

---

### Official Review · Reviewer_T2ys · 2023-07-05

**Soundness:** 3 good
**Presentation:** 3 good
**Contribution:** 3 good
**Rating:** 6
**Confidence:** 4

**Summary:**

The paper considers a collaborative mean estimation
setting where a set of agents can all collect
i.i.d samples from an underlying Gaussian distribution,
and their goal is to share data with each other in order
to estimate the mean of the distribution.
Each agent has a fixed cost for collecting each sample
and its negative utility equals the sum of the estimation error and
the sampling cost.
Naturally, each agent has an incentive to free-ride
and under-collect, reporting false data to
the data collecter and using other agents' data instead.
The goal is to design a mechanism that can incentivize
the agents to share their data more truthfully, in order
to ensure successfull mean estimation.

The authors propose an
incentive compatible and individually rational mechanism based on the idea
of "punishing" agents whose estimates deviate too much
from the true mean by sharing false data with them,
and use a minimax estimator on the collected
dataset to estimate the mean.
The authors show that the mechanism's social
penalty is at most twice the global optimum.


**Strengths:**

The authors propose an interesting problem, and provide a nice solution to it.
The paper is well-written.
The result is somewhat surprising and I believe may be important for future research.

**Weaknesses:**

The presentation of the result suggests that the mechanism is incentive
compatible globally and not just in the equilibrium point. This
should be clarified earlier in the paper.


**Questions:**

Can the authors point to a reference for their definition of IC and IR?
Specifically, the definitions I'm familiar with (e.g., see [1]) require the mentioned properties
to hold in any point, not just at the equilibrium.
In any case, it would be better to emphasize that the mentioend claims only hold
at the equilibrium early on. As is, the abstract and introduction suggest
to me that the properties hold globally, which is a much stronger claim.

Can the authors provide a formal proof of the footnote in Section 4? While the claim is not central to their paper,
I think it should be either be proved formally, or the authors should adjust the language of the footnote
so it is clear that they do not have a proof.

Also, regarding the appendix, is there any reason the authors have chosen to put appendix F in the end, rather than the beginning?

**Limitations:**

---

> ### Author Rebuttal · Authors · 2023-08-09
>
> Thank you for the comments and suggestions.
>
> *Definition of IC and IR:* We agree with you on this. There are two common notions of IC that are used in the literature, dominant strategy (DSIC) and Bayes-Nash (BNIC). We don't have a DSIC and while we have a Nash equilibrium, we are clearly non-Bayesian so we couldn't use BNIC either. However, we agree that IC might be taken to mean DSIC, so we will switch to "Nash Incentive Compatibility" as a compromise.
> As for IR, we will also clarify that it only holds at the Nash equilibrium. (We do wish to point out that it may be difficult to design an IR mechanism regardless of the strategies of the other agents. For instance, if all agents except one agent are malicious and submit false data, then the lone agent will lose out.)
>
> *Proof of footnote*: Are you implying the footnote on page 4 (and not footnote 4)? Yes, we will include proof sketch. In fact, our current analysis uses a sequence of Normal priors and Bayesian estimators to analyze the maximum (frequentist) risk. We believe these calculations can be used as is, to prove a result for Normal priors.  The techniques can also be applied to non-normal priors; however, the results will be prior-dependent, and if the prior is complex, the results may be hard to interpret.
>
> *Order in the appendix:* Appendix F consists of technical (algebraic) lemmas that are important but not central to most of the proofs. For a more coherent presentation, we decide to defer them to the end.

---

> > ### Comment · Reviewer_T2ys · 2023-08-15
> >
> > Thank you for your response.
> > I don't have any additional questions.

---

### Official Review · Reviewer_jdRd · 2023-07-06

**Soundness:** 4 excellent
**Presentation:** 4 excellent
**Contribution:** 3 good
**Rating:** 7
**Confidence:** 4

**Summary:**

The author consider the problem of designing a data-sharing mechanism that encourages a group of $m$ agents to share their iid collected samples truthfully and further uses the shared data to refine their estimations of the normal mean. To ensure truthful reporting, the mechanism introduces additional noise into the shared data.  The amount of noise is determined based on the discrepancy between the mean of an agent's reported data and the mean reported by other agents, with the noise variance increasing proportionally. The authors demonstrate that this mechanism achieves both individual rationality and incentive compatibility. In addition, the mechanism is also efficient compared to the minimum social penalty. The authors further extend the result to estimation of the mean in high-dimensional settings with a bounded variance.

**Strengths:**

The problem of data sharing is well motivated by real world applications and has gained great attention in recent years. The difficulties associated with this type of problem often involve determining the appropriate pricing strategy for data to incentivize genuine data collection while discouraging data fabrication. This paper addresses the problem in an elegant way in the language of normal mean estimation.

The authors explore the methods of inserting noise into the reallocated data whose variance depends on the quality of an agent’s report. The idea has occurred in many previous work, but the authors show that by carefully designing the noise level, the mechanism can be at the same time IC, IR and efficient. Moreover, the results hold for any number of agents and can also be extended to non-normal distribution. I believe that this work may be of interest to the community.

The paper is well-written and the results are sound.

**Weaknesses:**

It seems that the current result is quite limited to the specific form of the social/individual penalty. The proof of the IC also seems to heavily rely on this specific structure. Can the results be extended to more general penalty functions?


**Questions:**

What if there exists a common prior of $\mu$? Does the result still hold in general?


**Limitations:**

No concerns here.

---

> ### Author Rebuttal · Authors · 2023-08-09
>
> Thank you for the comments and suggestions.
>
> *Specific form of penalty:* Yes, you are correct. We do believe that these results can be extended to other penalty forms and supervised learning problems, but we may need to relax from an exact to an approximate Nash equilibrium. This is because for other penalty forms, it is hard to design exactly minimax-optimal estimators and it is customary to settle for rate(order)-optimal estimators. See 2nd para of Section 5. We are working on extending the ideas in this work to more general settings.
>
> *When there is a common prior $\mu$:* Yes, our results can be adapted when there is a common prior -- see footnote 1. In fact, our analysis constructs a sequence of Normal priors to analyze the maximum risk, so if this prior is Normal, you can directly use the calculations in our proofs to obtain a Bayesian result which depends on the priors. The techniques can also be applied to non-normal priors; however, the results will be prior-dependent, and if the prior is complex, the results may be hard to interpret.

---

> > ### Comment · Reviewer_jdRd · 2023-08-18
> >
> > Thank you for your response. I have no further questions.

---

### Official Review · Reviewer_ipny · 2023-07-06

**Soundness:** 4 excellent
**Presentation:** 4 excellent
**Contribution:** 3 good
**Rating:** 6
**Confidence:** 4

**Summary:**

The paper designs a mechanism that collects data from n agents to estimate the mean of a Gaussian distribution. The agents incur costs to collect data, they can misreport data, and they strategically choose the level of effort and the data to report. They propose a mechanism that corrupts the returned datasets according to the difference between an agent's reported data and others' data. They prove that their mechanism is IC, IR, and achieves a 2-approximation of the optimal social welfare.

**Strengths:**

The paper studies an interesting problem and the presentation is clear.

**Weaknesses:**

The result may be overshadowed by the results in (Cai et al., 2015), which is not cited in the current paper. Although the problem is formulated without payments, it is pretty much a mechanism design problem with payments, because the designer knows the agents' utility function exactly and can add arbitrary noise to adjust an agent's utility freely. This is very similar to adding a numerical payment to the allocation function, which makes the problem very close to (Cai et al., 2015). However, (Cai et al., 2015) achieves a much stronger result: they are able to achieve optimal social welfare at a dominant strategy equilibrium. In addition, when their mechanism is used, the agents do not have the incentive to misreport data, which means that truthfully reporting data will be a weak BNE. In this paper, only 2-approximation is achieved at a BNE, which is pretty far from the potential optimal. It may not be straightforward to apply (Cai et al., 2015) because adding noise can only give negative payments, but it is also not clear whether (Cai et al., 2015) can yield a better result than 2-approximation.

Cai et al., 2015, "Optimum Statistical Estimation with Strategic Data Sources"

**Questions:**

Is it possible to use the VCG-like mechanism from (Cai et al., 2015) in your problem?

**Limitations:**

Yes.

---

> ### Author Rebuttal · Authors · 2023-08-09
>
> Thank you for mentioning the paper by (Cai et al., 2015). This is a nice and relevant paper that we were not aware of, but will be sure to include it in the revision. While Cai et al., 2015 study a general supervised learning problem, when applied to mean estimation, their setting is more restrictive than ours.
>
> The **primary** difference is that the agents' strategy spaces are different. In the main part of our paper (Section 2 and 3), the agents can choose how much data to collect, what to submit, and most importantly how to estimate the mean from the information received from the mechanism. In (Cai et al., 2015), the agents can only choose how much effort to exert (where the effort is similar in spirit to the number of data points in our paper). This means that the mechanism design problem is significantly more challenging in our setting. Based on this, we will highlight the key differences below, while also offering rebuttals to the reviewer's claims.
>
>
> 1. *the setting is similar to payments in Cai et al, since the designer knows the utility and can add noise freely:* There are two key components to designing a mechanism under our strategy space. First, the mechanism should decide 'how much' to give away (which is similar to payments), and second 'how to give away', since the agents can use the information they receive to decide how to estimate the mean. While several prior work have studied the former component (e.g Cai et al, Karimireddy et al, Blum et al), the latter component (which we believe is significantly more challenging), has not been studied to our knowledge.
> To illustrate this further, suppose the mechanism decides on a 'payment' and returns a mean estimate that corresponds to this payment. However, there is no guarantee that the agent will simply accept this mean. The agent may post process this estimate, for instance based on the original data it collected, and aim to achieve a higher 'payment' than the mechanism intended. Agents can in fact be even more strategic; they can only submit part of the data they collected, and use the remaining hidden data to refine their estimate. The mechanism should account for this strategic behavior - this explains why we had to design careful allocation spaces, corruption mechanisms, and minimax optimal estimators for the corrupted datasets.
>
> 2. *In Cai et al, the agents do not have the incentive to misreport data:* We would like to respectfully point out that this was not proved in their paper. As we pointed out above, they limit the strategy space to the amount of effort. We are looking at a 17-page COLT 2015 paper; please let us know if there is an updated version.
> (We do however believe that it is possible to prove this result for a modified version of their mechanism in a simpler setting using techniques from our paper; more on this below)
>
> 3. *Cai et al are able to achieve optimal social welfare [penalty]:* First of all, we would like to point out that their definition of social penalty is different from ours.
> Moreover, there is no way to reduce one to the other as we are summing over all agents' errors, and each agent could have a different error since they may use different estimators and may have different allocations.
>
>    * However, it is worth observing that when we were first studying the problem in Section 4.2 (a simpler strategy space than Section 2 and 3 where the agents have to accept a mean estimate from the mechanism), we started with exactly the same corruption method to the one in Cai et al. We were able to prove that it was truthful; however, it was able to achieve, at best, a $1.5\times$ factor of the global minimum according to our definition of the social penalty. We were able to improve the mechanism with a different corruption strategy to get $(1+\epsilon)\times$ factor for arbitrarily small $\epsilon$ while still ensuring truthful reporting. This is the mechanism presented in Section 4.2.
>    * To summarize, when restricted to mean estimation, our results in 4.2 are already stronger than Cai et al: it achieves a lower social penalty according to *our* definition, and we prove that agents should submit the data truthfully (they did not). We wish to emphasize that our main contributions in Section 2 and 3 are **significantly** more challenging than in 4.2.
>
>
> 4. *Cai et al achieve a dominant strategy equilibrium:* while it is possible to prove a DSE under a restrictive strategy space, it is easy to see that there may be multiple (infinite) Nash equilibria in any nontrivial mechanism if we allow agents to alter the data they submit and then estimate/post-process the information they receive. For instance, all agents can add some constant $a$ to each sample they collect, and then subtract $a$ from the final estimate. Each value of $a$ would correspond to a different Nash equilibrium.
>
>
> Hopefully, these points alleviate your concerns that the results of Cai et al. "overshadow" this work. We will make sure to cite that paper and clarify the differences.

---

> > ### Comment · Reviewer_ipny · 2023-08-19
> >
> > Thank you for the response. But my concerns remain.
> >
> > First of all, it can be proved that the agents do not have the incentive to misreport data in (Cai et al., 2015), if we adopt the same worst-case analysis used in your paper. This is because the worst-case expected payment (equation 3) is maximized when the agent truthfully reports in the setting of normal mean estimation. If you look at the decomposition of the expected payment (the first equation below equation 3) and look at the term inside the expectation. First, the expectation of the third term is zero when the data is i.i.d. Second, agent i does not have control over the second term. And finally, the worst-case expectation of the first term is minimized when truthfully reporting, because as you cited in the paper, the sample mean is the minimax optimal estimator.
> >
> > Therefore, even if the model of Cai et al. (2015) does not explicitly allow data modification, their mechanism guarantees IC in your setting.
> >
> > For the definition of social welfare, I am not convinced that your objective function is fundamentally different from that of (Cai et al., 2015). They are both expected error + costs. The social welfare definition used in (Cai et al., 2015) is the standard definition commonly employed in mechanism design literature. There seems to be little justification for straying from this convention.

---

> > > ### Author Response · Authors · 2023-08-20
> > >
> > > Thank you for your questions.
> > >
> > > 1. To begin with, we would like to reiterate that your concerns above relate to Section 4.2, which is only a small part of our contribution (1/2 page). Our primary contributions are in Sections 2 and 3, where agents can use the data/estimate they received the way they wish to, and are not restricted to accepting the estimate provided by the mechanism. As we highlighted in our previous reply, the mechanism design problem is significantly more challenging since the strategy space is much richer (also see point 4 below). In any case, below, we shall address the questions you have raised about Section 4.2.
> > >
> > > 2. *Definition of social welfare [penalty]:*
> > > The two notions of social penalty become very different when you account for the fact that agents are not rewarded by payments, but by a model. To understand this, let us first look at the social penalty in Cai et al (using the notation in our mean estimation setting):
> > > \\[
> > > \sup_\mu E[(\hat\mu-\mu)^2] + \eta\sum_{i=1}^m c n_i,
> > > \\]
> > > where, $\hat\mu$ is the estimate of the *mechanism*, $c n_i$ is the effort by each agent, and $\eta$ is a trade-off parameter in their setting. In contrast, the social penalty for us is,
> > > \\[
> > > \sum_{i=1}^m\sup_\mu (E[(\hat\mu_i-\mu)^2] + c n_i).
> > > \\]
> > > where, $\hat\mu_i$ is *the estimate assigned by the server to agent $i$* and $c n_i$ is the effort by each agent. This definition is justified in settings where agents are interested in the accuracy of their own estimates and not in any payments. The difference in the settings lies in the fact that when there are payments involved, the payments do not affect the social penalty (as the mechanism's negative is the agent's positive). However, if you are rewarding the agent with a better/worse model, that affects the social penalty in our setting.
> > > To illustrate this further, suppose one agent does not collect a sufficient amount of data. In Cai et al, the mechanism will still choose the best possible estimator $\hat\mu$ for itself to minimize the social penalty. It will pay only a small amount to the agent to penalize its lower effort, but this does not affect the penalty.
> > > On the other hand, in our setting, the mechanism cannot simply assign this best possible estimate $\hat\mu$ as each agent's estimate $\hat\mu_i$ as it also needs to reward/penalize agents via the estimate $\hat\mu_i$.
> > > So, our mechanism should offer a poor estimate $\hat\mu_i$ to this agent, which will reduce the social penalty.
> > >
> > >     * As we mentioned in our previous reply, when we studied a similar corruption strategy used by Cai et al in our setting, the best we could obtain was a $1.5\times$ factor over the global minimum in our setting. We had to improve the mechanism further, to obtain a $(1+\epsilon)\times$ factor.
> > >     * We do wish to clarify that we view the results in Cai et al as complementary to our results in Section 4.2 (in case the phrasing in our previous reply was confusing). We both study different settings and design arguably optimal mechanisms for our respective problems.
> > >
> > > 3. *IC proof intuition:* In our proof of Theorem 3, we do use a similar intuition to prove IC to the one that you mentioned above, although we have to deal with a more complex corruption strategy and the differences in the settings highlighted above. However, as we mentioned in our previous post, we do not think this proof is anywhere nearly as challenging as the proof of IC for our main result (Theorem 1) in Section 2 and 3. See the proof sketch in Section 3.1.

---

> > > > ### Author Response · Authors · 2023-08-20
> > > >
> > > > 4. *Differences between Sections 2/3 and Section 4.2:* As we mentioned in our previous reply, while several papers, including Cai et al, have studied the problem of incentivizing effort in several settings, none of them have studied how strategic considerations change when the agents are allowed to use the data (or the estimate/model) they receive freely. They have all assumed that the agents will simply use the model returned by the mechanism.
> > > >
> > > >     * As we mentioned in the paragraph after Theorem 3 (lines 386-391), if we apply our mechanism in Section 4.2 to the problem in Sections 2 \& 3, the optimal action for the agent is not to simply accept the estimate provided by the mechanism. Instead, using a convex combination of the mean returned by the mechanism and the mean of its own data will improve the estimation error. The agent could even go further: they may use this insight that they could get a lower penalty than the mechanism intended; hence, they could collect less data, use a better estimator, and lower their overall penalty. This adversely affects the social penalty since less data is being collected overall.
> > > >     * It is however not clear to us what the optimal strategy for an agent is in this mechanism when applied to Sections 2 \& 3. While we can prove that accepting the estimator from the mechanism is sub-optimal as mentioned above, computing the optimal strategy or finding an equilibrium appears to be very challenging for this mechanism. This is why we designed a different corruption strategy in our main mechanism in Algorithm 1.

---

> > > > ### Comment · Reviewer_ipny · 2023-08-20
> > > >
> > > > Thank you for the clarification. I am convinced that the social welfare is different. However, the motivation for considering this more complex corruption strategy space is not clear to me.
> > > >
> > > > Suppose I just use the mechanism from (Cai et al. 2015) in your setting, will the agents have the incentive to modify their data or not accept the straightforward estimator? If not, why do you have to consider this more complex strategy space? As you mentioned, the mechanism in Cai et al. (2015) achieves a 1.5 approximation factor, which is better than the 2-approximation in Theorem 1. Is it the case that their mechanism is better, even in your setting with a more complex strategy space (if the agents do not have the incentive to modify their data or not accept the straightforward estimator when their mechanism is used)?

---

> > > > > ### Author Response · Authors · 2023-08-21
> > > > >
> > > > > Thank you for the prompt reply. We think there may be some confusion as to which mechanisms apply to which strategy spaces.
> > > > >
> > > > > 1) First, we would like to remind you that our paper presents and studies three mechanisms: the main one in Algorithm 1 related to Section 2 and 3, and two simpler ones in Section 4.1 and 4.2 respectively corresponding to restricted strategy spaces. Of these, the setting in section 4.2 is most similar to Cai et al. The mechanisms are also very similar, but we use a slightly different corruption method to the one used by Cai et al.
> > > > >
> > > > > 2) Second, consider the setting in Section 4.2 where the agents *have to* accept the estimator from the mechanism. Here, the method of Cai et al achieves a $1.5\times$ factor. Our corruption method in Section 4.2 improves this to a $(1+\epsilon)$ factor for all $\epsilon>0$. In this restricted strategy space, it would not be necessary (or advisable) to use our main mechanism in Algorithm 1 as it only achieves a $2\times$ factor improvement.
> > > > >
> > > > > 3) Third, consider our main setting in Sections 2 and 3 with the richer strategy space where agents receive data and can construct their own estimator. See responses to your questions below:
> > > > >
> > > > > 3.1) *Suppose I just use the mechanism from (Cai et al. 2015) in your setting, will the agents have the incentive to modify their data or not accept the straightforward estimator?*
> > > > >
> > > > > Yes! We have already answered this in point 4 of our previous reply and in lines 386-391 of the paper, but will elaborate below.
> > > > > - First, we should point out that it is not clear what the Nash equilibrium of the mechanism by Cai et al is in the richer strategy space as they simply have not studied this setting. We tried to analyze the Nash equilibrium of the mechanism in Section 4.2 in the richer strategy space of Sections 2/3, but this appears to be challenging. In fact, we are not even sure that an equilibrium exists. We suspect that this is the case for the corruption method by Cai et al as well, as their mechanism shares a similar structure but with a different corruption method.
> > > > > - However, it is clear that he optimal action for the agent using our mechanism in Section 4.2 (or, for that matter, a mechanism which uses the corruption method by Cai et al) is \emph{not} to simply accept the estimate provided by the mechanism. Instead, using a convex combination of the mean returned by the mechanism and the mean of its own data will improve the agent's estimation error (although this may not the optimal estimate either). As explained in the previous post, the agent could  use this insight that they could get a lower penalty than the mechanism intended; hence, they could collect less data, use a better estimator, and lower their overall penalty. For instance, if the agent uses the convex combination, we can plug in the risk of this estimator and then optimize for the amount of data the agent should collect - the answer is strictly smaller than $n^\star$. This adversely affects the social penalty since less data is being collected overall.
> > > > > - However, as mentioned above, the optimal estimator itself is hard to analyze. (All we can say now is that it is sub-optimal to accept the estimate from the mechanism, both for the mechanism in Section 4.2 or if we use the corruption by Cai et al.) Therefore, it is also challenging to compute the optimal strategy and the Nash equilibrium.
> > > > > - This is why we designed a different mechanism in Algorithm 1 for which it is possible to compute the minimax optimal estimator and subsequently a Nash equilibrium in the larger strategy space. Moreover, the social penalty of this Nash can also be bounded by a $2\times$ factor.
> > > > >
> > > > > 3.2. *Does it mean that their mechanism is better, even in your setting with a more complex strategy space?*
> > > > >
> > > > > No. You cannot claim the $1.5\times$ factor of Cai et al (or the $1+\epsilon$ factor of our mechanism in 4.2) is better  without analyzing their Nash equilibrium. As mentioned above, it is not clear if either mechanisms even have a Nash in the richer strategy space.
> > > > >
> > > > > 4) *the motivation for studying the complex strategy space is not clear to me:*
> > > > > We think this is a more natural setting to study collaborative data sharing than the setting in Sec 4.2, where multiple agents may come together and share data with each other. In this case, we have to account for how the agents will use the data, and cannot simply assume that the agents will use the estimate  provided by the mechanism. As explained in lines 36-39, lines 225-229, and lines 359-364, data fabrication becomes a serious issue when we allow agents to construct their own estimate using the information provided by the mechanism.
> > > > > The setting in Section 4.2 is only sufficient in settings such as federated learning where the estimate (model) is deployed in the agent's device and the agent cannot control it. Otherwise, we have to assume that the agent will try to minimize their penalty by post-processing the estimate or using a smart estimator.

---

> > > > > > ### Comment · Reviewer_ipny · 2023-08-21
> > > > > >
> > > > > > Thank you for the response. It would be quite surprising if an agent's optimal action when using Cai et al. is not to simply accept the estimate provided by the mechanism. It is not proven in this paper, but I will trust your claim. It should be highlighted more in the final paper, and a detailed comparison with Cai et al should be included.

---

### Official Review · Reviewer_Sj1t · 2023-07-07

**Soundness:** 3 good
**Presentation:** 3 good
**Contribution:** 3 good
**Rating:** 7
**Confidence:** 2

**Summary:**

This paper studies collaborative normal mean estimation, where strategic agents collect i.i.d samples from a normal distribution at a cost. This paper designs a "truthful" mechanism such that the strategic players will try to collect data instead of doing some "random" thing that harms the system and benefits themselves for federated learning systems.

**Strengths:**

The problem it studies is very interesting, and has the potential impact for further directions and research. The theory is solid. I think this kind of research that consider the robustness of the system / robust statistics will gain much attention for federated learning (and related) community.

**Weaknesses:**

I am not very familiar with AGT, and thus will not point out the weakness on the theory part. However I do have some minor concerns with respect to the model and motivation.

+ In this paper, the author only considers estimating the mean of Gaussian distribution from samples, which is a one-round scenario. The clients can communicate with each other and send the raw data. While in real applications, it is always not good to directly send the data because of the privacy issue. Thus the optimization problem may include several round of interactions. Is it possible to extend the current single-round results to a multi-round protocol?
+ In federated learning or collaborated data-sharing applications, different clients may have different data-distribution (in the current paper, different clients may have different mean $\mu_i$). Thus, some clients may also untruthfully report the data to take advantages for their own sake. The current model may be too easy (all clients consider the same data distribution). Is it possible to extend the current results / mechanism to the non-IID setting where different clients may care about different distribution?

**Questions:**

See the weakness section.

---

> ### Author Rebuttal · Authors · 2023-08-09
>
> Thank you for the comments and ideas.
>
> *Multi-round mechanisms and privacy:* We are actually looking at multi-round mechanisms now :) While we can build on our current paper, there is still more work needed to solve this problem. As for privacy, we believe the rigorous way to study this would be to include it as part of the agent's penalty (or utility) in the formulation. We think this is an interesting avenue for future work.
>
> *Heterogenous agents:* Studying heterogeneity will be interesting, both in terms of what the agent can collect and also in terms of what each agent wishes to estimate. We believe a 'complete' mechanism will use something similar to our mechanism when there is overlap between the agents, and some other protocol when there is no overlap. Our focus on this paper was to isolate and study the free-riding issue which is quite challenging as is.

---

> > ### Comment · Reviewer_Sj1t · 2023-08-14
> >
> > Thanks for your response. I think the paper in general is interesting and makes enough contribution (though I am not super familiar with the AGT backgrounds). I will keep my score and confidence.

---

### Official Review · Reviewer_LHqC · 2023-07-29

**Soundness:** 4 excellent
**Presentation:** 4 excellent
**Contribution:** 4 excellent
**Rating:** 8
**Confidence:** 4

**Summary:**

The authors study a collaborative normal mean estimation problem, where m strategic agents are trying to estimate the mean \mu of an unknown normal distribution with given variance. The agents can acquire samples drawn from the distribution at a cost of c per sample. In addition, each agent can share their samples with a mechanism, which will reward them by providing some of the samples other agents have submitted. Given that the agents are strategic, they can also omit or alter their samples prior to sharing, or even fabricate additional samples at no extra cost.

The strategy of each player therefore is how many samples to obtain, what to report and how to estimate \mu given the response of the mechanism. The cost of the agent has a worst case flavour: it is the cost of the samples plus the maximum expected quadratic estimation error (with the supremum taken over all possible true values of \mu and the expectation over the samples received directly or by the mechanism, which further depends on the strategies of other agents). The solution concept used is the Nash equilibrium. The mechanism needs to satisfy Incentive Compatibility (IC) and Individual rationality (IR). IC is about the players following the 'honest' strategy, which in this case is suggested by the mechanism and includes taking a certain number of samples and honestly sharing all of them. If IC holds (more on that in the questions), then following this honest strategy is the best option for any agent, as long as every other agent is also honest. IR dictates that for the honest strategy, every player should have lower cost than if playing in isolation.

The mechanism also needs to efficient, which means that the expected sum of costs (given the recommended IC strategy) should be a close approximation of the 'optimal' cost, which is the minimum that can be achieved by any mechanism and agent strategy (ignoring IC and IR constraints). In this case, it is shown that the optimal non-strategic mechanism simply collects samples are forwards them to all agents, who use a minimax estimator.

The designed mechanism called C3D collects samples from agents and then for each agent i: it splits all collected samples into two sets Z_i and Z_i', where Z_i contains min(|Y_i|, \sigma / \sqrt{c m}) samples where Y_i are the samples submitted by agent i. The samples Z_i' are then perturbed by adding random noise, increasing in the difference between Y_i and Z_i. This mechanism is both IC and IR (which is implied since submitting no samples is a valid strategy) and has an approximation ratio of 2.

The authors also consider extensions where the agents have to submit their true samples or where the mechanism itself calculates \mu (thus the agents cannot misreport and then ignore their input from their estimation). In the first case, a simpler mechanism is completely efficient, while in the second a 1+ \epsilon mechanism can be designed for all \epsilon > 0.

**Strengths:**

The setting of collaborative mean estimation is very interesting and particular care has been taken to establish a nuanced model where this question can be meaningfully posed and answered. The presentation of the paper is excellent, with all ideas communicated clearly and in the right order. Before any rigorous proof, an appropriate amount of intuition is provided.

The mechanism itself is very natural and seems robust, while achieving a good approximation.

**Weaknesses:**

No lower bound is provided for the 2-approximation.

Usually, Incentive Compatibility refers to a strategy being optimal no matter what the other players are doing, whereas here the definition given is essentially the same as the Nash equilibrium.

In any sensible mechanism IC would imply IR in this setting. Given that the paper is very well presented this is a minor point, but a bit of content could be cut and simplify the notation by restricting mechanisms wlog.

**Questions:**

None.

**Limitations:**

Yes.

---

> ### Author Rebuttal · Authors · 2023-08-09
>
> Thank you for the comments and suggestions. Yes, we agree it would be interesting to establish the lower bound for the 2-approximation, and leave it to future work.
>
> *Definiton of IC:* We agree with you on this. There are two common notions of IC  that are used in the literature, dominant strategy (DSIC) and Bayes-Nash (BNIC). We don't have a DSIC and while we have a Nash equilibrium, we are clearly non-Bayesian so we couldn't use BNIC either. However, we agree that IC might be taken to mean DSIC, so we will switch to "Nash Incentive Compatibility" as a compromise.

---

### Decision · Program_Chairs · 2023-09-21

**Decision:**

Accept (spotlight)

**Comment:**

This paper studies a collaborative normal mean estimation problem, where m strategic agents are trying to estimate the mean of an unknown normal distribution with given variance. As main contribution, the paper proposes a mechanism that can prevent free-riding and data fabrication among dishonest agents. It is proven that the proposed mechanism is incentive compatible, individual rational, and a bound on the efficiency is provided.

The reviewers found the paper to be relevant for the Neurips community, well-written, and recommend acceptance. The results presented in this paper could inspire more practical multi-round protocols for mean estimation, such as e.g. needed in many Federated Learning algorithms.

Some reviewers noted that relation to prior work should be discussed a bit more carefully (especially the reference Cai et al. 2015, "Optimum Statistical Estimation with Strategic Data Sources"). We would like to urge the authors to include this comparison (see discussion below) in the final version of the paper, and also to expand the discussion of prior work---not only comparing the different settings considered in the respective works, but also commenting on how the results of these work would compare to the proposed method if they are applied in the same (a bit less general) setting.